# IGF2BP1 phosphorylation in the disordered linkers regulates ribonucleoprotein condensate formation and RNA metabolism

Harald Hornegger[1,2,3], Aleksandra S. Anisimova[1,2,3], Adnan Muratovic [1], Benjamin Bourgeois[4,5], Elena Spinetti [6,7], Isabell Niedermoser [1,2], Roberto Covino[7,8], Tobias Madl [4,5] & G. Elif Karagöz [1,2] ✉

The insulin-like growth factor 2 mRNA binding protein 1 (IGF2BP1) is a conserved RNA-binding protein that regulates RNA stability, localization and translation. IGF2BP1 is part of various ribonucleoprotein (RNP) condensates. However, the mechanism that regulates its assembly into condensates remains unknown. By using proteomics, we demonstrate that phosphorylation of IGF2BP1 at S181 in a disordered linker is regulated in a stress-dependent manner. Phosphomimetic mutations in two disordered linkers, S181E and Y396E, modulate RNP condensate formation by IGF2BP1 without impacting its binding affinity for RNA. Intriguingly, the S181E mutant, which lies in linker 1, impairs IGF2BP1 condensate formation in vitro and in cells, whereas a Y396E mutant in the second linker increases condensate size and dynamics. Structural approaches show that the first linker binds RNAs nonspecifically through its RGG/RG motif, an interaction weakened in the S181E mutant. Notably, linker 2 interacts with IGF2BP1's folded domains and these interactions are partially impaired in the Y396E mutant. Importantly, the phosphomimetic mutants impact IGF2BP1's interaction with RNAs and remodel the transcriptome in cells. Our data reveal how phosphorylation modulates low-affinity interaction networks in disordered linkers to regulate RNP condensate formation and RNA metabolism.

RNA-binding proteins (RBPs) play important roles in post-transcriptional control of RNA[1–6]. IGF2BPs are a conserved family of RBPs that regulate RNA localization, translation and stability[7–11]. There are three IGF2BP paralogs (IGF2BP1-3) in mammals. Discovered in chicken embryos, IGF2BP1 was the founding member of the IGF2BP family[12,13]. IGF2BP1 is highly conserved in sequence and function across species (Supplemenetary Fig. 1A). It is highly expressed during mid to late embryogenesis and its expression decreases in adult tissues. In line

with embryonic functions, *Igf2bp1* knockout mice show developmental abnormalities[14]. However, IGF2BP1 expression is not restricted to early development, and it is detected later in differentiated gonads and the kidneys. Consistent with post-developmental functions, loss of IGF2BP1 in intestinal epithelial cells impairs intestinal homeostasis in adults[15,16]. IGF2BP1 is highly expressed in various tumors and its over-expression correlates with tumor aggressiveness[9,17]. Importantly, IGF2BP1 depletion impairs tumor growth, indicating that inhibition

[1]Max Perutz Laboratories Vienna, Vienna BioCenter, Vienna, Austria. [2]Medical University of Vienna, Vienna, Austria. [3]Vienna BioCenter PhD Program, Doctoral School of the University of Vienna and Medical University of Vienna, Vienna, Austria. [4]Otto Loewi Research Center, Medicinal Chemistry, Medical University of Graz, Graz, Austria. [5]BioTechMed-Graz, Graz, Austria. [6]Institute of Biophysics, Goethe University Frankfurt, Frankfurt am Main, Germany. [7]Frankfurt Institute for Advanced Studies, Frankfurt am Main, Germany. [8]Institute of Computer Science, Goethe University Frankfurt, Frankfurt am Main, Germany. ✉e-mail: guelsuen.karagoez@meduniwien.ac.at

may have therapeutic potential in cancer cells[18,19]. This link to disease underlines the importance of obtaining a mechanistic understanding of how IGF2BP1 exerts its function.

IGF2BP1 is a canonical multi-domain RBP, which contains six RNA-binding domains: two RNA recognition motif (RRM) domains and four hnRNP K homology (KH) domains that are linked by two intrinsically disordered regions (Fig. 1A). The KH domains are arranged into pseudodimers (KH1-2, and KH3-4). RNA recognition by IGF2BP1 is mediated by the KH domains, which interact with single-stranded RNAs through 4 nucleotide long recognition motifs[20]. In contrast, the RRM domains provide little specificity and promiscuously recognize dinucleotide sequences, as shown for the IGF2BP3 paralog[21,22]. These multivalent interactions increase the specificity and affinity of IGF2BPs for substrate RNAs[20,21,23].

Genome-wide cross-linking and immunoprecipitation (CLIP) studies identified a large number of IGF2BP1 targets, suggesting roles in cell growth, migration, synaptic plasticity in healthy tissues, as well as tumor growth and metastasis in cancer cells[9,17,24-27]. These data also revealed that IGF2BP1 binds to the coding regions, 5′-untranslated regions (UTRs), and 3′-UTRs of target RNAs, with the highest number

of binding sites residing in 3′-UTRs[28,29]. Since the binding sites for IGF2BP1 and microRNAs overlap, it was proposed that IGF2BP1 can stabilize RNAs by competing with the microRNA binding sites[19]. IGF2BP1 also binds to and stabilizes $N^6$ methyl adenosine-modified RNAs during heat shock stress[8]. Although IGF2BP1 has been proposed to stabilize RNAs, binding to a subset of its target RNAs correlates with destabilization[30]. However, what regulates these distinct functional outputs remains largely unknown

IGF2BP1 assembles into various ribonucleoprotein granules (RNP) to regulate RNA fate. In neurons, IGF2BP1 is part of transport granules, which transport select mRNAs from soma to neurites to regulate site-specific protein synthesis[7]. During cellular stress, IGF2BP1 is sequestered into stress granules that have been proposed to protect mRNAs from degradation until translation resumes. Intriguingly, IGF2BP1 also localizes to P-bodies, which are sites of RNA recapping and degradation[8]. Yet, the regulation of IGF2BP1 assembly into RNP granules with opposite functions is not well understood.

One well-defined mechanism that regulates IGF2BP1 function is through its phosphorylation. In the best-studied example, phosphorylation of IGF2BP1 controls its binding to the ß-actin-encoding *ACTB*

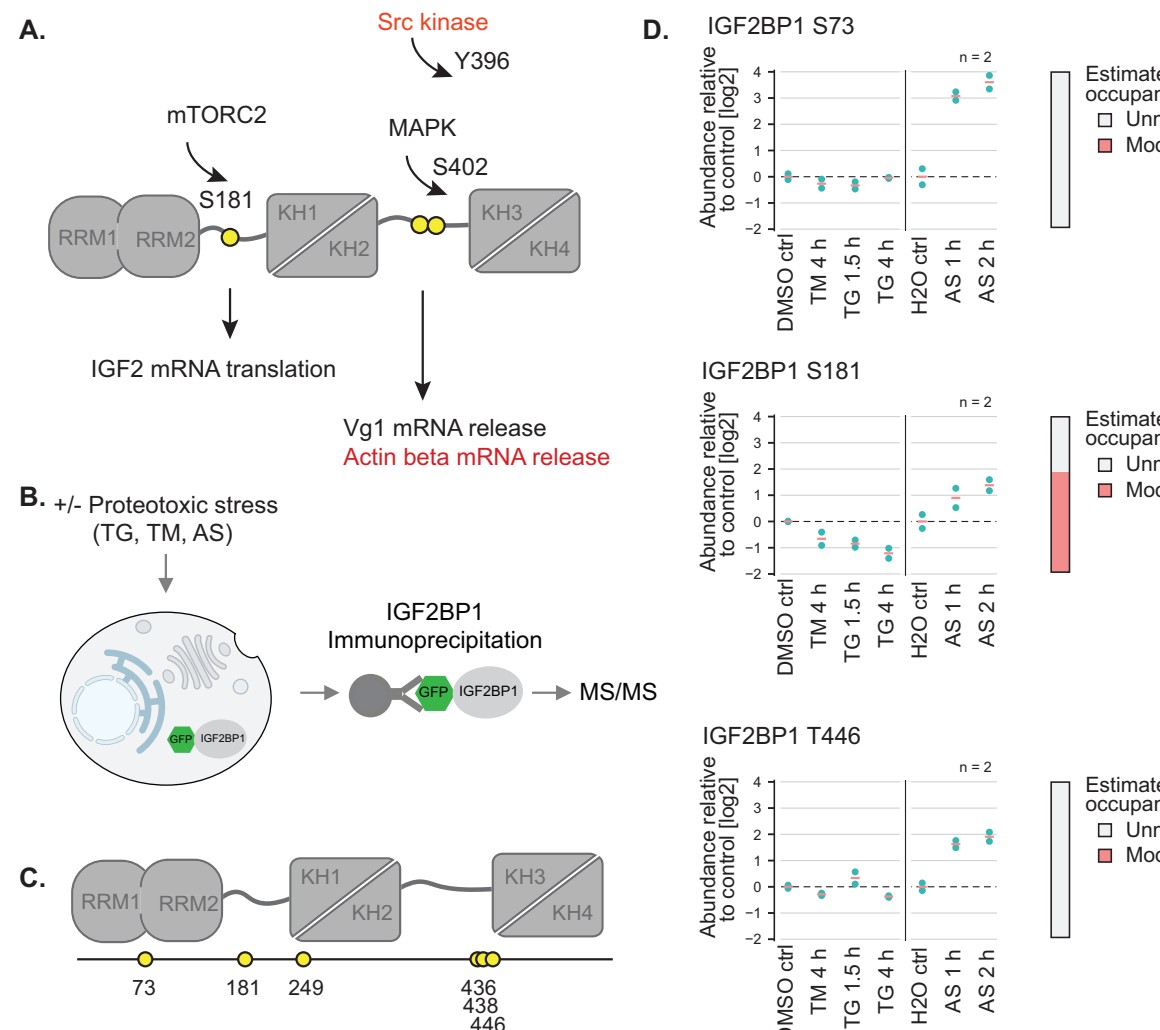

**Fig. 1 | IGF2BP1 is phosphorylated at distinct sites in a stress-dependent manner. A** Schematic depiction of IGF2BP1 domain architecture. IGF2BP1 consists of 6 RNA binding domains: two RRM domains and four KH domains, which are arranged as pseudo-dimers connected via two linkers. The well-studied phosphorylation sites (S181, Y396, S402), the respective kinases and the effect of the phosphorylation are also depicted. **B** Schematic overview of the workflow for mass spectrometry experiments to determine the stress-regulated phosphorylation

sites in IGF2BP1. **C** Representation of IGF2BP1 phosphorylation sites identified by MS analyzes. **D** Relative abundance of the indicated IGF2BP1 phosphorylation sites in cells exposed to various forms of proteotoxic stress compared to the control conditions. Tunicamycin (TM) and thapsigargin (TG) induces ER stress, whereas sodium arsenite (AS) leads to oxidative stress. The time-points on the bottom indicate length of exposure to the stress-inducing drug.

mRNA, providing a regulatory switch to allow for spatial control of *ACTB* mRNA translation[7]. IGF2BP1 binds to the 3′-UTR of the *ACTB* mRNA and prevents its translation. Phosphorylation of IGF2BP1 at Y396 by the Src kinase, which is localized to the leading edge of the cell or axons, releases IGF2BP1 from *ACTB* mRNAs, allowing their translation at those sites[7]. In contrast, the phosphorylation of IGF2BP1 at S181 was suggested to enhance its binding to the 5′-UTR *of IGF2* mRNA, thereby increasing *IGF2* mRNA translation[31,32]. Importantly, S181 phosphorylation regulates stabilization of IGF2BP1 target RNAs and impacts dendritic branching in hippocampal neurons underlining its functional importance[33,34]. Yet, although IGF2BP1 phosphorylation at distinct sites has been suggested to impact RNA binding, the mechanistic details of IGF2BP1 regulation by phosphorylation remains only partially understood. Moreover, whether IGF2BP1 is regulated by other phosphorylation events has yet to be examined.

Here, we map stress-regulated phosphorylation sites in IGF2BP1 by targeted mass spectrometry analyzes to uncover mechanisms that control IGF2BP1 outputs. Using in vitro reconstitution, biochemistry, and structural methods, we dissect how IGF2BP1 phosphorylation in its disordered linker regions regulates function. We show that phosphorylation of the disordered linkers regulates the propensity of IGF2BP1 to form RNP granules in vitro and in cells by modulating low affinity interaction networks. Our data reveal how disordered regions provide highly tunable regulation of RNP condensate formation through a single phosphorylation event.

## Results

### IGF2BP1 is phosphorylated during proteotoxic stress

IGF2BP1 stabilizes a subset of RNAs during proteotoxic stress and we therefore tested whether IGF2BP1 is regulated through phosphorylation under those conditions[8,35]. To this end, we mapped phosphorylation sites in IGF2BP1 by mass spectrometry (MS) in mammalian cells under control conditions and under conditions where the cells were exposed to proteotoxic stress (Fig. 1B). To compare IGF2BP1 phosphorylation sites under various forms of proteotoxic stress, we exposed the cells to oxidative stress using sodium arsenite (1 or 2 hours) or endoplasmic reticulum (ER) stress using tunicamycin (4 hours) or thapsigargin (1.5 and 4 hours).

To increase specificity and stringency in our analyzes, we enriched for IGF2BP1 by immunoprecipitation from HEK293 cells engineered by CRISPR/Cas9 gene editing to express GFP-tagged IGF2BP1 using split-GFP technology[36]. We obtained a sequence coverage with identified peptides covering 78.7% of the IGF2BP1 amino acid sequence. Peptides covering the disordered linker 2, spanning from amino acids 347 to 423, were not detected in the MS analyzes. This likely resulted from the low complexity nature of this region, which may be inaccessible to tryptic digestion to generate MS-compatible peptides. To overcome this problem, we used different peptidases. The linker 2 region possesses several prolines and treatment with ProAlanase, a peptidase which cleaves after prolines, yielded 87.9% coverage of the whole IGF2BP1 sequence, including the linker 2 in vitro and in cells (Supplementary Data 1).

The subsequent MS analyzes identified several IGF2BP1-derived phosphopeptides whose levels increased or decreased in a stress-dependent manner (Fig. 1C-D, Supplementary Fig. 1B). The identified phosphopeptides mapped to the RRM1 domain (aa S73), disordered linker 1 (aa S181), KH1 (aa T249), KH2 and KH3 domains (aa S436, S438, T446) (Fig. 1C, Supplementary Fig. 1B). The ratio of signal intensities of phosphorylated to unmodified peptides was relatively low (<1% of total protein) for most of the identified phosphopeptides (S73, T249, S436, S438, T446). The most prominent phosphorylation site we identified mapped to S181 (aa 176-QPRQGSPVAAGA-187 and >64% of total peptide signals), the intensity of which increased by around two-fold during oxidative stress induced by arsenite treatment. In contrast, the signal of the phosphorylated peptide at S181 decreased two-fold when cells were treated with ER stress-inducing drugs, indicating that

this phosphorylation event depends on stress type. In line with the published results[7], we detected phosphorylation of the recombinant IGF2BP1 at Y396 by the Src kinase in vitro. However, we did not identify any phosphorylated peptides mapping to the linker 2 under control conditions or under arsenite and thapsigargin-induced stress in cells (>95% of detected peptides were not phosphorylated; please see materials and methods) (Supplementary Fig. 1C, D).

Multiple amino acid sequence alignments revealed that S181 is highly conserved from fish to mammals, suggesting its functional importance (Supplementary Fig. 1A). This site has previously been proposed to be phosphorylated in all three IGF2BP paralogs by mTORC2[31,32]. Notably, apart from mTORC2, kinase motif prediction algorithms derived from experimental data indicated that S181 might be phosphorylated by members of the CMGC kinase family (i.e. SRPK2 and DYRK3), suggesting that other kinases might be involved in this regulation[37]. Altogether, we found that IGF2BP1 is phosphorylated at multiple sites in a stress-dependent manner.

### Phosphomimetic mutants do not impact IGF2BP1 interaction with RNA

Since S181 was the most prominent stress-regulated phosphorylation site identified by our MS approaches, we went on to dissect how it regulates IGF2BP1 function. Interestingly, phosphorylation of S181 in linker 1 is proposed to increase its binding to RNAs, while phosphorylation of Y396 in disordered linker 2 was proposed to decrease it (Fig. 1A)[7,31,32]. However, how the phosphorylation of IGF2BP1 at these two well-described residues regulates its function remained only partially understood. To test whether phosphorylation in the disordered regions impacts IGF2BP1 interaction with RNAs, we measured the binding affinity of model RNAs to wild-type IGF2BP1 and its phosphomimetic mutants (IGF2BP1 S181E and Y396E). For these experiments, we selected two IGF2BP1 target mRNAs based on the published CLIP data sets: i. the unfolded protein response transcription factor *XBP1*, ii. the translation initiation factor *EIF2A*. The CLIP data showed that IGF2BP1 crosslinks to a distinct region in the 3′-UTR of the *XBP1* (Supplementary Fig. 2A, B, Supplementary Table 1) and that *EIF2A* mRNA is enriched in several predicted IGF2BP1-binding motifs[29] (Supplementary Table 1). In addition, as a comparison, we designed model RNAs derived from the functionally well-described IGF2BP1 target RNAs such as *MYC*[38] and *ACTB*[7] (Supplementary Table 1).

IGF2BP binds to its RNA targets by six RNA-binding domains with conserved folds. The individual RNA-binding domains mediate low-affinity interactions with RNA with low specificity, and the combinatorial recognition of RNAs by multiple domains ensures specificity and results in high affinity[21,23]. This interaction involves the recognition of a cluster of distinct and regularly spaced RNA elements covering a - 100 nucleotide-long target RNA region[21]. We in vitro transcribed approximately 200 nt-long regions from the *MYC*, *XBP1* and *EIF2A* RNAs (Supplementary Table 1) and tested their binding to IGF2BP1 using Electrophoretic Mobility Shift Assays (EMSA). The affinities are represented as the apparent dissociation constants ($K_{1/2}$) representing the sum of interactions of IGF2BP1's individual domains with different RNA sequences, causing a shift from unbound to IGF2BP1-bound RNA (please see the materials and methods). IGF2BP1 interacted with RNAs derived from the 3′-UTR of *XBP1* and *EIF2A* with similar affinities (for *XBP1*, $K_{1/2} = 41.0$ nM and for *EIF2A* $K_{1/2} = 48.2$ nM) and bound to the *MYC*-derived RNA with a slightly higher affinity ($K_{1/2}$, wild-type: 14.4 nM) (Fig. 2A, Supplementary Fig. 2C-H, Supplementary Table 2).

Next, we tested whether introducing phosphomimetic mutations to IGF2BP1 at the linkers would impact RNA binding. IGF2BP1 phosphomimetic mutants S181E and Y396E bound to the *XBP1*-derived 201 nt RNA with a similar affinity as the wild-type IGF2BP1 (Fig. 2B, C, Supplementary. Fig. 2C, Supplementary Table 2) ($K_{1/2}$, S181E = 17.1 nM, $K_{1/2}$, Y396E = 22.9 nM). Similarly, the phosphomimetic mutants showed only small differences in their affinity for longer *MYC* and *EIF2A*

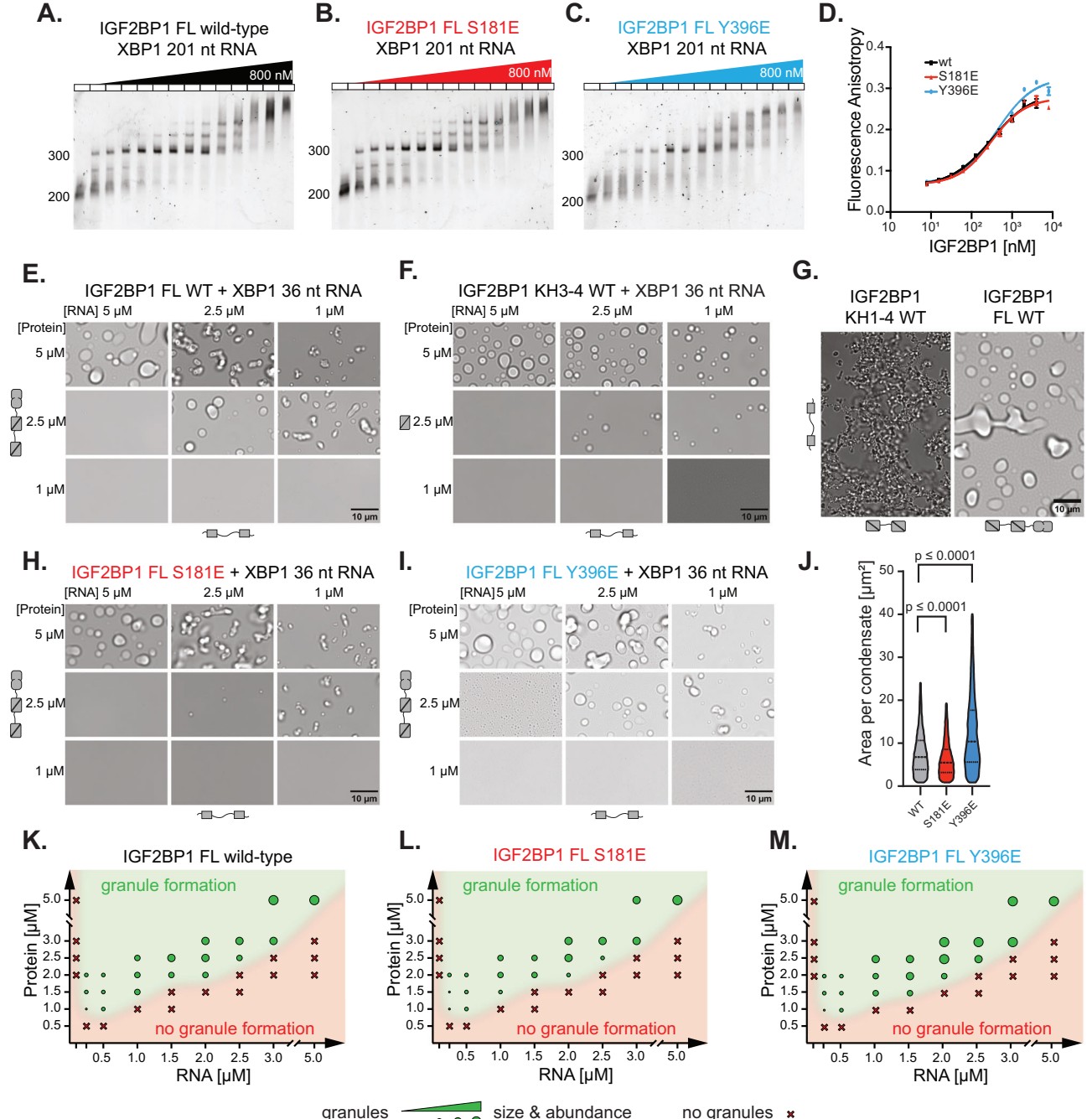

**Fig. 2 | The phosphomimetic mutants alter the size of IGF2BP1 RNP condensates without impacting RNA binding.** Electrophoretic Mobility Shift Assay (EMSA) of XBP1 201 nt RNA with IGF2BP1 wild-type (A), S181E (B) or Y396E (C) at concentrations from 0 to 800 nM. (D) Fluorescence anisotropy experiments of wild-type IGF2BP1 (black), S181E (red) and Y396E (blue) and 5′-fluorescein-labeled XBP1 36 nt RNA. Represented as mean, error bars indicating standard deviation (n = 3 technical replicates). The curves represent the fit of the Hill equation (see Materials and Methods). X-axis in log-scale. (E) RNP granule formation of wild-type IGF2BP1 with XBP1 36 nt RNA at different concentrations after 90 min. The protein (left) and RNA (bottom) valency is depicted by the number of folded domains and the number of predicted binding motifs. (F) RNP granule formation assay of IGF2BP1 KH3-4 pseudodimers under the same conditions as Fig. 2E. Valencies depicted as in Fig. 2E. (G) RNP granule formation assay of 5 μM IGF2BP1 KH1-4 with 5 μM XBP1 36 nt RNA after 90 min incubation in comparison to RNP granules formed by 5 μM full length IGF2BP1 with 5 μM XBP1 36 nt RNA at similar conditions (150 mM NaCl). Scale bar is 10 μm. Valencies are depicted for the protein (bottom) and RNA (left). (H) RNP granule formation of IGF2BP1 S181E and (I) IGF2BP1 Y396E with XBP1 36 nt RNA after 90 min of incubation. Scale bar is 10 μm. Valencies depicted as in Fig. 2E. (J) Violin plot of the area per condensate after 90 min with 5 μM IGF2BP1 full-length wild-type (black), S181E (red) and Y396E (blue) with 5 μM XBP1 36 nt RNA. Two-tailed Mann-Whitney test was used to compare wild-type with S181E (p ≤ 0.0001) and Y396E (p ≤ 0.0001). (K) Phase diagram for RNP granule formation by IGF2BP1 wild-type, (L) IGF2BP1 S181E and (M) IGF2BP1 Y396E with XBP1 36 nt RNA at different protein and RNA concentrations after 90 min of incubation. Granule size and abundance is represented by the circle sizes.

-derived model RNAs (*EIF2A*, $K_{1/2}$, wild-type: 48.2 nM, $K_{1/2}$, S181E: 35.4 nM, $K_{1/2}$, Y396E: 40.2 nM and *MYC*, $K_{1/2}$, wild-type: 14.4 nM, $K_{1/2}$, S181E: 17.8 nM, $K_{1/2}$, Y396E: 16.7 nM) (Supplementary Fig. 2D-J, Supplementary Table 2).

As a complementary quantitative approach, we set up fluorescence anisotropy assays to measure the affinity of IGF2BP1 for shorter RNAs. By truncating the 3′-UTR of the *XBP1* mRNA, we identified a 36 nt-long RNA composed of two predicted IGF2BP1 recognition motifs (Supplementary Fig. 2B, Supplementary Table 1). Fluorescence anisotropy assays showed that wild-type IGF2BP1 bound to the 5′-fluorescein-tagged 36 nt-long *XBP1*-derived RNA with an order of magnitude lower affinity than the 201 nt-long version (Fig. 2D, Supplementary Table 2, K1$_{1/2}$, wild-type: 311.7 nM). We speculate that the lower affinity is due to a reduced number of binding sites in the RNA, resulting in decreased avidity by the RNA-binding domains. IGF2BP1 phosphomimetic mutants S181E and Y396E bound to *XBP1*-derived 36 nt-long RNA at a comparable affinity (Fig. 2D, Supplementary Table 2, K1$_{1/2}$, S181E: 310.1 nM, K$_{1/2}$, Y396E: 423.3 nM) to the wild-type IGF2BP1. Altogether these data indicated that the phosphomimetic mutants do not significantly impact IGF2BP1s interaction with model RNAs. These findings are consistent with previous data showing that canonical folded RNA-binding domains in IGF2BPs drive their interaction with target RNAs[21–23,39–41].

## IGF2BP1 forms RNA-mediated RNP granules in vitro

IGF2BP1 function has been associated with its assembly into RNP granules[42–44]. Therefore, we investigated whether its phosphorylation impacts formation of IGF2BP1 RNP condensates. To test this possibility, we aimed to reconstitute RNP condensates formed by IGF2BP1 and RNAs. Many RNP granules form through phase transitions driven by multivalent interactions between RBPs and RNAs[45–47]. To allow the formation of a multivalent interaction network between IGF2BP1 and RNAs, we used the *XBP1*-derived 36 nt-long RNA, which contains two predicted IGF2BP1-binding motifs (Supplementary Fig. 2B, Supplementary Table 1). We incubated IGF2BP1 with this RNA at different concentrations and stoichiometry and monitored whether they formed RNP condensates visible as droplets by bright-field microscopy (Fig. 2E).

Under physiological pH and salt conditions, 2.5 μM IGF2BP1 and 1 μM *XBP1* 36 nt RNA readily formed RNP condensates (Fig. 2E, Supplementary Fig. 3A, B). Likewise, IGF2BP1 formed condensates at the same protein and RNA concentrations with other model RNAs with similar sequence lengths and number of predicted binding motifs derived from *ACTB*, *MYC*, and *EIF2A* (Supplementary Fig. 3C, Supplementary Table 1). The sequence properties of the RNA had an impact on the morphology of the IGF2BP1-RNP condensates consistent with the published work[47]. The RNAs with a higher propensity to form secondary structures, such as *ACTB* (free energy of −8.50 kcal/mol, Vienna RNAfold WebServer[48]), formed irregular networks. In contrast, *EIF2A* RNA that was not predicted to form secondary structures formed condensates with droplet-like morphology (Supplementary Fig. 3C, D, Supplementary Table 1)[49].

Increasing concentration of the IGF2BP1 and RNA resulted in the formation of larger condensates, whereas excess RNA abrogated condensate formation (Fig. 2E, Supplementary Fig. 3B) as shown for other condensates. Charge repulsion due to excess RNA was proposed to contribute to condensate dissolution by RNAs[50]. It is also plausible that the excess RNA occupies individual RNA-binding domains and breaks the multivalent interaction network. While most RNAs we tested did not lead to condensate formation at stoichiometric concentrations with IGF2BP1, the *XBP1*-derived RNA did. We hypothesize that *XBP1*-derived RNA might have additional non-canonical binding sites that drive phase separation even at stoichiometric concentrations. Altogether, we show that IGF2BP1-RNA interactions can mediate phase separation in vitro.

## IGF2BP1's KH34 domains drive condensate formation

The KH domains drive RNA recognition in IGF2BPs, and the differences in their binding specificity contribute to substrate recognition by IGF2BPs[21,23,51]. As RNA binding drives the condensate formation by IGF2BP1, we next mapped which RNA-binding domains in IGF2BP1 contribute to RNP condensate formation. We first measured the affinity of IGF2BP1's individual domains for model RNAs. We found that KH3-4 dimers bound to a model *ACTB*-derived RNA (ACTB 28 nt)[39] and the *XBP1*-derived 36 nt-long RNA (Supplementary Table 1) at around 1.5 μM affinity (Supplementary Fig. 3E, F, Supplementary Table 2). Instead, KH1-2 bound to the same RNAs with an affinity of >15 μM. Likewise, the RRM1-2 dimer displayed very low apparent binding affinities for both of those RNAs (<100 μM) (Supplementary Fig. 3E, F). These data are consistent with the earlier work indicating that KH3-4 domains in IGF2BPs bind to RNA with the highest affinity[41]. Introducing GEEG mutations, which impede the RNA interaction of the respective KH domain[51,52], into the RNA-binding motif in the KH3 domain in KH3-4 decreased binding affinity to *XBP1*-derived 36 nt-long RNA by 10-fold (Supplementary Fig. 3G, Supplementary Table 2, K1$_{1/2}$ = 16.0 μM). In contrast, the KH3-4 mutant in which the binding site in KH4 is mutated, bound to RNA with a similar affinity as the wild-type KH3-4 dimers (K$_{1/2}$ = 2.1 μM) (Supplementary Fig. 3G). These data suggest that KH3 provides the major RNA binding site since in the KH3-4 construct.

A KH1-4 construct lacking the linker 1 bound to the *ACTB* and *XBP1*-derived short RNAs with similar affinity as the full-length IGF2BP1 (Supplementary Fig. 3E, F, Supplementary Table 2, *ACTB*: K$_{1/2}$ = 75.2 nM, full-length IGF2BP1: K$_{1/2}$ = 68.6 nM; *XBP1*: K$_{1/2}$ = 204.0 nM, full-length IGF2BP1: K$_{1/2}$ = 311.7 nM;). These data suggest that the binding of KH1-2 and KH3-4 domains to RNAs with two binding sites leads to an avidity effect. This can be due to the increased effective concentration for the subsequent binding events after the first KH dimer binds the RNA. Moreover, the incomplete dissociation because of proximal binding sites might increase the apparent affinity[53] (Supplementary Fig. 3H). To characterize this further, we mutated the KH3 and KH4 RNA-binding motifs to GEEG in the full-length IGF2BP1. Both the EMSA assays and fluorescent anisotropy experiments showed that full-length IGF2BP1 KH3-4 GEEG double mutant bound to RNA with similar affinity as the wild-type IGF2BP1 (Supplementary Fig. 3I,J, Supplementary Table 2, K1$_{1/2}$ mutant: *XBP1*-derived 36 nt-long RNA: 185.8 nM; *XBP1*-derived 201 nt-long RNA: 64.7 nM). This indicates that multivalent interactions formed by the simultaneous binding of RRM1-2 and KH1-2 dimers to RNA results in an avidity effect that significantly enhances the affinity compared to the individual dimers. Interestingly, the EMSA assays performed with the *XBP1*-derived 201 nt-long RNA showed that compared to wild-type IGF2BP1, the IGF2BP1 KH3-4 GEEG double mutant displayed differences in the high molecular weight assemblies formed at higher protein concentrations (>250 nM, Supplementary Fig. 3J). From these data, we concluded that the impaired RNA-binding of KH3-4 pseudo-dimers results in a different mode of RNA recognition by the mutant.

In line with the fluorescence anisotropy experiments, which showed the role of the KH3-4 in RNA recognition by the IGF2BP1, the KH3-4 domains alone formed condensates in the presence of RNA (Fig. 2F). Importantly, impairing RNA binding to either KH3 or KH4 domains through GEEG mutations abolished condensate formation (Supplementary Fig. 4A). We propose that this is by impairing their ability to form multivalent interactions required for condensate formation. These data revealed that KH3-4 domains are sufficient to build the multivalency that drives IGF2BP1 RNP condensate formation. Consistent with these results, bright-field microscopy analysis showed that the full-length IGF2BP1 KH3-4 GEEG mutant did not form condensates under the same conditions (Supplementary Fig. 4B). These data indicated that even though this mutant binds to RNA with high affinity (Supplementary Fig. 3I, J, Supplementary Table 2, K$_D$:185.8 nM), the low RNA-binding affinities of the individual KH1-2 and RRM1-2

domains (Supplementary Fig. 3E, F, Supplementary Table 2) do not allow the formation of networks that are necessary for condensate formation under the same conditions. Supporting this, KH1-2 domains alone did not form condensates under conditions where KH3-4 formed droplets (Supplementary Fig. 4C). Notably, a model RNA with a single IGF2BP1-binding motif did not mediate condensate formation when incubated with KH1-4, validating that multiple binding sites in both the RNA and protein are required for condensate formation (Supplementary Fig. 4D). Remarkably, KH1-4 formed mesh-like networks upon incubation with the *XBP1*-derived 36 nt-long RNA under conditions where full-length IGF2BP1 formed droplets (Fig. 2G). These findings indicated that IGF2BP1-RNA granules form via phase separation coupled to percolation[54]. These data also suggested that promiscuous RNA interactions by the RRM1-2 domains increases the dynamics in IGF2BP1-RNA interactions and possibly the valency in the network. In the presence of 250 mM NaCl, KH1-4 formed condensates similar to full-length IGF2BP1 (Supplementary Fig. 4E). We speculate that presence of a high concentration of salt weakens the interaction of KH1-4 with RNA, increasing their binding dynamics. In summary, our data revealed that binding of the KH3-4 pseudodimers to RNA drives IGF2BP1 condensate formation. We speculate that once the condensates are formed, due to the high protein and RNA concentration in the condensed phase, the RRM1-2 and KH1-2 pseudodimers can form additional contacts with RNA.

## Phosphomimetic mutations modulate IGF2BP1 dynamics in RNP granules

After establishing that IGF2BP1 assembled into condensates together with RNA, we next tested whether phosphomimetic mutants S181E and Y396E would display differences in the formation of RNP condensates compared to wild-type IGF2BP1. To this end, we used fluorescence microscopy to quantify the size and area of the condensates formed by wild-type IGF2BP1 and its phosphomimetic mutants at 90 min after induction of condensate formation by the addition of RNA. Quantification of the IGF2BP1-RNA condensates (Supplementary Fig. 3A, see materials and methods) revealed that the average size of condensates and the total area of condensates formed by the S181E mutant were smaller compared to the wild-type IGF2BP1, indicating that the IGF2BP1 S181E mutant is impaired in condensate formation (Fig. 2H, J, Supplementary Fig. 4F-I, Supplementary Table 3, median area per condensate: wild-type: 7.0 μm, S181E: 5.7 μm, mean total area: wild-type: 7753 μm², S181E: 4691 μm², at 5 μM protein and RNA concentration). While IGF2BP1 S181E is largely impaired in condensate formation at 2.5 μM when incubated with stoichiometric amounts of *XBP1*-derived 36 nt-long RNA (Fig. 2H), the presence of 5% mCherry-labeled construct leads to the formation of small condensates (Supplementary Fig. 4F-I). We observed that the mCherry-tag enhances the phase separation propensity of IGF2BP1. This effect was prominent when 2.5 μM IGF2BP1 full-length S181E was incubated with 2.5 μM *XBP1*-derived 36 nt-long RNA (Supplementary Fig. 4G), likely because the saturating concentration of this protein is very close to 2.5 μM. Thus, we used sub-stochiometric amounts of mCherry-labeled IGF2BP1 to quantify the condensate area. Intriguingly, in contrast to the S181E mutant, the Y396E mutant formed larger condensates with a larger total area under the same experimental conditions (Fig. 2I, J, Supplementary Fig. 4 F, H, I, Supplementary Table 3, median area per droplet: Y396E 10.8 μm, mean total area: Y396E 9298 μm², at 5 μM protein and RNA concentration). Similarly, incubation of *XBP1*-derived 36 nt-long RNA with KH1-4 Y396E mutant led to the formation of condensates with regular droplet-like shape compared to the condensates formed by the wild-type KH1-4 under the same conditions (250 mM NaCl). These data confirmed that the Y396E mutation impacts condensate formation, and this effect does not depend on RRM1-2 domains in IGF2BP1 (Supplementary Fig. 4J). Notably, the impact of S181E and Y396E of the phosphomimetic mutants on the morphology of the

IGF2BP1-RNP condensates were identical for four different model RNAs, where the condensates formed by the S181E constructs were smaller and irregular. In contrast, Y396E formed of larger and round droplet-like granules (Supplementary Fig. 5A-I).

We next tested whether phosphomimetic mutants affect the condensation threshold of IG2BP1 RNP condensates. We generated phase diagrams of wild-type IGF2BP1 and its phosphomimetic mutants at different protein and RNA concentrations to address this possibility. Surprisingly, all of the three constructs showed similar threshold concentrations for condensate formation (Fig. 2K-M), with the only difference being IGF2BP1 Y396E, which shows condensate formation at 1.5 μM protein and RNA concentration, while the wild-type did not. The diagram indicated that phase transitions occur at RNA concentrations above 125 nM which coincides with the binding affinity of IGF2BP1 to RNA. At the same time, the saturation concentration of the protein is above 0.5 μM indicating that at low concentrations the percolation required for condensate formation depends on an excess of protein. However, at all the conditions close to the saturation threshold, S181E mutant formed smaller condensates compared to the wild-type IGF2BP1 (Fig. 2K-M). These data indicated that while the phosphomimetic mutants do not impact the saturation concentration, they modulate biophysical properties of the condensates.

Polymers that undergo phase separation coupled to percolation form clusters in sub-saturation concentrations[55]. Thus, we investigated whether the phosphomimetic mutations affect the formation of pre-percolation clusters by using Dynamic Light Scattering (DLS). At low protein and RNA concentrations under the saturation threshold (250-800 nM), we observed particles with around 100 nm hydrodynamic radius for wild-type IGF2BP1 and the phosphomimetic mutants (Supplementary Fig. 5J). Under most conditions, the particles with a hydrodynamic radius ($R_{hyd}$) around 100 nm were most abundant species but at higher protein and RNA concentrations we observed larger clusters ($R_{hyd}$ > 1000 nm) for all three constructs. Remarkably, under the conditions where we used 2:1 protein to RNA ratio, we observed even larger clusters ($R_{hyd}$ > 10000 nm), which is in line with the phase separation assays (Supplementary Fig. 4F, Supplementary Fig. 5J). These data highlighted that the phase separation of IGF2BP1 with RNA is coupled to the formation of clusters at sub-saturation conditions and driven by similar types of interactions. Based on our data, we hypothesize that the effect of the phosphomimetic mutants on the condensate size is not primarily a result of a shift in the saturation concentration but due to changes in the low-affinity interaction network that define the biophysical properties of these RNP granules.

Next, we used turbidity assays to monitor the kinetics of condensate formation following the addition of RNA to the protein. The turbidity assays showed that IGF2BP1 S181E mutant formed condensates with slower kinetics, indicating that phase separation is impaired for this mutant (Supplementary Fig. 5K, Supplementary Table 4, wild-type: $t_{1/2}$ = 167 s, S181E: $t_{1/2}$ = 252 s). Moreover, the condensates formed by the IGF2BP1 S181E mutant showed lower scattering intensity compared to the wild-type protein (Supplementary Fig. 5K, Supplementary Table 4, wild-type: $OD_{480}$ = 0.068, S181E: $OD_{480}$ = 0.035), consistent with the microscopy data that showed the formation of smaller condensates (Fig. 2H, J). In contrast, the Y396E mutant had slightly faster formation kinetics compared to the wild-type IGF2BP1 with similar scattering intensity (Supplementary Fig. 5K, Supplementary Table 4, Y396E: $t_{1/2}$ = 115 s, $OD_{480}$ = 0,064,). We cannot exclude the possibility that IGF2BP1 phosphomimetic mutants slightly impact the threshold concentrations close to the phase boundary and this might affect the kinetics of granule formation in this assay. To sum up, our data indicate that phosphomimetic mutations in the IGF2BP1 disordered linker regions impact the formation of RNP condensates in opposing directions and in a context-dependent manner.

Protein dynamics in condensates impact fusion and growth and affect condensate size[56]. Moreover, the dynamics of the condensate components often correlate with their function[57,58]. We hypothesized that the differences in the sizes and morphology of IGF2BP1 condensates formed by the phosphomimetic mutants might stem from differences in the dynamics of IGF2BP1 molecules in the condensates. Fluorescence recovery after photobleaching (FRAP) experiments measure protein diffusion and mobility in condensates and are widely used to determine the nature of protein interactions in the condensates[59,60]. To test whether IGF2BP1 phosphomimetic mutants had different diffusion dynamics in RNP condensates by FRAP experiments, we formed IGF2BP1-RNA condensates with sub-stochiometric mCherry-tagged IGF2BP1 and monitored the recovery of mCherry fluorescence in the condensates after photobleaching. To decouple condensate growth from fluorescence recovery and allow the formation of large condensates that are tractable for the FRAP measurements, we incubated IGF2BP1 with RNAs for 90 min before performing the photobleaching experiments. The FRAP data revealed that mCherry-IGF2BP1 fluorescence did not recover even 15 min after photobleaching, indicating that IGF2BP1 formed stable complexes with RNAs in the condensates. The long recovery time likely reflects the multivalent nature of IGF2BP1's interaction with RNA, which results in high residence times and long-lived interactions[61]. Consistent with this explanation, FRAP experiments showed that similar to wild-type IGF2BP1, the condensates formed by IGF2BP1 phosphomimetic mutants did not recover fluorescence intensity even after 15 min (Fig. 3A).

We hypothesized that due to competition with other RNAs and RBPs, IGF2BP1 might exhibit more dynamic interactions with RNAs in a cellular environment. Therefore, we reconstituted IGF2BP1-RNA granules under conditions that mimic the nature of RNP interactions in a complex environment. It has been recently shown that supplementing mammalian cell lysates with G3BP1, the RBP that drives stress granule assembly, results in the formation of RNP condensates, which closely resemble stress granules in protein and RNA composition[62]. IGF2BP1 is a component of the stress granules, and we used this method to reconstitute IGF2BP1-containing RNP condensates in cell lysates obtained from HEK293 cells expressing GFP-tagged IGF2BP1. The addition of recombinant G3BP1 to cell lysates induced the formation of condensates that were positive for GFP fluorescence, indicating that recombinant G3BP1 leads to the formation of IGF2BP1-containing RNP condensates in cell lysates (Fig. 3B). We confirmed these results by adding recombinant mCherry-tagged IGF2BP1 after the formation of stress granules. We found that the mCherry-tagged IGF2BP1 was sequestered into G3BP1 induced-RNP condensates (Fig. 3B). We used this experimental setup to assess the dynamics of wild-type mCherry-IGF2BP1 and its phosphomimetic mutants S181E and Y396E in RNPs by FRAP experiments. We found that wild-type IGF2BP1 formed condensates with a mobile fraction of 68.8 % 100 min after the induction of RNP condensate formation with a recovery half-time of 21.6 sec (Fig. 3C, D, Supplementary Table 5). Intriguingly, FRAP experiments performed with the IGF2BP1 S181E mutant showed an almost two-fold increase in recovery half-time (37.5 s), and a slight decrease in the mobile fraction (62.2 %) compared to the wild-type IGF2BP1. In contrast, the IGF2BP1 Y396E mutant showed a faster recovery (15.5 s) and a slightly higher mobile fraction (74.1 %) than the wild-type. These data revealed that while the S181E mutant forms more stable interactions in the RNP condensates, the Y396E forms more dynamic ones. These data showed a correlation between the size of RNP condensates, the mobility and dynamic population of IGF2BP1 wild-type and the phosphomimetic mutants. Therefore, we hypothesize that the S181E mutant forms a more stable interaction with RNAs, and the decreased dynamics leads to the formation of smaller condensates by the S181E mutant. Instead, the Y396E mutant leads to increased dynamics and results in the formation of larger condensates.

## Phosphomimetic mutants impact the size and number of IGF2BP1 RNP granules in cells

To study the impact of the phosphomimetic mutations on IGF2BP1's assembly into RNPs in cells, we established mammalian cell lines stably expressing mCherry-tagged wild-type human IGF2BP1 or its phosphomimetic mutants using a lentiviral transduction approach. We selected two mammalian cell lines for these experiments: U2OS cells (human osteosarcoma) and HCT116 cells (colon carcinoma cells). HCT116 cells do not express IGF2BP1. To exclude the possibility of functional redundancy between IGF2BP paralogs, we knocked out IGF2BP2/3 in HCT116 cells using CRISPR-cas9 gene editing (Supplementary Fig. 6A). As IGF2BP1 expression levels might impact condensate formation, we used fluorescence-activated cell sorting (FACS) to select cells that expressed mCherry-tagged wild-type IGF2BP1 and its phosphomimetic mutants at similar levels. For the HCT116 cells, we used FACS to select single clones to ensure similar expression levels between wild-type and the mutants (Supplementary Fig. 6B, C, Supplementary Data 2). In addition, we studied U2OS cells due to their extended morphology, which is well-suited for microscopy experiments. To monitor stress granules and IGF2BP1 condensates simultaneously, we used engineered U2OS cells expressing GFP-tagged stress granule marker G3BP1[63]. We picked a population of U2OS cells where mCherry-IGF2BP1 and its mutants were expressed similarly to the wild-type protein levels (Supplementary Fig. 6D, E, Supplementary Data 3).

We assessed whether, similar to the in vitro results, IGF2BP1 phosphomimetic mutants impact RNP condensate formation in cells by studying IGF2BP1's assembly into stress granules. Treatment of cells with sodium arsenite resulted in stress granule formation, evidenced by IGF2BP1 sequestration into G3BP1 positive RNP granules (Supplementary Fig. 6F). Quantification of the number, size and total area of mCherry-IGF2BP1 positive granules revealed that HCT116 cells expressing wild-type and phosphomimetic mutants showed similar total granule area per cell (Fig. 3E, F, Supplementary Table 6, median total area per cell, wild-type: 10.3 $\mu m^2$, S181E: 10.3 $\mu m^2$, Y396E: 10.2 $\mu m^2$). Notably, the S181E mutant formed a higher number of condensates per cell (Fig. 3G, Supplementary Table 6, mean condensate number, wild-type: 5.7, S181E: 7.2) with a slightly smaller area per condensate (Fig. 3H, Supplementary Table 6, median condensate area, wild-type: 1.1 $\mu m^2$, S181: 1.0 $\mu m^2$) indicating that similar to what we found in vitro, the S181E forms smaller condensates in cells (Fig. 3H, Supplementary Table 6). These results were similar in both HCT116 and U2OS cell lines, with the differences being more pronounced in the HCT116 cell line. U2OS cells expressing the S181E mutant showed a lower total area of condensates and higher number of condensates (Supplementary Fig. 6G-J, Supplementary Table 6, median total area per cell, wild-type: 34.2 $\mu m^2$, S181E: 27.1 $\mu m^2$, mean number of condensates per cell, wild-type: 18.5, S181E: 20.7). The IGF2BP1 Y396E mutant did not display a large difference in size of the condensates but showed higher number of condensates in HCT116 cells (Fig. 3E-H, Supplementary Table 6, median condensate area, Y396E: 1.1 $\mu m^2$, mean condensate number per cell, Y396E: 6.3). Instead, condensates formed by the Y396E mutant in U2OS cells were smaller compared to those formed by wild-type IGF2BP1 (Supplementary Fig. 6G-J, Supplementary Table 6, median condensate area, wild-type = 0.98 $\mu m^2$, Y396E = 0.80 $\mu m^2$). We speculate that the expression of the wild-type IGF2BP1 and other IGF2BP paralogs, as well as cell-type dependent differences in signaling cascades, could result in the heterogeneity observed in U2OS cells. Overall, we found that the S181E mutation impacts the formation of IGF2BP1-containing granules in cells.

To further characterize the properties of IGF2BP1 in RNP granules in cells, we performed FRAP experiments in U2OS cells expressing mCherry-labeled wild-type IGF2BP1 and its phosphomimetic mutants, S181E and Y396E. Both the S181E and the Y396E mutants displayed reduced levels of dynamic population in stress granules compared to

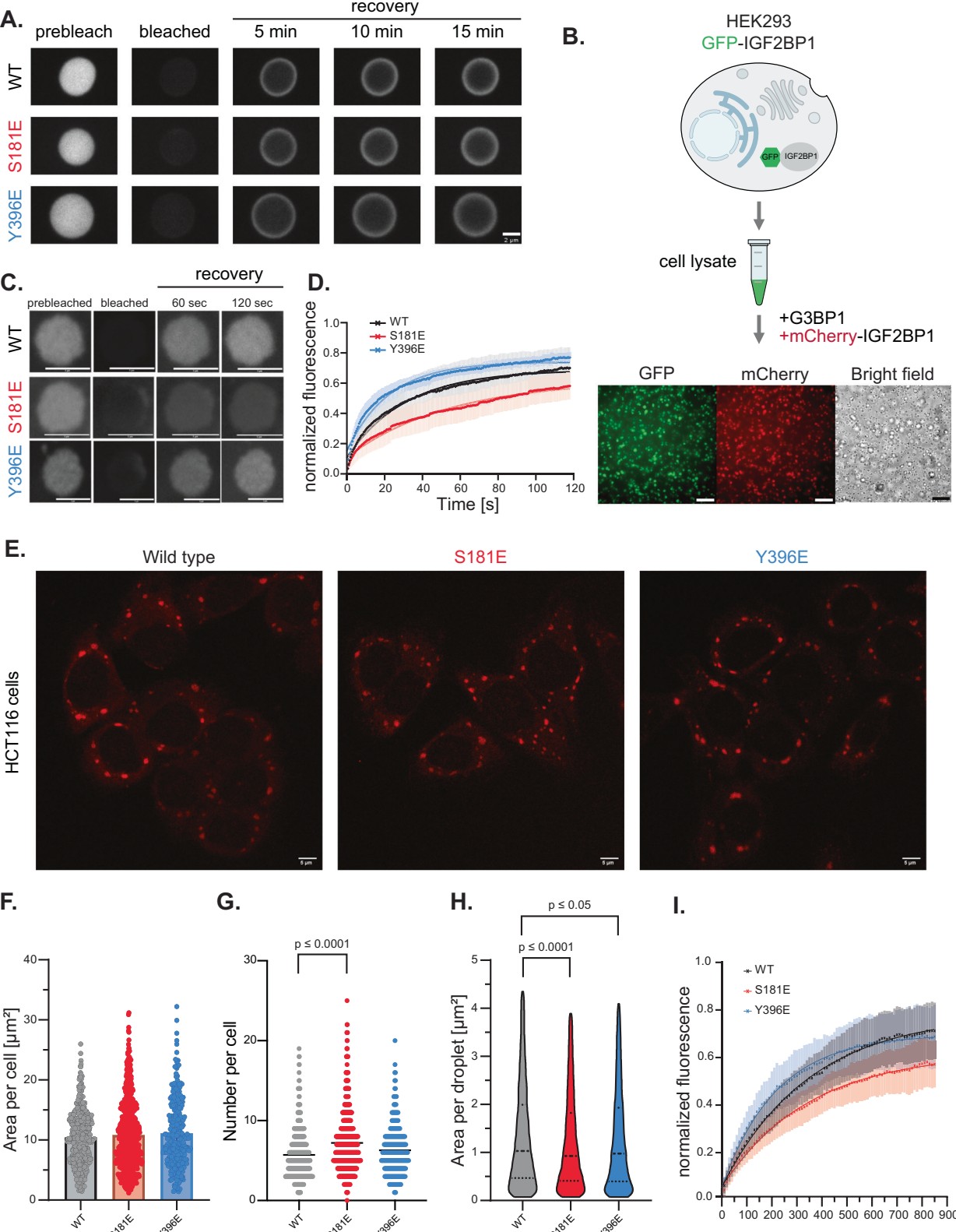

the wild-type IGF2BP1 (% dynamic population wild-type= 77.9, S181E = 61.3 Y396E = 69.4) (Fig. 3I, Supplementary Table 7). This data was inconsistent with the stress granules formed in the lysates, where the Y396E mutant displayed a higher dynamic population than the wild-type IGF2BP1 (Fig. 3D, Supplementary Table 5). In cells, the S181E mutant showed similar recovery half-time as the wild-type IGF2BP1, in contrast the Y396E recovered faster (Fig. 3I, Supplementary Table 7, t1/2

[s] wild-type= 241.7, S181E = 240.0, Y396E = 151.3). We speculate that the composition and the relative stoichiometry of the components in stress granules formed in vitro compared to the ones formed in cells, resulting in the differences observed here. Moreover, it is plausible that stress-induced phosphorylation of the Y396E mutant at different sites might contribute to the differences observed here. Altogether, our data show that the IGF2BP1 phosphomimetic mutants impact

**Fig. 3 | Phosphomimetic mutants affect formation and dynamics of IGF2BP1 RNP condensates in cells.** (A) Representative images of Fluorescence Recovery After Photobleaching (FRAP) of condensates formed by 5 μM full-length IGF2BP1 wild-type, S181E and Y396E and 5% mCherry-labeled IGF2BP1 constructs in the presence of 5 μM XBP1 36 nt RNA after 90 min incubation. Scale bar is 2 μm. (B) Schematic depiction of G3BP1-induced RNP granule formation in lysates and representative images of the incorporation IGF2BP1 into RNP granules. Scale bar is 20 μm. (C) Representative images of FRAP experiments with mCherry-IGF2BP1 wild-type, S181E and Y396E mutants in RNP granules after 100 min incubation. Scale bar is 5 μm. (D) Recovery curves of FRAP experiments with mCherry-IGF2BP1 wild-type (black), S181E (red) and Y396E (blue) mutants in G3BP1-induced granules after 100 min incubation. Data represented as mean with error margins indicating the standard deviation. Curves represent the fit to a one-phase association equation (see Materials and Methods). n = 10 for wild-type, n = 11 for S181E, n = 13 for Y396E. (E) Representative fluorescence images of fixed HCT116 cells expressing mCherry-

IGF2BP1 wild-type, S181E or Y396E. Cells were fixed 60 min after stress induction by 500 μM arsenite. Scale bar is 5 μm. Quantification of condensates in HCT116 cells represented as scatter plots: (F) total area of condensates per single cell (n = 449 for wild-type, n = 741 for S181E, n = 278 for Y396E, bar represents mean value; two-tailed t-test was for statistical analyzes) (G) number of condensates per single cell (bar represents mean value; two-tailed t-test was used to compare wild-type with S181E (p ≤ 0.0001) and Y396E (p ≤ 0.0001)) (H) area per condensate (bars represent the median and 25 % and 75 % quartiles; two-tailed Mann-Whitney test was used to compare wild-type with S181E (p ≤ 0.0001) and Y396E (p ≤ 0.05)). (I) Recovery curves of FRAP experiments of arsenite induced stress granules in U2OS cells expressing mCherry-IGF2BP1 wild-type (black), S181E (red) and Y396E (blue). Represented as mean, error margins indicate the standard deviation. Curves represent the fit to a one-phase association equation. n = 17 for wild-type, n = 16 for S181E, n = 14 for Y396E.

IGF2BP1 dynamics, thereby modulating the size of IGF2BP1 condensates in cells.

## IGF2BP1 adopts a compact conformation in solution

We aimed to elucidate how the phosphorylation of IGF2BP1 at the disordered linkers mechanistically impacts its condensation into RNP granules. We hypothesized that linkers might contact each other or the folded domains contributing to multivalency in condensates and that phosphorylation of the linkers could affect these interactions. Thus, we monitored the conformational status of IGF2BP1 by small-angle X-ray Scattering (SAXS) analyzes. SAXS is a solution scattering method that provides low-resolution structural information on the overall shape of molecules. Analysis of SAXS scattering curves (Supplementary Fig. 7A) showed that IGF2BP1 displays a maximal extension ($D_{max}$) of around 20 nm and a radius of gyration ($R_g$) of 4.18 nm. For flexible molecules such as IGF2BP1, scattering curves represent an average of conformational states sampled by the protein. The Kratky plot of IGF2BP1 highlighted the presence of flexibility in the protein as the curve does not converge to the s-axis (Supplementary Fig. 7B). This is a consequence of a lack of structure in the linker regions. We further analyzed SAXS data using the ensemble optimization method (EOM) to extract information about those states[64]. EOM generates a random pool of structures based on the protein's available structural data and amino acid sequence. EOM then selects the ensemble of structures that best fits the experimental SAXS data. To generate models for EOM for IGF2BP1, we provided high-resolution structures for the individual domains with the flexible linkers. The comparison of the radius of gyration ($R_G$) and maximal distance ($D_{max}$) distributions of the random pool of structures compared to the selected ensembles revealed that the selected ensemble displayed more compact structures compared to the random pool (Fig. 4A, B). These analyzes suggest that IGF2BP1 is in a conformational equilibrium between extended and compact states in solution, with a higher number of molecules found in the compact state at any given time. These data also indicate the presence of low-affinity intramolecular contacts within the molecule, leading to its compaction.

Next, to assess whether phosphomimetic linker mutants impact the overall conformation of IGF2BP1 in solution, we used SAXS analyzes. The SAXS data revealed that the scattering curves of the wild-type IGF2BP1 were very similar to the phosphomimetic mutants (Supplemenetary Fig. 7A). In line with this data, EOM analyzes of the phosphomimetic mutants showed comparable distributions of $D_{max}$ and $R_g$ values (Supplementary Fig. 7C, D). The similarity of $D_{max}$ and $R_g$ values indicated that wild-type IGF2BP1 and phosphomimetic mutants share a comparable overall conformational ensemble. These data revealed that the negative charges introduced to the disordered linkers do not result in significant conformational changes in the protein. As SAXS data provides low-resolution information on a conformational ensemble, we next used orthogonal methods to

study the impact of phosphorylation on the conformation and self-assembly of the linkers.

## IGF2BP1 linkers do not form condensates in isolation

Disordered regions contribute to the phase separation of proteins. Therefore, the phosphorylation of the linkers could regulate condensate formation through modulating self-association between these regions. To monitor the condensation propensity of the linkers in isolation, we performed coarse-grained Molecular Dynamic (MD) simulations of the disordered linkers and their corresponding phosphomimetic mutants (Fig. 4C-D). We ran 10 μs long Martini 3[65] simulations of 33 copies of the polypeptide chains of each linker in explicit solvent. These MD simulations revealed that over a 10 μs time scale, neither the wild-type linker, nor the phosphomimetic mutants assembled into clusters. A more detailed analysis of the intermolecular contact interactions revealed that linker 2 has a slightly higher tendency to engage in low-affinity, short-lived interactions. The wild-type linker 1 and its phosphomimetic mutant show the same lack of intermolecular interactions (Supplementary Fig. 7E). Interestingly, the phosphomimetic mutation in linker 2 significantly promotes the formation of intermolecular interactions (Supplementary Fig. 7F). While introduction of the phosphomimetic mutant Y396E to linker 2 does not lead to condensate formation, it might provide an additional source of low-affinity interactions to drive IGF2BP1 condensation with a liquid-like behavior.

Our in vitro microscopy experiments showed that the linkers did not form condensates at high protein concentrations (150 μM) even in the presence of a crowding agent supporting the MD simulations (Supplementary Fig. 7G). To dissect this further experimentally, we performed Nuclear Magnetic Resonance spectroscopy (NMR) experiments. For NMR analyzes, we produced $^{15}$N-isotope labeled proteins. $^1$H-$^{15}$N Heteronuclear Single Quantum Coherence (HSQC) experiments revealed a small dispersion of backbone amide signals, thus validating the disordered nature of the linker segments (Supplementary Fig. 8A-D). We assessed the self-association of the isolated disordered linkers by monitoring their NMR signal intensity at different linker concentrations. For monomeric non-self-associating molecules, NMR signal intensity increases linearly with increases in protein concentration. In contrast, if the linkers stably associate with each other, this would broaden the NMR signals and a drop in signal intensity. We acquired HSQC spectra of the linker 1 at protein concentrations ranging from 25-200 μM (Supplementary Fig. 8E). Our data revealed that the signal intensity of the backbone amide groups increased linearly with increased protein concentration, and there was no deviation from the predicted intensities. These data indicated that linker 1 does not stably self-associate under the conditions we tested. We made similar observations with linker 2 under the same experimental conditions, suggesting that both linkers do not form stable clusters or condensates under those experimental conditions (Supplementary

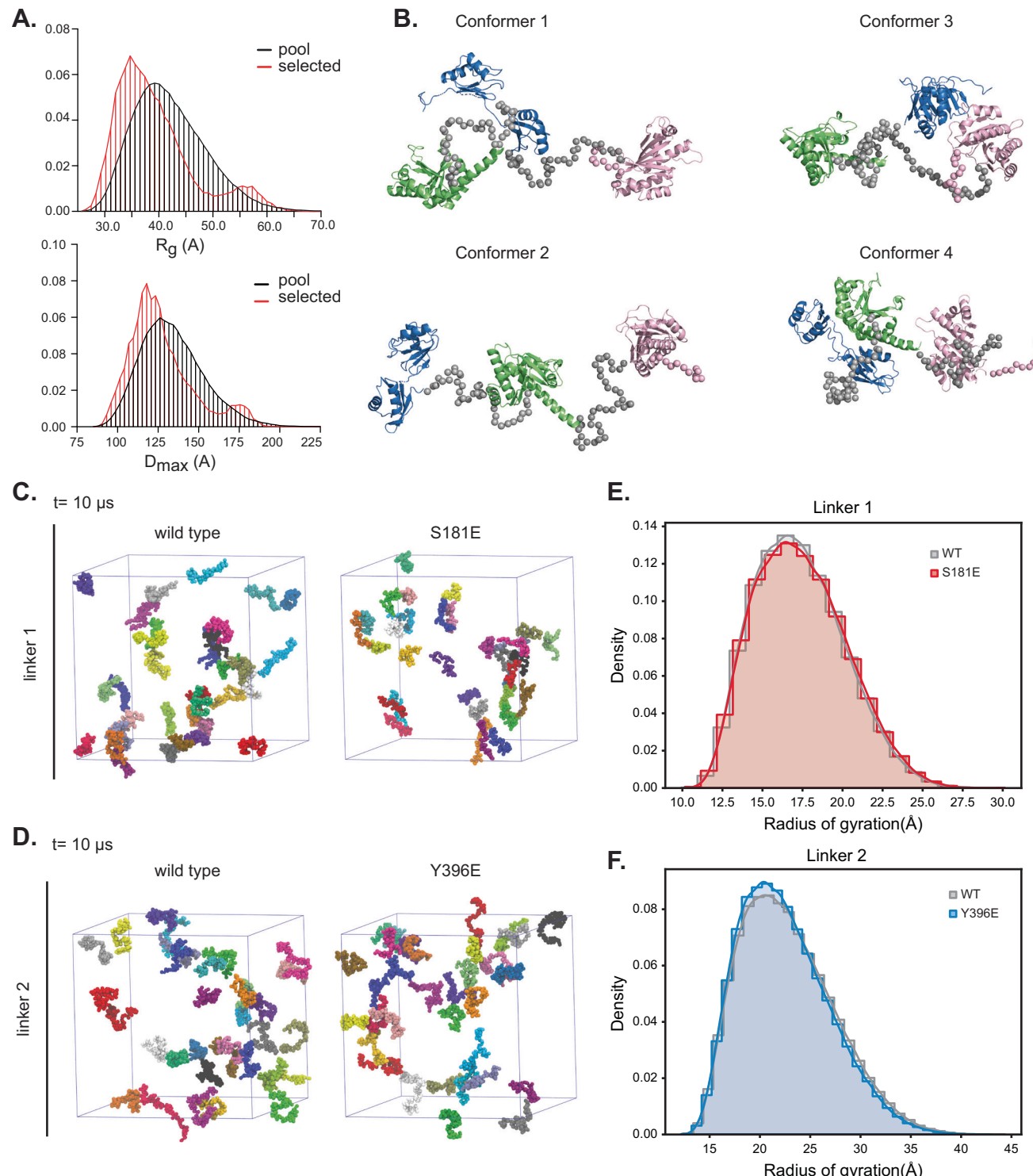

**Fig. 4 | IGF2BP1 adopts a compact conformation in solution.** (A) Comparison of radius of gyration ($R_g$, top) and maximum extension ($D_{max}$, bottom) distribution of random conformations of IGF2BP1 (black) and selected pool (red) that best fit the experimental SAXS data based on EOM analyzes. (B) IGF2BP1 structural conformers that best fit the experimental SAXS data (linkers are shown as gray spheres). The RRM1-2, KH1-2 and KH3-4 domains are colored in blue, green and pink, respectively. (C) Snapshot of coarse-grained Martini 3 Molecular Dynamics simulations of linker 1 and its mutant S181E at 10 μs. Each simulation box contains 33 copies of the polypeptide, depicted in different colors and sphere representation. Water and ions are not shown for clarity. (D) Snapshot of coarse-grained Martini 3 Molecular Dynamics simulations of linker 2 and its phosphomimetic mutant Y396E at 10 μs. Each simulation box contains 33 copies of the polypeptide, depicted in different colors and sphere representation. Water and ions are not shown for clarity. (E) $R_g$ probability density distribution of linker 1 and its phosphomimetic mutant during MD simulations. (F) $R_g$ probability density distribution of linker 2 and its phosphomimetic mutant during MD simulations.

Fig. 8F). To sum up, together with the MD simulations, the NMR data suggest that while the linkers do not drive phase separation on their own, linker 2 phosphomimetic mutant Y396E might form low-affinity contacts with other copies in trans. We anticipate that this might contribute to the changes in the dynamics of the condensates formed by Y396E mutant compared to the wild-type IGF2BP1.

## IGF2BP1 linkers adopt an extended conformation

The amino acid properties of interdomain linkers determine their effective solvation volume and impact the conformational space of linear multi-domain proteins[66]. Importantly, linker compaction could regulate phase separation properties in multi-domain proteins[66]. As the linkers we tested did not have a propensity to self-associate, we next tested whether they form short distance intramolecular contacts that underlie the compaction we measured by SAXS analyzes. These contacts might impact the phase separation propensity of IGF2BP1. In MD simulations, we calculated the frequency of contacts formed within the same polymer chain (intra-molecular, cis-interactions). The contact maps obtained (Supplementary Fig. 9A, B) indicated that cis-contacts are rarely established among residues separated by more than four amino acids within the disordered linker 1 and linker 2, and do not occur in the phosphomimetic mutants. These results highlight how cis-interactions are sparse in the wild-type as well as in the phosphomimetic mutants. We observed no long-range interaction within the chains. This was reflected in the lack of trans-interactions in our simulation box. In line with these observations, the distributions of the $R_g$ of the linkers (linker 1 and linker 2) and the phosphomimetic mutants in MD simulations revealed that both linker 1 and linker 2 and their phosphomimetic mutants adopt expanded conformations with similar median $R_g$ values (Fig. 4E-F, Supplementary Table 8). Notably, both linker 1 and linker 2 display higher $R_g$ values compared to an ideal polymer (i.e., an Analytical Flory Random Coil)[67] with perfectly balanced polymer-solvent and polymer-polymer interactions and with the same amino acid sequence for each linker. These higher $R_g$ distribution values correspond to a polymer model where polymer-solvent interactions are more prominent than polymer-polymer interactions (Supplementary Fig. 9C,D). The MD simulations suggested that the linkers form an extended solvent-exposed conformation. Based on these data and the NMR experiments, we excluded the possibility that the differences in linker compaction led to the different phase separation propensities observed for the IGF2BP1 phosphomimetic mutants.

## The linkers form low-affinity interactions with the folded domains and RNA

Besides compaction and self-association of the linkers, transient interactions between the linkers and folded domains could lead to the IGF2BP1 compaction observed in SAXS experiments. To test whether the linkers can interact with the folded domains of the protein, we performed $^1$H-$^{15}$N HSQC experiments for the titration series of the $^{15}$N-labeled linkers with the NMR invisible domains (RRM1-2, KH1-2, and KH3-4). Interactions between the linkers and domains are expected to induce chemical shift perturbations (CSPs) and/or peak broadening, reducing signal intensity for amino acids close to the interaction surface. We used a three-dimensional sequential assignment strategy to assign the signals in the NMR spectra of the linkers. We could unambiguously assign 70–80% of the signals in both linkers.

The titration of linker 1 with the folded domains showed very low CSPs (< 0.015 ppm) that were consistently alike for both the wild-type and S181E mutant protein (Fig. 5A, Supplementary Fig. 10A-I). Therefore, we concluded that linker 1 does not interact with the folded domains of the protein. Besides forming protein-protein interactions, the disordered linkers could form contacts with RNA, which could modulate the phase separation propensity of IGF2BP1.

The RGG motifs in RNA-binding proteins mediate RNA-protein interactions[68]. Linker 1 contains an RGG-RG motif which is in the vicinity of S181 that shows evolutionary conservation in mammals and some vertebrate species (Supplementary Fig. 1A). Therefore, we investigated whether linker 1 could contribute to protein-RNA interactions by measuring HSQC spectra of linker 1 in the presence of a model RNA (12xUG) that is recognized by RGG containing proteins through formation of an RNA quadruplex[69] and a 10 nt RNA derived from *XBP1*. Titration of the wild-type linker 1 with the 12xUG RNA resulted in large chemical shift perturbations and a decrease in signal intensity of the residues around the RGG-RG motif (G170, G172 and G175, Fig. 5B-D and Supplementary Fig. 10J, K). Notably, the linker 1 phosphomimetic mutant S181E was impaired in binding to RNA, evident in lower CSPs and a lower drop in the signal intensity around the phosphorylation site upon its titration with the 12xUG RNA. We observed similar, albeit weaker, CSPs of the identical residues upon titration of the linker 1 with the 10nt *XBP1* RNA, indicating that this RNA bound to linker 1 with a much lower affinity (Supplementary Fig. 10L-P). The difference between the wild-type and the S181E mutant was more pronounced due to low-affinity interactions (Supplementary Fig. 10L-P). Our data suggest that impairing non-specific RNA binding by linker 1 via phosphorylation tunes the formation of IGF2BP1-containing RNP condensates.

Next, we assessed whether linker 2 interacts with the folded domains of IGF2BP1 through NMR experiments. The NMR analyzes revealed that titration of wild-type linker 2 with RRM1-2 as well as the KH1-2 domains showed small CSPs around aa S388-P395 (aa 388-SSVTGAP-395) (CSP > 0.015) (Supplementary Fig. 11A-F) indicating that linker 2 interacts with these domains with a low affinity in the millimolar range. Importantly, titration of the linker 2 phosphomimetic Y396E mutant with both domains displayed reduced CSPs in this region. These data demonstrate that the Y396E mutation modulates the low-affinity binding of linker 2 to RRM1-2 and KH1-2 domains.

Titration of linker 2 with KH3-4 domains displayed the largest CSPs observed for all folded domains, revealing that linker 2 most strongly interacts with KH3-4 dimers (Fig. 5E-G, Supplementary Fig. 11G-H). Nevertheless, the fluorescence anisotropy experiments showed that a construct containing linker 2 and the KH3-4 domains (linker 2-KH3-4) bound to RNA with a similar affinity to the KH3-4 domains alone. This result indicates that the low-affinity interaction between linker 2 and the KH3-4 domains does not impair RNA binding (Supplementary Fig. 11I, Supplementary Table 2). The KH3-4 pseudo-dimers bound to a largely hydrophobic region in linker 2 covering Q361-F376 (aa 361-QSHLIPGLNLAAVGLF-376). Notably, of all IGF2BP1 domains, this segment only bound to KH3-4, emphasizing the specificity of this interaction. In addition, KH3-4 domains bound to a region covering the Y396 phosphorylation site (aa V390-M400). The linker 2 Y396E phosphomimetic mutant showed reduced CSPs upon KH3-4 binding close to the mutation site, yet the interaction of KH3-4 with the hydrophobic segment (aa Q361-F376) was not affected by the Y396E mutation. In summary, linker 2 forms low-affinity contacts with KH3-4 pseudodimers, which are weakened in the Y396E phosphomimetic mutant. Notably, MD simulations on linker 2 showed that the Y396E mutant had an increased propensity to self-associate compared to the wild-type protein. It is plausible that the reformation of the low-affinity interaction network by the linker 2 impacts condensates formed by the phosphomimetic mutant Y396E.

The NMR data revealed that while linker 1 binds to RNA, linker 2 forms low-affinity interactions with the folded domains in IGF2BP1, and both interactions are modulated in the relevant phosphomimetic mutants. We anticipate that these low-affinity interactions are more pronounced in the condensed phase with high protein and RNA concentration (up to 20 mM)[70], resulting in the prominent effect of these mutants in IGF2BP1 RNP condensates.

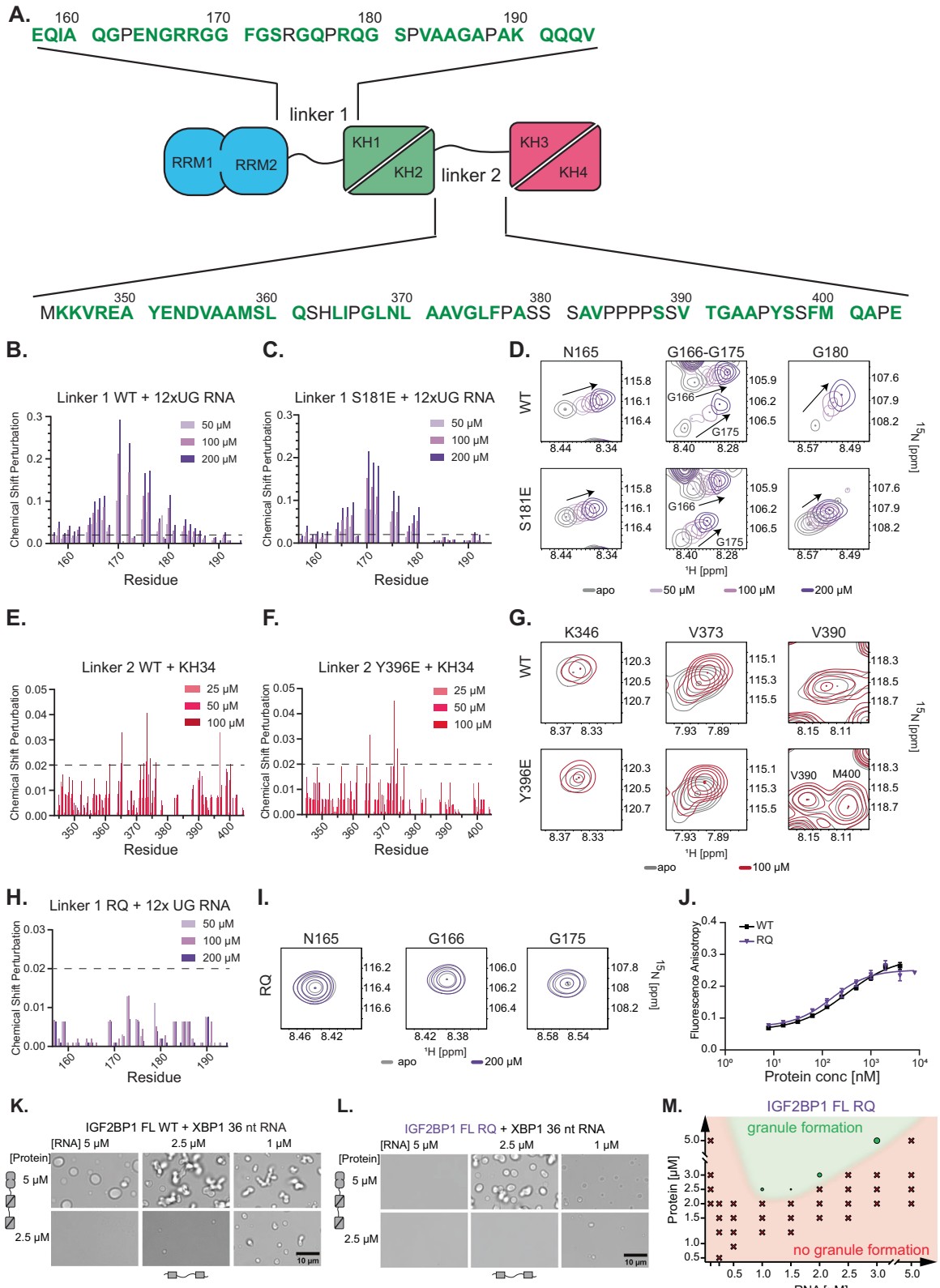

## Impairing low-affinity RNA interactions in linker 1 impacts IGF2BP1 phase separation

Based on our NMR data, which showed that the RGG-motif-mediated RNA interactions are weakened in the S181E mutant, we speculated that the loss of low-affinity RNA-protein interactions in S181E impacts the formation of RNP condensates. This model would predict that impairing low-affinity RNA interactions mediated by the RGG motif in the linker 1 would phenocopy the S181E phosphomimetic mutant in its capacity to form RNP condensates. To test this model, we impaired the interaction of the RGG motif with RNAs by mutating four arginine residues at the positions 167, 168, 174, and 178 to glutamine, and for simplicity, we called this mutant IGF2BP1 RQ mutant (Supplementary Table 9). To investigate the effect of the RQ mutations on the RNA binding by the linker 1, we performed HSQC titration experiments with a model RNA

**Fig. 5 | Disordered linkers form low-affinity contacts to regulate IGF2BP1 phase separation.** (A) Amino acid sequence of wild-type linker 1 and linker 2. (B) Chemical Shift perturbation (CSP) analyzes of ¹⁵N-labeled wild-type linker 1 in the absence (black) and presence of various concentrations of 12xUG RNA in ¹⁵N-¹H HSQC experiments. (C) CSP analyzes of ¹⁵N-labeled linker 1 S181E mutant in the absence and presence of various concentrations of 12xUG RNA. (D) Select backbone amide signals in the HSQC spectrum of wild-type linker 1 and the S181E mutant in the absence and the presence of various concentrations of 12xUG RNA. (E) CSP analyzes of ¹⁵N-labeled wild-type linker 2 in the absence and presence of different concentrations of KH3-4. (F) CSP analyzes of ¹⁵N-labeled linker 2 Y396E mutant in the absence and presence of various concentrations of KH3-4. (G) Representative signals of ¹⁵N-labeled wild-type linker 2 and Y396E in the absence and presence of 100 μM KH3-4. (H) CSP analyzes of ¹⁵N-labeled linker 1 RQ mutant in the absence and presence of various concentrations of 12xUG RNA. (I) Representative backbone

amide signals from the linker 1 RQ mutant in absence and the presence 200 μM of 12xUG RNA from HSQC spectra. (J) Fluorescence anisotropy experiments measuring binding of wild-type IGF2BP1 (black) and IGF2BP1 RQ (purple) to 5′-fluorescein labeled XBP1 36 nt RNA. Data is represented as mean with error bars indicating the standard deviation (n = 3). The curves represent the fit of the Hill equation (see Materials and Methods). X-axis is represented in log-scale. (K) Comparison of the RNP granule formation of wild-type IGF2BP1 and (L) IGF2BP1 RQ in the presence of XBP1 36 nt RNA after 90 min of incubation. The protein and RNA concentrations are indicated in the figures. The valency of the protein (left) is depicted by the number of folded domains. The valency of the RNA (bottom) is depicted as the number and position of binding motifs. (M) Phase diagram of RNP granule formation of IGF2BP1 RQ with XBP1 36 nt RNA at different protein and RNA concentrations after 90 min of incubation.

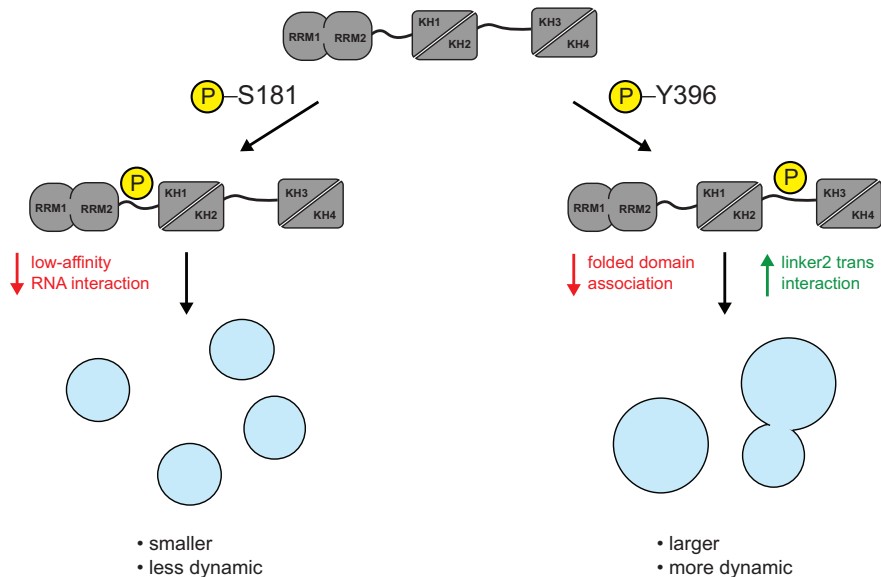

**Fig. 6 | Phosphorylation of IGF2BP1 in disordered linkers impacts RNP condensates.** Model depicting how phosphorylation affects IGF2BP1 RNP condensates.

(12xUG). We showed that the RQ mutant linker 1 was abolished entirely in binding to this model RNA (Fig. 5H, I, Supplementary Fig. 11J). The EMSA data revealed that the binding of IGF2BP1 RQ mutant to the *XBP1*-derived 201 nt-long RNA was not affected by the abrogation of low-affinity protein-RNA interactions in the linker 1 (Supplementary Fig. 11K, L, Supplementary Table 2). Accordingly, fluorescence anisotropy experiments did not indicate any observable difference in the RQ mutant's affinity to the *XBP1*-derived 36 nt RNA (Fig. 5J).

Remarkably, compared to the IGF2BP1 wild-type, the RQ mutant was largely impaired in RNP condensate formation in the presence of the *XBP1*-derived RNA (Fig. 5K-M). This effect was even more notable when we investigated the granule formation with the *MYC*-derived RNA (Supplementary Fig. 11M, N) where we could not observe condensates at any concentration used in these experiments. In summary, these data show that abrogating low-affinity protein-RNA interactions of the RGG/RG motif in linker 1 has no observable effect on the apparent binding affinity but impacts the RNP granule formation. Notably, the RQ mutant phenocopied the behavior of the S181E mutant, supporting the notion that low-affinity RNA interactions through the linker 1 regulate phase separation of IGF2BP1.

Based on our data, we hypothesize that the proteotoxic stress-dependent phosphorylation of S181 increases the rigidity and decreases the size of IGF2BP1-containing granules by abrogating low-affinity protein-RNA interactions. In contrast, Y396E phosphorylation increases the dynamics and size of these condensates by modulating low-affinity protein interaction networks (Fig. 6).

## Disordered linkers contribute to RNA regulation by IGF2BP1

While phosphorylation of the disordered linkers was shown to contribute to the posttranscriptional RNA regulation by IGF2BP1, our knowledge has been limited to select targets. To have a global view of the regulation of RNA metabolism by IGF2BP1 phosphorylation in cells, we performed transcriptomics analyzes in HCT116 cells that solely express wild-type mCherry-IGF2BP1 or its phosphomimetic mutants. To identify whether the linker mutations impact IGF2BP1's binding to RNAs in an unbiased manner, we employed RNA immunoprecipitation sequencing (RIP-seq) experiments under control conditions and upon exposure of the cells to arsenite stress. We immunoprecipitated mCherry-IGF2BP1 and its mutants (S181E and Y396E) using RFP-TRAP magnetic beads and isolated the associated RNAs for deep sequencing. We used nonengineered HCT116 cells as a control to account for the unspecific interaction of the cellular RNA with the beads. To determine the possible function of low-affinity RNA interactions by the linker 1, we engineered HCT116 cells to express mCherry-IG2BP1 RQ mutant using lentiviral transduction and characterized its binding to RNAs. To circumvent differences in the transcriptome that might arise from clonal selection, for those experiments, we selected a population using a very narrow gate for mCherry signal in the FACS experiments to ensure comparable expression levels of the wild-type mCherry-IGF2BP1 and its mutants (Supplementary 12A, Supplementary Data 4).

The transcriptomics data showed that the levels of several RNAs significantly (adjusted *p*-value < 0.05) increased or decreased more than 20% in HCT116 cells expressing IGF2BP1 mutants compared to the

wild-type in control conditions and upon induction of stress via arsenite treatment. These data validated the regulatory role of the disordered linkers for IGF2BP1 function to control RNA metabolism in cells. Principle component analyzes clustered mCherry-IGF2BP1 wild-type with the RQ mutant, whereas mCherry-IGF2BP1 S181E mutant clustered together with the Y396E mutant, underlining the regulatory role of the distinct sites S181 and Y396 in the linkers (Supplementary Fig. 12B, Supplementary Data 5).

The phosphomimetic mutants did not impact the RNA-binding affinity of IGF2BP1 for several model RNA substrates in our in vitro assays. While these assays are robust in providing quantitative information on these select substrates, we sought to obtain a genome-wide global overview of whether the phosphomimetic mutations impact the RNA-binding capacity of IGF2BP1 in cells. We compared the RIP-seq data from wild-type IGF2BP1 and its mutants toward this goal. To define the high confidence IGF2BP1-binding transcripts, we set an arbitrary cut-off of 4-fold enrichment of the IGF2BP1-bound RNAs compared to the total RNA levels. Moreover, we set a threshold of 4-fold enrichment of RNAs in the RIP-seq data compared to the background to eliminate unspecific RNA interactions. The RIP-seq analyzes showed that wild-type mCherry-IGF2BP1 and its mutants bound approximately 1400 RNAs under control conditions, aligning with the published data (Supplementary Fig. 12C)[8,71]. Gene Ontology term analyzes of IGF2BP1-bound RNAs confirmed IGF2BP1's regulatory role in development and metabolism[28,72] (Supplementary Fig. 12D). While 462 of these RNAs did not show stress-dependent differences in their binding to IGF2BP1, approximately 1000 RNAs only bound to IGF2BP1 during control conditions, and about 250 RNAs only during arsenite treatment (Supplementary Fig. 12E). These analyzes revealed that the arsenite stress impacts IGF2BP1's interaction with select RNA targets. Notably, the IGF2BP1 mutants bound to a largely overlapping set of RNAs as the wild-type protein under control conditions (1200 shared target RNAs), while only 200 RNAs interacted with either wild-type IGF2BP1 or the linker mutants (Supplementary Fig. 12E).

The differences in the RNA expression levels might impact IGF2BP1 binding to RNA. To circumvent this problem, we defined an RNA-binding score by normalizing the RIP-seq levels to the RNA-seq levels. The fold-change in the RNA-binding score for mCherry-IGF2BP1 versus its mutants revealed a set of RNAs differentially binding to wild-type or mutant IGF2BP1 (Fig. 7A, Supplementary Data 5). In general, the phosphomimetic mutants bound to fewer RNAs during arsenite stress compared to the wild-type IGF2BP1 (Fig. 7A, Supplementary Fig. 12E). The IGF2BP1 RQ mutant showed less regulatory potential, where the mutation only impacted IGF2BP1's interaction with a small number of RNAs (Fig. 7A, Supplementary Fig. 12E). Our analyzes revealed that while wild-type IGF2BP1 and its mutants bind to similar RNAs, the differences are instead in the "strength" of binding. Importantly, in most cases, the mutants showed decreased binding to the RNA targets compared to the wild-type IGF2BP1 (Fig. 7A, B, Supplementary Fig. 12E).

Intriguingly, the differentially regulated targets by the phosphomimetic mutants included several RNAs with a regulatory potential (Fig. 7B). For example, the Y396E phosphomimetic mutant showed largely impaired RNA-binding to the canonical IGF2BP1 target RNA HMGA2, where the other mutants showed similar binding efficiency as the wild-type protein. In contrast, all the mutants showed impaired interaction with the PABPC1 and PHC3 mRNAs compared to the wild-type IGF2BP1 (Fig. 7B). Interestingly, unlike most RNAs that showed decreased binding to IGF2BP1 during arsenite stress, select RNAs, including GARS1, showed preferential IGF2BP1 binding under stress conditions (Fig. 7B). We also observed increased binding capacity of the linker mutants to a small group of RNA targets (see CCDC127) compared to the wild-type IGF2BP1. To sum up, our data show distinct differences in RNA-binding properties of the IGF2BP1 phosphomimetic mutants in cells.

While for select RNAs, the changes in the IGF2BP1 binding efficiency correlated with a decrease in the RNA levels, for others, increased IGF2BP1 binding led to target destabilization (see LIPA, Fig. 7B), suggesting that IGF2BP1-binding causes opposing outcomes for its targets. Importantly, the changes in IGF2BP1-binding did not lead to changes in the RNA levels for several RNAs. IGF2BP1 regulates the stability, translation, and localization of its targets[8,25,35,73]. One possible explanation for these results might be that IGF2BP1 regulates the translation or localization of those targets. These data also suggest that the significant differences in the transcriptome in cells expressing IGF2BP1 mutants compared to the wild-type IGF2BP1 are the consequence of the remodeling of the transcriptome by a subset of IGF2BP1 RNA targets that encode for proteins involved in the regulation of transcription or RNA metabolism, such as HMGA2 and PABPC1. In summary, our data validate the regulatory potential of the disordered linkers in controlling RNA metabolism in cells.

## Discussion

Posttranslational modifications (PTMs) regulate protein function in a reversible, tunable manner that allows exquisite spatiotemporal control[74]. PTMs regulate many RBPs by modulating target RNA binding, interaction with partners, or subcellular localization, thereby contributing to the control of RNA metabolism in cells[75–77]. It has become increasingly clear that PTMs regulate assembly of RBPs into biomolecular condensates by phase separation. Using mass spectrometry, we mapped steady-state and stress-induced phosphorylation sites in IGF2BP1.

Targeted proteomics identified stress-dependent phosphorylation sites in IGF2BP1. Apart from the S181 site, whose phosphorylation decreased approximately two-fold during ER stress, ER stress only mildly impacted the phosphorylation status of IGF2BP1. Notably, the oxidative stress increased phosphorylation at various sites throughout the protein, including a two-fold increase in S181 phosphorylation, revealing the context-dependent nature of this phosphorylation event. IGF2BP1 residue S181 was previously proposed to be phosphorylated by mTORC2[31,32]. Importantly, recent data also predicted that the S181 site is phosphorylated by the CMKG kinase family[37]. Several members of the CMGC kinase family are activated during oxidative stress, and the family member DYRK3 partitions to stress granules and regulates their disassembly[78]. Under the induced ER stress conditions used here, cells did not form stress granules, suggesting their formation might be necessary for stress-induced IGF2BP1 phosphorylation at this site. Apart from prominent S181 phosphorylation (64% of the total protein pool), most of the identified phosphorylation sites in IGF2BP1 were only modified at sub-stoichiometric levels (< 1%). Those sites might be regulated in a spatial or cell-type-specific manner and could be present at a higher frequency in other cell types.

In the absence of RNA, purified IGF2BP1 displayed monodisperse, monomeric behavior. In contrast, in the presence of RNAs with multiple IGF2BP1-binding motifs, IGF2BP1 assembled into RNP condensates. By systematically analyzing IGF2BP1 truncation mutants, we revealed that the KH3-4 domains in IGF2BP1 drive the formation of RNP condensates. These data align with experiments in which only RNA-binding mutants of KH3-4 domains impaired IGF2BP assembly into stress granules in cells[43]. Minimal IGF2BP1 RNP condensates consisting of RNA and IGF2BP1 were highly rigid due to the multivalent interaction of the RNA with the multidomain IGF2BP1. Like other biological systems involving multidomain RBPs that mediate specific multivalent engagement with RNAs, we anticipate that IGF2BP1 RNP condensates form via phase separation coupled to percolation[54,66]. Notably, while full-length IGF2BP1 formed droplets, KH1-4 domains formed meshed networks. These data showed that binding KH1-4 domains to RNAs mainly results in percolation without phase separation. These data revealed that RRM domains contribute to the coupling of percolation with phase separation, ultimately

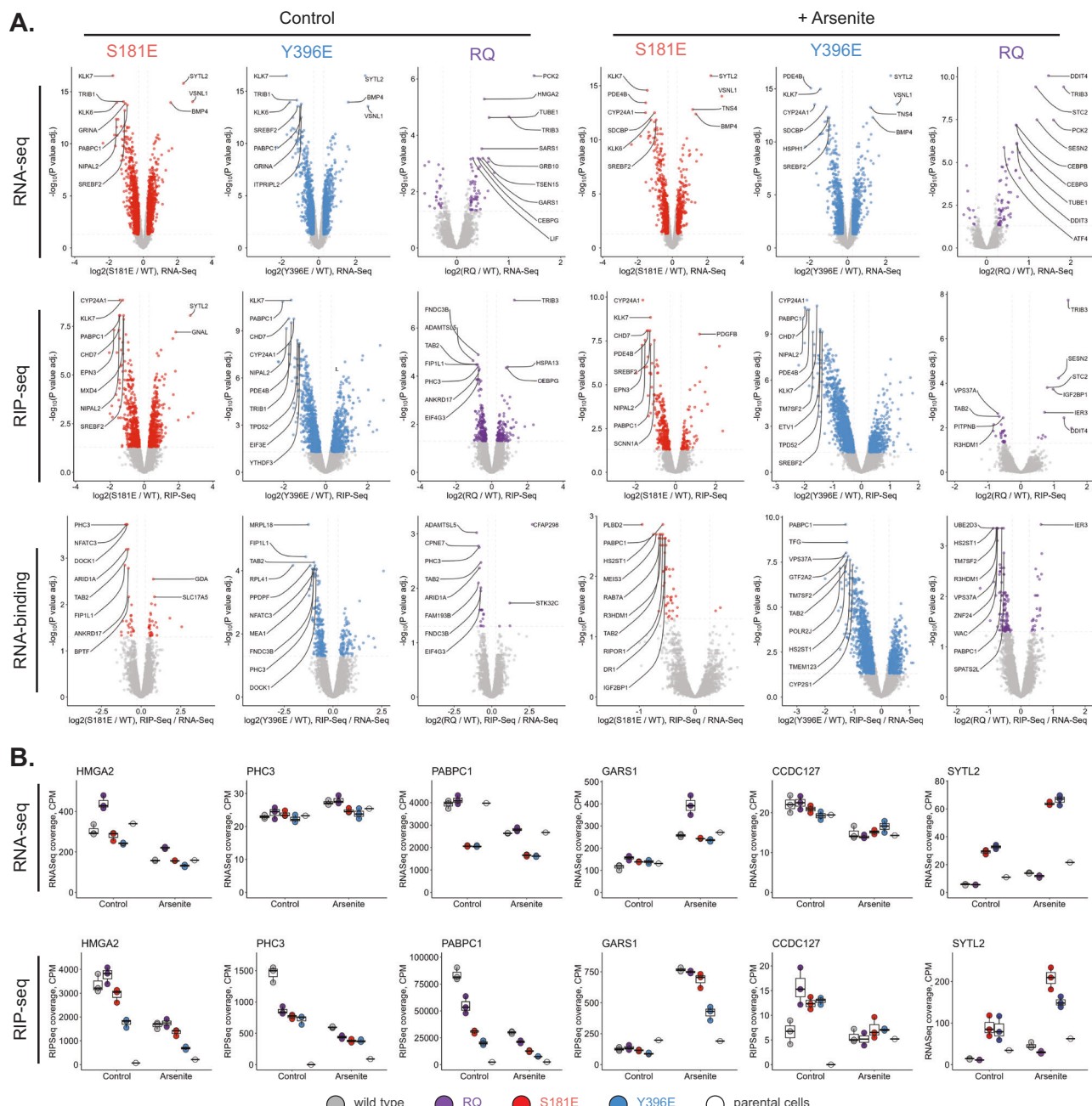

**Fig. 7 | Phosphomimetic mutants impact IGF2BP1's interaction with RNA in cells.** (A) Volcano plots showing changes in total transcriptome (RNA-Seq), IGF2BP1-bound transcript levels (RIP-Seq) and IGF2BP1 mRNA binding (RIP-Seq normalized to RNA-Seq) in HCT116 cells expressing mCherry-IGF2BP1 mutants S181E, Y396E, and RQ comparing to mCherry-IGF2BP1 wild-type. Arsenite stress was induced with a 2-hour treatment with 500 μM of sodium arsenite. Differentially regulated genes (fold change compared to the wild-type more than 20% with edgeR glmQLFTest[101] FDR-adjusted p value less than 0.05, n = 3 biological replicates) are highlighted, and their number is indicated in a figure. (B) RNA-Seq and RIP-Seq counts per million (CPM) graphs for selected genes. n = 3 biological replicates, data are shown as boxplots with all individual data points (center is a median, bounds of box are Q1 and Q3, and whiskers extend to ±1.5 × IQR, all individual data values are shown as dots). Source data are available in Supplementary Data 5.

leading to condensate formation. Our data suggest that the competing non-specific interactions of the highly promiscuous RRM1-2 pseudodimers increase the dynamics in RNA-protein interactions in IGF2BP1 condensates. In line with these observations, IGF2BP1 RNP condensates formed in vitro in cell lysates displayed increased dynamics compared to the RNP condensates reconstituted from minimal components. We anticipate that other RBPs compete with the available RNA-binding sites in cells, weakening the interactions between RBPs and RNAs and increasing the dynamics of RNA-protein interactions within condensates.

Systematic biochemical analyzes with truncation and point mutants revealed that the IGF2BP1 KH3-4 domains bind to RNA with the highest affinity, with KH3 showing the most substantial contribution. In contrast to what was shown for the IGF2BP1 chicken homolog ZBP1, we did not observe the looping of RNA around the KH3-4 domains, which results in an avidity effect by increasing the effective concentration for interactions with the second binding site for both the *ACTB*- and *XBP1*-derived RNAs[39,41]. However, we observed an avidity effect driven by the multivalent binding of the KH3-4 and KH1-2 domains to RNA. Importantly, RNA-binding mutants of the KH3 and

KH4 domains in full-length IGF2BP1 are bound tightly to RNA through multivalent interactions driven by RRM1-2 and KH1-2 domains. These data support earlier findings for the IGF2BP3 paralog, underlining the importance of combinatorial recognition of the IGF2BP targets[20,21]. We propose that the KH3-4 domains dock onto the RNA with a medium affinity and fast kinetics, and an avidity effect driven by the KH1-2 and RRM1-2 domains increases the affinity and specificity of IGF2BP1 binding to its targets. Phosphomimetic mutants of IGF2BP1 linkers bound to short model RNAs containing two IGF2BP1 binding motifs with an affinity similar to the wild-type protein. Notably, phosphomimetic mutants showed slight differences in their binding to the 200 nt-long *EIF2A*- and *XBP1*-derived RNAs but not for the *MYC*-derived RNA in EMSA. These data suggested the possibility of RNA-dependent differences in IGF2BP1-RNA interactions depending on phosphorylation. Altogether, our data converge on the model that linker phosphorylation does not abolish IGF2BP1's interaction with RNAs, yet might modulate IGF2BP1-RNA assemblies in cells as proposed earlier[7,32].

Combining cell biology and in vitro reconstitution methods, we found that IGF2BP1 phosphomimetic mutations introduced to the disordered linkers (S181, Y396) modulate its assembly into RNP granules. Intriguingly, while phosphomimetic mutant S181E at linker 1 decreases the dynamics and the size of IGF2BP1 RNP condensates, phosphomimetic mutant Y396E showed an opposite effect in vitro highlighting the regulatory potential of these sites. Notably, the phosphomimetic mutants did not largely impact the critical concentration of protein and RNAs for phase separation. This aligns with our findings that RNA interactions with folded domains drive IGF2BP1 phase separation. Using structural methods, we dissected how phosphomimetic mutants modulate the conformational state of IGF2BP1 and, thereby, the formation of RNP condensates. We hypothesized that linkers might be regulating IGF2BP1 condensation by three possible mechanisms: (i) self-association, (ii) interaction with folded domains, or (iii) interaction with RNAs. The NMR experiments, in vitro reconstitution, and MD simulations showed that linker 1 and linker 2 do not display a propensity to form condensates by themselves. Notably, the probability contact map derived from the MD simulations showed that linker 2 forms short-lived contacts with other linker 2 copies in trans, and the Y396E mutant showed an increased number of contacts compared to the wild-type, indicating that the mutation increases the propensity of linker 2 to self-associate. In addition, using NMR spectroscopy, we found that linker 2 forms low affinity contacts with all the folded domains in IGF2BP1, showing the strongest binding to the KH3-4 dimers. Notably, all folded domains bound weakly to the C-terminal segment (aa V390-M400) in linker 2 covering the Y396 phosphorylation site and those interactions were impaired for the Y396E phosphomimetic mutant. Those data uncovered that the molecular contacts between linker 2 and the folded domains were partially impaired in the Y396E mutant. Together with the MD simulations, these data suggest that the low-affinity interaction network of the linker 2 is rewired for the Y396E phosphomimetic mutant. The in vitro FRAP experiments showed that the IGF2BP1 Y396E mutant is more dynamic in condensates than the wild-type protein, consistent with the droplet-like morphology of the Y396E condensates. Our data suggest that changes in the low-affinity interaction networks formed by linker 2 increase IGF2BP1 dynamics in RNP condensates and facilitate condensate formation, resulting in the larger condensates we observed in vitro. Notably, the IGF2BP1 Y396E mutant expressed in cells formed smaller condensates with reduced dynamics. Increased compositional heterogeneity and complexity of the RNPs in cells compared to the in vitro experiments might cause these differences compared to the in vitro results.

The NMR experiments revealed that while linker 1 does not interact with folded domains of the IGF2BP1, it formed low-affinity contacts with RNA through its RGG/RG motif. We propose that these low-affinity contacts increase the dynamics of IGF2BP1 in the RNP

condensates. Significantly, these low-affinity interactions were impaired in the phosphomimetic mutant S181E. In vitro FRAP experiments showed that IGF2BP1 S181E displayed lower mobile fraction and dynamics than wild-type protein, indicating that it forms more stable assemblies in the condensates compared to the wild-type IGF2BP1. Our results converge on the model that IGF2BP1 S181 phosphorylation impairs low-affinity non-specific interactions of the linker with the RNA. Lack of competing low-affinity interactions result in more stable binding of S181E with RNA in the condensates, observed as decreased dynamics of IGF2BP1 in FRAP experiments. We speculate that decreased dynamics interfere with condensate growth and lead to the formation of smaller condensates, which we observed in vitro and cells for the S181E mutant. While phosphomimetic mutations do not completely recapitulate the biophysical properties of the phosphorylation of serine or tyrosine residues, perturbing the amino acid composition of the linker by a single point mutation has profound effects on IGF2BP1-RNP formation in vitro and in cells. Future studies using genetic code expansion to introduce phosphorylated residues at distinct sites in IGF2BP1 might reveal whether there are distinct differences between phosphorylation compared to the phosphomimetic mutants introduced here[79].

Remarkably, the RIP-seq analyzes revealed that compared to the wild-type IGF2BP1, the phosphomimetic mutants were less efficient in binding to thousands of target RNAs, with a more substantial effect observed during arsenite stress. Using genome-wide transcriptomics approaches, we found that the expression of IGF2BP1 phosphomimetic mutants in mammalian cells resulted in significant changes in the expression levels of several IGF2BP1-target RNAs. While we lack IGF2BP1 separation-of-function mutants to precisely decipher the role of RNP condensate formation in regulating mRNA metabolism, our data show the functional importance and the regulatory capacity of the linker regions for IGF2BP1-dependent posttranscriptional regulation in cells. In addition to phosphorylation, RGG sequences are methylated by protein arginine methyltransferase 1 (PRMT1), which can impact RNA binding of linker 1[80]. Therefore, the linkers might be subjected to more intricate regulation in cells in a context and tissue-specific manner.

Our findings highlight a highly tunable regulatory mechanism where modulation of low affinity interactions through phosphorylation could, in turn, impact the physical properties of RBPs in RNP condensates as observed for other systems[45,77,81]. Our data exemplify how the effects of PTMs are amplified in the condensate environment, thus providing an increased regulatory capacity to control biomolecular interactions in membrane-less organelles.

## Methods

### In vitro construct cloning

All mCherry-IGF2BP1 constructs were cloned into a pET-47 plasmid with an N-terminal mCherry using Gibson assembly. Single point mutations were generated through site-directed mutagenesis as well as Gibson assembly. IGF2BP1 (Uniprot: Q9NZI8) mutation involved changing the wild-type Ser181 or Tyr396 residues to glutamic acid. In order to establish U2OS and HCT116 cell lines stably expressing IGF2BP1, a vector that contains hPGK promoter was used. The promoter, together with NheI restriction site was introduced into the pLX303 expression vector using XhoI and BsrgI restriction enzymes. mCherry-tagged mutants of IGF2BP1 were amplified with pLX303_IGF2BP1_R(5'-CTCGCTAGCTCACTTCCTCCGTGCC-'3)and pLX303_mCherry_F (5'-CTCACCGGTGCCACCATGGTGAGCAAGG-'3) primers and incorporated into the modified pLX303 using the AgeI and NheI restriction sites. Sequencing confirmed IGF2BP1 integration.

### Cell culture

U2OS cell lines used in this study were grown in DMEM high glucose medium (Sigma) with 10% Fetal Bovine Serum (Gibco), 2 mM

Glutamine (Sigma), 1% Pen/Step (Sigma). HCT116 conditionally expressing Tet-OsTIR1 were obtained from Masato Kanemaki Lab (Natsume et al. 2016) and grown in McCoy's 5 A (Modified) medium supplemented as above. The cells were maintained at 37 °C, with 5% CO2, and were used for all biochemical experiments, live cell imaging, and Immunofluorescence. The cells were tested for mycoplasma contamination which was not detected.

### Establishment of IGF2BP2 and IGF2BP3 knockout cell lines

For knockout cell line generation, gRNA sequences (IGF2BP2: 5′-GA GCTGCCGGAGGTCGTCGG-3′; IGF2BP3: 5′-ACGCGTAGCCAGTCTTCA CC-3′) were cloned into the pSpCas9 (BB)−2A-GFP (PX458) (plasmid #48138; Addgene, (Ran et al. 2013)). Cells were transiently transfected using jetOPTIMUS reagent (Tamar, 101000051) and GFP-positive single-cell clones were FACS sorted at BD FACSAria IIIu at Max Perutz Labs BioOptics FACS Facility.

### Mass spectrometry– Sample Preparation

For mass spectrometry of IGF2BP1, we used HEK293T cells expressing IGF2BP1 tagged with split-GFP at the endogenous locus. These cells were the kind gift of Manuel Leonetti (Chan Zuckerberg BioHub, USA[82]). Briefly, the cells were generated by integrating GFP[1–10] into HEK293T cells by lentiviral integration. This cell line was then used to introduce GFP[11] into the IGF2BP1 using CRISPR-Cas9 gene editing approaches[83]. For the immunoprecipitation experiments, HEK293T cells were treated with the respective diluent as a control (DMSO for thapsigargin and tunicamycin or PBS for sodium arsenite). To induce stress, cells were treated with 250 μM sodium arsenite for 1 and 2 h, with 5 μg/mL tunicamycin for 4 h and with 250 nM thapsigargin for 1.5 and 4 h. IGF2BP1 was immunoprecipitated from HEK293T cells by GFP-trap magnetic beads (ChromoTek). The cells were lysed using cold lysis buffer (20 mM HEPES pH 7.4, 150 mM NaCl, 0.5 mM EDTA, 0.1% SDS, 0.5% Triton X-100, 0.2% Deoxycholate, 2x Complete™, EDTA-free Protease Inhibitor Cocktail (Roche), 0.5 mM PMSF and 1x PhosSTOP™ phosphatase inhibitor (Roche)). 25 μL of bead slurry was used for around 50 million HEK293T cells. The lysate was incubated with the GFP-trap beads for 2 h at 4 °C to allow binding of the protein. After binding, the beads were washed 5 times with 1 mL of ice-cold wash buffer (20 mM HEPES, 500 mM NaCl, 0.5 mM EDTA, 0.1% SDS, 0.5% Triton X-100, 0.2% Deoxycholate, 1x Complete™, EDTA-free Protease Inhibitor Cocktail (Roche)). Protein was eluted in 50 μL of 1x SDS sample buffer (50 mM Tris-HCl pH 6.8, 2% SDS, 0.1% Bromophenol blue, 10% glycerol+ 20 mM DTT) at +70 °C for 10 min. The eluate was loaded on SDS-PAGE and stained by Colloidal Coomassie[84]. The band corresponding to IGF2BP1 was cut and subjected to mass spectrometry analyzes at the Max Perutz Labs Mass Spectrometry Facility.

For mass spectrometry of IGF2BP3, HEK293T cells expressing IGF2BP3-tagged with split-GFP at the endogenous locus and HCT116 cells were used. IGF2BP3 from HEK293T split-GFP cell lines were immunoprecipitated under same experimental conditions as IGF2BP1. For the immunoprecipitation experiments, both cell lines were treated with DMSO or PBS as control. To induce stress, cells were treated with 250 μM of sodium arsenite for 1 or 2 hours, or 250 nM of thapsigargin for 1.5 or 4 hours or 5 μg/ml tunicamycin for 4 hours. For mass spectrometry of IGF2BP3 from HCT116 cells, we used antibodies against endogenous IGF2BP3. For immunoprecipitation three 15 cm (diameter) dishes of 60% confluent HCT116 per condition (around 50 million cells) were washed in ice-cold PBS, scraped, pelleted, and resuspended in 750 μL of ice-cold lysis buffer (25 mM HEPES pH 7.3, 150 mM NaCl, 0.5% NP-40, 0.5 mM EDTA, 10% Glycerol, 0.1% SDS, 0.2% Sodium Deoxycholate, 2x Complete™, EDTA-free Protease Inhibitor Cocktail (Roche), and 1x PhosSTOP™ phosphatase inhibitor (Roche)). Cells were lysed by incubation with the lysis buffer on ice for 15 min with intermittent vortexing and passing the cell suspension three times through a 25 G needle. The lysate was clarified using two-step

centrifugation for 5 min at 1,000 g and for 15 min at 13,000 g, and treated with 1 U/μL RNase T1 (Thermo Scientific) rotating at room temperature for 15 min. For IP from three 15 cm (diameter) dish 30 μg of anti-IGF2BP3 antibody was coupled to Dynabeads in 1 μg: 4 μL antibody: beads ratio. We used the MBL antibody (RN009P, lot 005) for the samples treated with tunicamycin and the respective 0.001% DMSO controls, and the Proteintech (14642-1-AP, lot 00090203) antibody for the thapsigargin, 0.0002% DMSO, arsenate and untreated control samples. The lysates were rotated at +4 °C for 4 hours for the IP. The unbound fraction was removed using a magnetic rack and the immunoprecipitated complexes were washed five times in 1 mL of ice-cold high salt wash buffer (25 mM HEPES pH 7.3, 400 mM NaCl, 0.5% NP-40, 0.5 mM EDTA, 10% Glycerol, 1x protease inhibitors cocktail) with 3-min incubations on ice. Protein was eluted in 50 μL of 1x SDS sample buffer (50 mM Tris-HCl pH 6.8, 2% SDS, 0.1% Bromophenol blue, 10% glycerol) without DTT at +70 °C for 10 min. DTT at 20 mM concentration was added to the collected eluates and the samples were heated at +70 °C for 10 min. Samples were separated by SDS-PAGE and stained with colloidal Coomassie. The band corresponding to IGF2BP3 was cut from the gel and submitted for tandem mass spectrometry at the Max Perutz Labs Mass Spectrometry Facility.

### Mass Spectrometry – Sample processing

Coomassie stained gel bands were excised, cut into small pieces and destained with a mixture of acetonitrile (ACN) and 50 mM ammonium bicarbonate (ABC). After shrinking the gel pieces in ACN, 20 mM dithiothreitol (DTT) was added to reduce disulfide bridges. After washing with ABC and ACN, free SH-groups were subsequently alkylated in 50 mM iodoacetamide. The in-gel digestion with trypsin was carried out overnight at 37 °C and was stopped by adding 10% formic acid to an end concentration of approximately 5%. Peptides were extracted from the gel with 5% formic acid by repeated sonication.

### Mass Spectrometry · Nano LC-MS/MS Analysis

Peptides were separated on an Ultimate 3000 RSLC nano-HPLC system using a pre-column for sample loading (Acclaim PepMap C18, 2 cm × 0.1 mm, 5 μm), and a C18 analytical column (Acclaim PepMap C18, 50 cm × 0.75 mm, 2 μm, all HPLC parts Thermo Fisher Scientific), applying a linear gradient from 2% to 35% solvent B (80% ACN, 0.08 % formic acid; solvent A 0.1 % formic acid) at a flow rate of 230 nL/min over 60 min. Eluting peptides were analyzed on a Q Exactive HF-X Orbitrap mass spectrometer coupled to the HPLC via Proxeon nanospray-source (all Thermo Fisher Scientific) equipped with coated emitter tips (New Objective).

The mass spectrometer was operated in data-dependent acquisition mode. Survey scans were obtained in a mass range of 375-1500 m/z with lock mass on, at a resolution of 120000 at 200 m/z and a normalized AGC target of 3E6. The 10 most intense ions were selected with an isolation width of 1.6 m/z, for max. 200 ms at a normalized AGC target of 1E5, and then fragmented in the HCD cell at 28% normalized collision energy. MS/MS Spectra were recorded at a resolution of 30000. Peptides with a charge of +1 or >+6 were excluded from fragmentation; the peptide match feature was set to "preferred" and the exclude isotope feature was enabled. Selected precursors were dynamically excluded from repeated sampling for 30 s.

For the parallel reaction monitoring (PRM) analysis survey scans were acquired in a mass range of 375-1500 m/z with lock mass off, at a resolution of 30000 at 200 m/z and a normalized AGC target of 3E6. The PRM parameters - precursor m/z and retention time - were built based on the gel samples measured with DDA. Precursors of 33 peptides of interest (14 phosphorylated peptides and their unmodified counterparts plus 5 reference peptides) were isolated in a 0.7 m/z window and fragmented with 28% HCD collision energy. Orbitrap resolution was set to 30000, the normalized AGC target to 2E5. Maximal injection time for modified peptides was set to 200 ms.

## MS data analysis for identification of phosphorylation sites

The RAW MS data were analyzed with FragPipe (20.0), using MSFragger (3.8)[85], IonQuant (1.9.8)[86], and Philosopher (5.0.0)[87]. The default FragPipe workflow for label free quantification (LFQ-MBR) was used, except that "Add MaxLFQ", "Match between runs (MBR), and "Normalize intensity across runs" were turned off. Cleavage specificity was set to Trypsin/P, with two missed cleavages allowed. The protein FDR was set to 1%. A mass of 57.02146 (carbamidomethyl) was used as fixed cysteine modification; methionine oxidation, protein N-terminal acetylation, and serine/threonine/tyrosine phosphorylation were specified as variable modifications. MS2 spectra were searched against the *H. sapiens* reference proteome from Uniprot (Proteome ID: UP000005640, release 2023.03) containing 20598 entries, concatenated with a database of 379 common laboratory contaminants (in house database).

Computational analysis was conducted using Python along with two in-house Python libraries, "MsReport" (version 0.0.20) and "XlsxReport" (version 0.0.6)[88]. To compile a list of confidently identified phosphorylation sites for IGF2BP1, IGF2BP2, and IGF2BP3, we utilized the individual "ion.tsv" tables generated by FragPipe. Initially, the ion tables were concatenated and non-phosphorylated peptide ions were filtered out. Subsequently, multiple phosphorylated peptide ions were selected, and separate entries were generated for each phosphorylated site by duplication. The specific site localization probability was then extracted for each of the duplicated site entry. Entries with a peptide probability lower than 95% or a phosphorylation site localization probability less than 80% were removed. The expanded ion table was summarized by aggregating entries of individual phosphorylation sites, and the best peptide probability and site localization probability for each site were extracted. Total spectral counts were calculated as the sum of all PSMs (peptide spectrum matches) identifying specific phosphorylation site, excluding LC-MS runs with PRM measurements.

## MS data analysis of PRM measurements for phosphorylation site quantification

LC-MS runs with PRM measurements were analyzed in Skyline (22.2.0.351)[89]. A list containing the 33 peptides targeted by PRM and the raw LC-MS files were imported into Skyline. All peptides and their transitions were validated manually based on retention time, relative ion intensities, and mass accuracy. Extracted ion chromatograms (XICs) were generated for the product ions of all selected peptides, and peak areas were exported from Skyline. The intensity of each peptide was calculated by summing the XICs of the three most intense, interference-free transitions. Subsequently, peptide intensities across different samples were normalized using four reference peptides. First, the sum of the reference peptide intensities was calculated for each sample. Next, these summed intensities were divided by the average across all samples to derive normalization factors. Finally, peptide intensities of each sample were divided by the respective normalization factor. To ensure reliable data, peptides with incomplete quantification or exhibiting an average coefficient of variation exceeding 2 between the two replicates were excluded from further analysis. The intensities of modified peptides covering the same phosphorylated protein site were summed to create site-level intensities. The intensities of the corresponding unmodified peptide counterparts of each site were also summarized to create site level counter intensities. Estimated site occupancy was calculated as "Site intensity" / ("Site intensity" + "Counter intensity") * 100. For plotting, site-level intensities were log2 transformed and normalized to the average intensity of the respective control samples.

## In vitro phosphorylation of IGF2BP1 by Src kinase

5 μM IGF2BP1 full-length wild-type were incubated with 0.5 μM Src (Merck 23-042), 1 mM ATP, 2 mM $MgCl_2$, 150 mM NaCl, 25 mM HEPES pH 7.3 and 0.5 mM TCEP for 2 h at room temperature. Control sample was incubated without Src. Samples were subsequently flash frozen in liquid nitrogen.

## ProAlanase digest of recombinant Src-treated IGF2BP1 for LC-MS/MS peptide mapping

1.5 μg of purified recombinant IGF2BP1 were denatured in 6 M urea and 50 mM TEAB, pH 8. Disulfide bridges were reduced with 10 mM dithiothreitol and free thiols subsequently alkylated with 20 mM iodoacetamide. Remaining iodoacetamide was quenched with 5 mM DTT before adjusting the pH to 1.5 using 30 mM HCl – this step also diluted the urea concentration to 0.3 M to allow for efficient digestion. ProAlanase (Promega) was added in a 1:50 ratio of enzyme to protein and the sample incubated at 37 °C for 1.5 h. The resulting peptides were digested using C18 Stagetips[90].

## ProAlanase digest of immuno-enriched IGF2BP1 for LC-MS/MS peptide mapping

For the ProAlanase digestion analysis, cells were treated as described in "Mass spectrometry– Sample Preparation". Control, thapsigargin and arsenite treated cells were prepared in duplicates. After removing the final wash, the beads were resuspended in 1 M urea in 50 mM ammonium bicarbonate. Disulfide bridges were reduced with 10 mM DTT and free thiols subsequently alkylated with 20 mM iodoacetamide. Remaining iodoacetamide was quenched with 5 mM DTT. 100 mM glycine was added to the beads until pH 2 was reached and the urea concentration was diluted to 0.3 M. 300 ng ProAlanase (Promega) were added to the beads and incubated for 2 h at 37 °C. The supernatant was desalted using C18 Stagetips[90].

## Nano LC-MS/MS analysis of ProAlanase digested samples

Peptides were separated on a Vanquish Neo nano-flow chromatography system (Thermo Fisher), using a trap-elute method for sample loading (Acclaim PepMap C18, 2 cm × 0.1 mm, 5 μm, Thermo Fisher), and a C18 analytical column (Acclaim PepMap C18, 50 cm × 0.75 mm, 2 μm, Thermo Fisher), applying a segmented linear gradient from 2% to 35% and finally 80% solvent B (80 % acetonitrile, 0.1 % formic acid; solvent A 0.1 % formic acid) at a flow rate of 230 nL/min over 60 min.

Eluting peptides were analyzed on an Exploris 480 Orbitrap mass spectrometer (Thermo Fisher), which was coupled to the column with a Nanospray Flex ion-source (Thermo Fisher) using coated emitter tips (PepSep, MSWil). The mass spectrometer was operated in data-dependent acquisition mode. Survey scans were obtained in a mass range of 375–2000 m/z, at a resolution of 120,000 at 200 m/z and a normalized AGC target of 3E6 and a cycle time of 2 s. The most intense ions were selected for fragmentation with an isolation width of 1.4 m/z, for max. 200 ms injection time at a normalized AGC target of 200%, and then fragmented in the HCD cell at 30% normalized collision energy. MS/MS Spectra were recorded at a resolution of 30,000. Peptides with a charge of >+6 were excluded from fragmentation. Selected precursors were dynamically excluded from repeated sampling for 20 s. In addition, target peptides covering the IGF2BP1 site Y396 (unmodified and phosphorylated) were added to an inclusion list to increase the likelihood of fragmentation.

## Data analysis for identification of phosphorylation sites in ProAlanase digested samples

The RAW MS data were analyzed with FragPipe (22.0/22.0), using MSFragger (3.8/4.1)[85], IonQuant (1.9.8/1.10.27)[86], and Philosopher (5.0.0/5.1.1)[87]. The default FragPipe workflow for label free quantification and identification of phosphopeptides (LFQ-phospho) was used, except that "Normalize intensity across runs" was turned off. Cleavage specificity was specified to cut C-terminal to proline and alanine, with five missed cleavages allowed. The protein FDR was set to 1%. A mass of 57.02146 (carbamidomethyl) was used as fixed cysteine modification; methionine oxidation, protein N-terminal acetylation, and serine/

threonine/tyrosine phosphorylation were specified as variable modifications. MS2 spectra were searched against the *E. coli* reference proteome (UniProt, UP000000625, release 2024.01) and the IGF2BP1 recombinant sequence or against the *H. sapiens* reference proteome (UniProt, UP000005640, release 2024.01), in both cases concatenated with a database of 379 common laboratory contaminants (https://github.com/maxperutzlabs-ms/perutz-ms-contaminants).

Downstream computational analysis and reporting was conducted using Python with two in-house libraries, "MsReport" (version 0.0.24) and "XlsxReport" (version 0.0.6)[88], as described above. Extracted ion chromatograms were generated in Skyline-daily (24.0.9.197)[89].

### Estimation of the linker 2 phosphorylation levels from in vitro and in vivo results

The intact mass spectrometry analyzes of in vitro phosphorylated IGF2BP1 by SRC kinase showed an approximate 1:1 ratio of unmodified to single phosphorylated protein (Supplementary Data 1). Peptide mapping from these samples revealed that the Y396 phosphopeptides showed 10–20% of the signal of the unmodified peptide (please see Supplementary Fig 1C). In the samples, where we IP-ed IGF2BP1 from cells, we detected the unmodified peptide with 2 to 5 E10 intensity (Supplementary Fig. 1D). Under the conservative assumption that phosphopeptides display 10% signal of the unmodified peptide, under those conditions phosphopeptides found in 100% stoichiometry of total protein would be expected to have an intensity in the range of 2–5E9. Similarly, at 1% stoichiometry, we would expect 2–5E7 signal intensity for the phosphopeptides. The lowest IGF2BP1 peptide signal detected in those experiments was in the range of 1E7, meaning that it would have been possible to detect the phosphopeptides at 1% stoichiometry at that intensity in our experiments. Altogether, our data suggest that the abundance of the phosphorylated Y396 is probably below 1%, if it is present at all.

### Protein Expression and Purification

Full-length IGF2BP1 constructs were cloned into a pGEX-6P-2 vector containing an N-terminal GST-tag and a 3 C cleavage site. These proteins were expressed in Rosetta (DE3) cells grown to an OD of -0.7 and induced with 400 μM IPTG. Cells were grown over night at 20 °C, resuspended in GST lysis buffer (25 mM HEPES pH 7.2, 1 M NaCl, 5% glycerol, 2 mM DTT, 0.5 mM PMSF, 2 mM EDTA), pelleted and frozen in liquid nitrogen.

mCherry-IGF2BP1 constructs were cloned into a pET-47b vector with an N-terminal mCherry- and a C-terminal Deca-His-tag with a 3 C cleavage site and expressed in Rosetta (DE3) cells. Shorter IGF2BP1 proteins (RRM1-2, KH1-4, KH1-2, KH3-4) were cloned into a pET-47b vector with an N-terminal Hexa -His-Tag and expressed in BL21 cells. His-tagged proteins were induced with 800 μM IPTG and cells were resuspended in His lysis buffer (25 mM HEPES pH 7.2, 1 M NaCl, 5% glycerol, 5 mM beta-mercaptoethanol, 0.5 mM PMSF, 20 mM Imidazole).

Linker peptides were cloned into a pET-21 vector with an N-terminal His- and SUMO-tag. These proteins were expressed in BL21 cells by growing them in to an OD of -0.7, induction with 1 mM IPTG and incubation for 3.5 h at 25 °C. The cells were than resuspended in phosphate lysis buffer (20 mM phosphate buffer pH 7.2, 1 M NaCl, 5% glycerol, 5 mM beta-mercaptoethanol, 0.5 mM PMSF, 20 mM Imidazole). The obtain $^{15}$N labeled linker proteins, the cells were grown in M9 minimal medium supplemented with $^{15}$NH$_4$Cl. To get $^{13}$C-$^{15}$N-labeled proteins, glucose was replaced with $^{13}$C-glucose.

Proteins were purified by resuspending the pellet in lysis buffer and lysing the cells in a EmulsiFlex C3 homogenizer. After pelleting non-soluble cell debris by centrifugating the lysate at 40,000 g for 30 min at 4 °C, the supernatant was loaded onto the respective affinity column. Full-length proteins were loaded onto two serially connected GST HiTrap HP 5 mL columns, washed with GST wash buffer (25 mM

HEPES pH 7.2, 500 mM NaCl, 5% glycerol, 2 mM DTT, 0.5 mM PMSF, 2 mM EDTA), and eluted with a gradient of 20 mM reduced glutathione in GST wash buffer. His-tagged or His-SUMO-tagged protein were purified on a HisTrap HP 5 mL column by washing them with His wash buffer (25 mM HEPES pH 7.2, 500 M NaCl, 5% glycerol, 5 mM beta-mercaptoethanol, 0.5 mM PMSF, 20 mM Imidazole) and eluted with a gradient of 1 M Imidazole in wash buffer. Bound nucleotides were removed by loading the proteins onto a Heparin HiTrap HP 5 mL column after diluting the NaCl concentration to 100 mM (with wash buffer without NaCl), washing them with wash buffer (25 mM HEPES pH 7.2, 100 mM NaCl, 5 % glycerol, 2 mM DTT, 0.5 mM PMSF, 2 mM EDTA, which was excluded when purifying His-tagged proteins) and eluting them with a gradient of 1 M NaCl in wash buffer. The GST- and His- buffer was cleaved off by incubating the eluted protein with GST-3C or His-3C overnight at 4 °C. For cleaving off the SUMO-tag, proteins were incubated with SENP. The respective tag and the 3 C protease were extracted by running the cleaved protein over a GST HiTrap 5 mL or HisTrap HP 5 mL column by using the same buffers as in the affinity purification. For KH1/2 the His-tag was not removed and thus the overnight incubation with protease and the negative His-affinity purification was omitted. The flow through was pooled and further purified via size exclusion chromatography (SEC). Full-length and KH1/2, KH1/4, KH3/4 (no DTT in the buffer) and all respective constructs were eluted with SEC buffer (25 mM HEPES, 150 mM NaCl, 2 mM DTT). Linker constructs were eluted in phosphate SEC buffer (20 mM phosphate buffer pH 7.2, 150 mM NaCl). Protein concentration was measured by using a Nanodrop (full length proteins, linker) or BCA assay (shorter constructs).

### In vitro transcription and RNA purification

The RNAs *XBP1* 201 nt, *EIF2A* 200 nt and *MYC* 191 nt were produced by adding a T7 promoter sequence (TAATACGACTCACTATAGGG) and using the HiScribe T7 RNA Synthesis Kit (NEB E2040S). Template DNA was digested by adding DNase I and incubation for 15 min at 37 °C. RNA was denatured by boiling it for 5 min in denaturing buffer (10 m urea, 1 mM EDTA, 0.1% SDS, 0.5 mg/mL xylene cyanol, 0.5 mg/mL bromophenol blue) and run on a 6% acrylamide, 10 M urea TBE (89 mM Tris, 89 mM borate, 2 mM EDTA) denaturing gel in 1x TBE buffer for 2 h at 100 V. The gel was stained with SYBR Gold and the bands containing the RNA with the correct size cut out. The RNA was extracted by crushing the gels with a pestle and shaking in RNase-free H$_2$O with 1x SUPERaseIn for 1 h at room temperature. The gel pieces were separated by centrifugation in Spin-X filter tubes for 5 min at 20.000 g. The RNA was further purified via Phenol-Chloroform extraction by mixing the RNA sample first with a 1:1 volume of phenol, vortexing and centrifugation for 2 min at 20,000 g then taking off the supernatant and mixing it with a 1:1 volume of chloroform, vortexing and centrifugation for 1 min at 20,000 g. The supernatant is then mixed with a 10: 1 volume of 3 M sodium acetate and 1: 1 volume of ice-cold isopropanol. After precipitating the RNA overnight at −80 °C, the RNA was washed twice with ice-cold 80% ethanol and centrifugation for 30 min at 20,000 g and 4 °C, dried for 10 min and resuspended in RNase free H$_2$O. The RNA concentration was determined by using a Nanodrop.

### EMSA Assays

To investigate protein-RNA interaction via Electrophoretic Mobility Shift Assays, proteins were thawed and centrifuged for 15 min at 20000 g and 4 °C. RNA was refolded by heating up to 95 °C and cooling down to 25 °C in steps of 2 °C/min. The protein and RNA was mixed in EMSA buffer (25 mM HEPES pH7.3, 150 mM NaCl, 5% glycerol, 2 mM EDTA, 10 μg/mL Heparin, 0.5 mM TCEP) to the specified concentrations. All samples were incubated on ice for 30 min. A 5% TBE gel with 5% glycerol was prepared by pre-running it for 30 min at 210 V in 0.5x TBE buffer at 4 °C. 10 μL per sample and 0.5 μL ladder (RiboRuler low range SM1831) were loaded onto the gel and run for 30 min at

210 V in 0.5x TBE buffer at 4 °C. RNA was stained with SYBR Gold and the gel was imaged with BioRad ChemiDoc or iBright CL1500 and quantified with the corresponding software supplied by the manufacturer. RNA binding was determined by using mean intensities and defining the sample with 0 nM protein as 0 and the gel background as 1 on the y-axis. $K_{1/2}$'s were calculated in GraphPad Prism by using the Hill equation (1) $Y=Y_{free}+Y_{bound}/(1 + 10\wedge((\log10(K_{1/2})-\log10(x))*n))$

## Fluorescence anisotropy assays

Proteins were thawed and centrifuged for 15 min at 20,000 g and 4 °C. To be able to measure fluorescence anisotropy the corresponding RNAs were obtained from IDTDNA with a 5′ fluorescein tag. The sample with the highest concentration was prepared in anisotropy buffer (25 mM HEPES 7.3; 150 mM NaCl, 0.025% Tween 20, 10 µg/mL Heparin, 2 mM EDTA, 0.5 mM TCEP) with 10 nM of fluorescence labeled RNA. A concentration series was created by diluting the sample 1:1 with 10 nM RNA in anisotropy buffer. Subsequently, the samples were incubated on ice for 30 min. The fluorescence anisotropy was measured in a cuvette using an Edinburgh Instruments FS5 spectrofluorometer at threes wavelengths (515, 520, and 525 nm) with a slit width of 5 nm at 20 °C with an excitation wavelength of 485 nm, a slit width 5 nm and a dwell time of 1 s. The G-factor, which was measured once for every experiment and then used to calculate the anisotropy for each sample in the experiment, was determined in the beginning of the experiment by measuring 10 nM RNA in anisotropy buffer. The fluorescence anisotropy value was calculated by the manufacturer's software. $K_{1/2}$'s were calculated in GraphPad Prism by using a dose response curve to fit the data using a model following the Hill equation (1) $Y=Y_{free}+Y_{bound}/(1 + 10\wedge((\log10(K_{1/2})-\log10(x))*n))$

## RNP granule formation

To investigate the formation of RNP granules, proteins were thawed and centrifuged for 15 min at 20,000 g and 4 °C. RNA was refolded by heating up to 95 °C and cooling down to 25 °C in steps of 2 °C/min. The proteins were diluted in RNase free buffer (25 mM HEPES pH 7.3, 150 mM NaCl, 0.5 mM TCEP). RNP granule formation was induced by the addition of the RNA with the respective final concentration. Right after RNA addition and mixing 50 µL of the sample were pipetted into the well of a Greiner sensoplate, black, 96-well, glass bottom plate which was coated with 1% Pluronic F-127 by washing the well with 200 µL H₂O, then incubation with 1% Pluronic F-127 for 2 h at room temperature, four times washing with 200 µL H₂O, and once washing with 100 µL buffer. The plate was incubated on the microscope for 90 min to allow droplet formation while avoiding moving and disturbing the granules. To determine the formation of RNP granules for the phase diagram, 20 µL of each sample were prepared and granule formation induced by adding RNA. These samples were pipetted into a 384-well plate that was similarly coated with 1% Pluronic F-127 and incubated for 90 min. To avoid evaporation in these low volume sample, the wells were covered with mineral oil (CAS: 8042-47-5). RNP formation was qualified by the presence of droplet- or mesh-like condensates. The samples were imaged with a Zeiss Axio Observer Z1 in bright field mode, EC Plan−Neofluar 100x/1.3 Oil M27, Orca Flash 4.0 LT+ Camera, VIS-LED at 50% intensity and 20 ms exposure time.

For the quantification of the condensates, 5% of the respective N-terminal mCherry-labeled IGF2BP1 construct was used in the samples. To avoid non-specific interaction of the mCherry-labeled proteins with the glass bottom, the wells were coated with 2 mg/mL BSA in addition to 1% Pluronic F-127 by washing the well with 200 µL H₂O, then incubation with 100 µL 1% Pluronic F-127 for 2 h at room temperature, twice washing with 200 µL H₂O, incubation 100 µL 2 mg/mL BSA in PBS, twice washing with PBS and once washing with 100 µL buffer. RNP granule formation was induced by the addition of the RNA with the respective final concentration. Right after RNA addition and mixing 50 µL of the sample were pipetted into the well. Imaging was carried

out at the microscope setting describe above with additional acquisition of fluorescence images with a Colibri 7 lamp using the 567 nm LED module and a Zeiss filter set 90 HE, 80% intensity and 250 ms exposure time. Four adjacent tiles were imaged to increase to field of view area. Two fields of view were imaged and analyzed per well and the experiment was performed in triplicates. To analyze the data, the tiles were stitched together using the Zeiss ZEN Blue software and cropped to 3200 ×3200 pixels to have similar sized images. Condensate quantification was performed in ImageJ. Background was subtracted with a rolling ball radius of 200 and a sliding paraboloid, the "Default Dark" threshold was set and the image converted to a mask, watershed was used to separate very close condensates and finally the particles were analyzed with a minimum size of 200, outlines and a summary were created. The two-sided Welch's t test was used to determine statistical significance.

## FRAP of RNP granules

To investigate the dynamics of IGF2BP1 in RNP granules, proteins were thawed and centrifuged for 15 min at 20000 g and 4 °C. RNA was refolded by heating up to 95 °C and cooling down to 25 °C in steps of 2 °C/min. The proteins were diluted in RNase free buffer (25 mM HEPES pH 7.3, 150 mM NaCl, 0.5 mM TCEP). IGF2BP1 full-length wild-type, S181E and Y396E were mixed with 5 % of the respective mCherry-labeled IGF2BP1 construct. A Greiner sensoplate, black, 96-well, glass bottom plate was coated with 1% Pluronic F-127 by washing the well with 200 µL H₂O, then incubation with 1% Pluronic F-127 for 2 h at room temperature, twice washing with 200 µL H₂O, incubation 100 µL 2 mg/mL BSA in PBS, twice washing with PBS and once washing with 100 µL buffer. RNP granule formation was induced by the addition of the RNA with the respective final concentration. Right after RNA addition and mixing 50 µL of the sample were pipetted into the well. The plate was incubated at the microscope for 90 min to allow droplet formation and avoiding moving and disturbing the granules. FRAP was performed on a Zeiss Axio Observer equipped with a Yokogawa CSU-X1 Nipkow spinning disk unit, EC Plan−Neofluar 100x/1.30 Oil Iris objective, 561 nm DPSS laser, Visitron controller, ET605/70 emission filter and a pco.edge sCMOS camera. Images were acquired with 20% laser power and 100 ms exposure time. FRAP was performed with a time interval of 3 s, 3 images were taken before bleaching, then selected regions were bleached with 30% FRAP-Laser power for 1 ms per pixel in 5 cycles and 361 frames were taken in total per experiment.

## Phase separation assay

To determine whether Linker 1 or Linker 2 can phase separate on their own, the proteins were thawed and centrifuged for 15 min at 20000 g and 4 °C. The proteins were diluted in buffer (25 mM HEPES pH 7.2, 150 mM NaCl). A Greiner sensoplate, black, 96-well, glass bottom plate was coated with 1% Pluronic F-127 by washing the well with 200 µL H2O, then incubation with 1% Pluronic F-127 for 2 h at room temperature, four times washing with 200 µL H2O, and once washing with 100 µL buffer. 25 µL protein dilution were pipetted into the well. 25 µL of 30% PEG 8000 in buffer were added and carefully mixed to induce phase separation. The plate was incubated on the microscope for 60 min to allow droplet formation. The wells were imaged with a Zeiss Axio Observer Z1 in bright field mode, EC Plan−Neofluar 100x/1.3 Oil M27, Orca Flash 4.0 LT+ Camera, VIS-LED at 50% intensity and 20 ms exposure time.

## Turbidity assay

Proteins, RNA and the plate were prepared as described in "RNP granule formation". After induction of granule formation by adding RNA, 50 µL of the samples were pipetted into the plates, the plate put into the plate reader and the experiment started immediately. We used a BioTek Synergy H1 plate reader to measure turbidity. At first, the samples were shaken for 5 s to ensure homogenous distribution of the

granules, then the turbidity was monitored by measuring the absorbance of the sample at 480 nm 46 times with a time interval of 20 s. The data was analyzed with GraphPad Prism. The one-phase association equation (2) $Y = Y0 + (Plateau\text{-}Y0)*(1\text{-}exp(\text{-}K*x))$ was used to quantify condensate formation.

## Dynamic light scattering (DLS)

Proteins, RNA and buffer (25 mM HEPES pH 7.3, 150 mM NaCl, 0.5 mM TCEP) were filter with Spin-X filter tubes for 5 min at 20000 g. Proteins were diluted with buffer to the final concentration ranging from 500 nM to 1 μM, and cluster formation was induced by adding RNA at concentrations ranging between 250 nM- 1 μM (please see figure legends Supplementary Figure 5J). The samples were pipetted into a 1536-well plate, covered with silica oil and centrifuged at 500 g for 1 min to remove air bubbles. The plate was incubated for 90 min at 25 °C in a Wyatt Dynapro II DLS plate reader. Data acquisition and processing was performed using the software DYNAMICS V7 from Wyatt. 20 acquisitions were recorded with 2 s acquisition time. Acquisitions that did not baseline within a ± 0.01 interval or showed aggregation or vibration artefacts were discarded. The data was measured in technical triplicates and the resulting curves from three experiments were fitted with the Raynals online tool at the EMBLEM website (https://spc.embl-hamburg.de/app/raynals)[91]. Data is represented as mean of these fits, error bars show the standard deviation. Numbers of curve fits averaged per figure: 500-250: WT 6, S181E 5, Y396E 6; 600-300: WT 7, S181E 8, Y396E 7; 800-400: WT 5, S181E 5, Y396E 5; 600-600: WT 5, S181E 9, Y396E 7; 800-800: WT 5, S181E 9, Y396E 8; 1000-1000: WT 6, S181E 6, Y396E 5).

## MD simulations

We performed molecular dynamics (MD) simulations of linker 1 wildtype, linker 1 S181E, linker 2 wild-type, and linker 2 Y396E using the Martini 3 force field with rescaled protein-water interactions[65]. We set the scaling parameter λ = 1.06. All linkers were modeled as polypeptide chains with no secondary structure (coils) using UCSF Chimera[92] and then martinised. Each simulation box (30x30x30 nm³) was set up with 33 randomly placed copies of the same polypeptide chain, water and 0.15 M NaCl. We performed an energy minimization using the steepest descent algorithm. Then we equilibrated the system by running a 10 ps-long simulation using a 1 fs time step and restraining the position of protein backbone beads by using harmonic potentials with force-constants of 1000 kJ mol⁻¹ nm⁻². Afterwards, we ran another 2.1 ns without restraints in the NVT ensemble and a final equilibration of 21 ns in the NPT ensemble, in both equilibration steps we used a 30 fs time step ns. After equilibration, we ran the production phase using a 20 fs time step. The temperature in the simulation box was controlled by a velocity rescale thermostat[93](reference temperature T_ref = 300 K, coupling time constant tau_T = 1 ps). The Parrinello–Rahman barostat [Parrinello, 1981] (reference pressure p_ref = 1 bar; coupling time constant τ_p = 24 ps) was used for the last equilibration step and for the production run. The simulations were performed using the Martini 3.0 forcefield[94] and the GROMACS 2020.5 software[95].

The contact maps for cis-interactions were calculated using the Contact Map Explorer Python analysis package [https://contact-map.readthedocs.io/en/latest/index.html] (version 0.7.0). For each simulation, we calculated all the contacts between all atoms of the same polypeptide at each frame of the trajectory (ignoring atoms of 2 neighboring residues in each direction). A contact is defined between two atoms that are within a distance of 0.45 nm. The contributions from all chains at each frame were summed up in a single matrix and normalized by the number of frames and chains. For the final plots, the results were shown in a matrix where a value of contact frequency p corresponds to each pair of residues, where p is the max value of the contact frequencies computed for every atom in the residue pair.

The radius of gyration probability distributions computed from MD simulations were compared to the Analytical Flory Random Coil (AFRC) distribution[67]. AFRC is an analytical model of unfolded polypeptides that behave as ideal chains, so it is suitable to be used as a reference. We computed the AFRC counterparts of linker 1, linker 2 and their phosphomimetic mutants using the AFRC model available via Google Colab notebook: (https://colab.research.google.com/drive/1WHw8ous7IgcKd2LKYuJLeBTlkdEYoRAk?usp=sharing).

## SAXS experiments

SAXS samples were prepared at concentrations > 10 mg/ml in a 250 μL volume and each experiment performed in duplicates. All samples were measured at beamline BM29 at the ESRF facility in Grenoble. A Superdex 200 Increase 10/300 GL size exclusion column was equilibrated with SAXS buffer (25 mM HEPES, 150 mM NaCl, 0.5 mM TCEP). Samples were applied to the column and run with 0.5 ml/min. 1300 Frames with 0.5 frames per second were acquired. We used the ATSAS 3.1.3 data analysis software for data processing. SEC-coupled SAXS data was analyzed with CHROMIXS. 25 frames for buffer and sample were selected and averaged. The buffer subtracted data was analyzed and plotted in PRIMUSQT.

## SAXS analyzes and EOM calculations

The data were processed with the SAXSQuant software (version 3.9), and de-smeared using the programs GNOM[96] and GIFT (PCG software). EOM analysis was performed with the ATSAS 2.5 package (EMBL, Hamburg). EOM calculations were carried out using the EOM program[64] and using default settings. A random pool of 100,000 independent structures was generated using the primary sequence and the available structure of IGF2BP1 domains. All disordered regions were randomized. Using the built-in genetic algorithm and using the default settings, a subset of a few independent structures were selected that described the experimental SAXS best and used to prepare the figures showing $R_G/D_{max}$ distributions.

## NMR Experiments

NMR spectra were recorded on a 600 MHz Bruker Avance Neo 600 spectrometer. ¹H-¹⁵N Heteronuclear Single Quantum Coherence (HSQC) experiments were performed by thawing the proteins and centrifuging for 15 min at 20,000 g and 4 °C. The proteins were diluted with 20 mM phosphate buffer pH 7.2, 150 mM NaCl, 10% D₂O to the desired concentration. For the titration experiments with the folded domains, His-RRM1-2, His-KH1-2 or His-KH3-4 were prepared as above. The sample with the highest protein concentration was prepared first, measured and then diluted 1: 1 with 50 μM ¹⁵N-Linker 1 or 25 μM ¹⁵N-Linker 2 for higher dilution samples. All HSQC experiments with Linker 1 were performed at 15 °C. All experiments with 12xUG RNA or XBP1 10 nt RNA were performed in RNase free buffer with 25 mM HEPES pH 7.3, 150 mM NaCl. Experiments with 15N-labeled KH1-4 and L2-KH3-4 were conducted in 25 mM HEPES pH 7.2, 150 mM NaCl, 2 mM DTT. For all His-RRM1-2 experiments 2 mM DTT was used.

The spectra were processed and phased using NMRPipe[97] and further analyzed with CCPNMR Version 3[98]. Chemical shift perturbations were calculated with the following formula:

$$CSP = \sqrt{\left(\delta H^2 + 0.14 * \delta N^2\right)} \qquad (3)$$

δH and δN are the chemical shift differences compared to the apo protein. Peaks unresolved in the concentrations used for the titration experiments were excluded from the analysis.

Peaks were assigned by obtaining HNCO, HNCACO, HNCACB, HNCOCACB, HNCANNH and HNCOCANNH spectra of 300 μM ¹³C-¹⁵N-Linker 1 wild-type and 300 μM ¹³C-¹⁵N-Linker 2 wild-type and the respective HSQC spectra. Peak assignments were performed in

CCPNMR. NMR assignments are available in Biological Magnetic Resonance Bank.

## Stress granule reconstitution in cell lysates

Split-GFP tagged IGF2BP1 cells were seeded in a 15 cm dish and grown until they reached 90% confluency. Cells were washed with ice cold PBS, and harvested by scraping. Subsequently cells were centrifuged at 1123 g for 4 min. PBS was removed by aspiration, and cell pellet was flash-frozen. In order to lyse the cells, they were thawed and flash-frozen three times. Next, 500ul of lysis buffer, composed of 25 mM Tris pH 7, 0.5% NP-40, 1x Protease inhibitor cocktail, 2,5% murine RNAse inhibitor, 100 mM NaCl and 2 mM DTT, was added. After the addition of the lysis buffer, the cells were further resuspended using a 25 G needle 10 times. Next, two centrifugation steps ensued, first at 1500 g for 5 min, then 16,000 g for 8 min. Supernatants were transferred to a new tube, lysate concentration was measured using a BCA kit (ThermoFisher) and concentration was adjusted to 5 mg/ml using the lysate buffer. 20uM of purified G3BP1 were added to induce LLPS followed by a 40 min incubation step. Afterwards, 5uM of mCherry-tagged IGF2BP1 was spiked in, gently mixed and incubated for 20 min. The μ-Slide Angiogenesis with ibiTreat 20 ul was used as an imaging vessel. Fluorescence recovery after photobleaching (FRAP) experiments were performed on a Zeiss Axio Observer, using a Plan-Apochromat 63x/1.4 Oil DIC III objective with a Yokogawa CSU-X1 Nipkow spinning disk (50 μm pinholes, spacing 253 μm, 5000 rpm). Imaging was conducted for 3 min with 150 ms exposure time, with a time interval of 500 ms. Three pre-bleach images were taken. 70% of FRAP laser power was used, and a 40% excitation laser.

## Transfection of packaging cells

Plasmids used for transfection were purified using an endotoxin-free plasmid kit from Qiagen. Transfections were performed using 1100 ng of DNA in total, with 500 ng of plasmid of interest, 500 ng pCMVR8.74 (Addgene plasmid # 22036), and 100 ng of pCMV-VSV-G (Addgene plasmid # 8454). Supplement free DMEM was used to mix DNA and Polyethylenimine (PEI) in a 1:3 ratio. HEK293T HiEX cell were used as packaging cells. $2 \times 10^5$ cells were seeded in a 6-well plate a day before transfection and grown in a fully supplemented DMEM. The Plasmid mixture containing a transfection reagent was added dropwise onto the cells and they were incubated for 48 h.

## Transduction of U2OS (GFP-G3BP1) and HCT116 cells (ΔIGF2BP2, ΔIGF2BP3)

After a 48 h incubation, virus was collected from the supernatant with a syringe and sterile filtered. U2OS and HCT116 cells were seeded a day before in fully supplemented medium. The sterile filtered virus was mixed with fully supplemented DMED with 8ug/ml Polybrene at a 1:50 ratio. Cells were grown for 48 h up to 15 cm dishes. The BD Melody Fluorescence Activated Cell Sorting (FACS) system was used to sort U2OS cells in purity mode and gated for high and low expression. The high-expressing population was used for further experiments. HCT116 cells were sorted in a single-cell sorting mode, gated for high and low expression. Three clones were selected for further characterization. None of the cell lines are authenticated. The cell lines were not tested for Mycoplasma contamination.

## Western blotting

80% confluent cells were lysed with RIPA buffer (150 mM NaCl, 1% NP-40, 0.5% Sodium deoxycholate, 0.1% SDS, and 25 mM TRIS pH 7.4) containing 1x Protease inhibitor (Roche). The protein concentration was determined using a commercially available BCA kit (ThermoFisher). 10 μg of protein containing lysate in sample buffer was denatured at 95 °C for 5 min. Following denaturing, the samples were loaded onto the 10% sodium dodecyl sulfate (SDS) gel. Proteins were transferred onto a Nitrocellulose membrane (Amersham) in transfer

**Table 1 | Antibodies and dilutions used for Western Blot analysis**

| Antibodies and dilutions used | | | | |
|---|---|---|---|---|
| Antibody | Dilution | Manufacturer | Catalog number | Lot number |
| Anti-IGF2BP1 | 1:2500 | Proteintech | 22803-1-AP | 00018571 |
| Anti-GAPDH | 1:20000 | Proteintech | 10494-1-AP | 00113796 |
| Anti-Rabbit IgG HRP conjugate | 1:20000 | Promega | W401B | 0000573275 |
| Anti-Rabbit IgG IRdye 800CW secondary antibody | 1:20000 | LI-COR | 926-32211 | D30307-15 |

buffer (25 mM TRIS, 190 mM glycine, 20% ethanol) for 110 min at 120 V. Membranes were stained with Ponceau S and blocked in 5% milk for 1 h, or in LI-COR blocking buffer (part number: 927-60001). The primary antibody was diluted in 2.5% milk (Table 1) and incubated overnight at 4 °C, or in case of the LI-COR secondary antibody in the blocking buffer. The membrane was washed 5 times with TBST (20 mM TRIS, 150 mM NaCl, 0.1% Tween 20), and the secondary antibody was applied and incubated for 1 h, which was diluted in 2.5% milk (Table 1). LI-COR secondary antibody was diluted in the manufacturers antibody diluent (927-65001). After the incubation the membranes were washed 5 times with TBST and an enhanced chemiluminescent (ECL) horse radish peroxidase substrate (ThermoFisher) was added. In case of LI-COR secondary antibodies, after the TBST washes, the membrane was washed three more times with 1x TBS. Membranes were imaged using a BioRad Chemidock, and analyzed using the manufacturers image analysis software (Biorad Image Lab), or using LICOR Odyssey CLx fluorescence imager, and analyzed using Fiji.

## Immunofluorescence and Image analyzes

20.000 U2OS and HCT116 cells were seeded in an Ibidi μSlide 8 well dish one day before the experiment and incubated at 37 °C with 5% $CO_2$. In order to induce the stress granule formation, cells were stressed with 500 uM of Sodium arsenite (As) for 30 or 60 min. Afterwards cells were washed with PBS and fixed with ice-cold Methanol for 5 min. After the fixation the slide was washed 3 more times with PBS and incubated with blocking buffer (5% BSA in TBST) for 1 h. In the case of HCT116 cells, the primary ChromoTek RFP-Booster Alexa Fluor 568 (rb2AF568) was incubated overnight. Following the incubation, three washes with PBS ensued and the sample was imaged in PBS using a Zeiss inverse point LSM980 scanning confocal microscope. The objective used was an oil 63x Plan-Apochromat objective with 1.4 NA. The software used for microscope operation is Zeiss ZEN v3.3. The fluorophores were excited with 488 nm and 561 nm lasers respectively and image analysis was done in Fiji/ImageJ. Cell pose was used in order to get the individual cells in the image, which were saved as an ROI. Cells that touch the image border were excluded, gaussian blur was applied (σ=2), background was subtracted using a rolling ball algorithm with a radius 5, and an auto-threshold "Yen" was applied, after which the particles were analyzed, from sizes 10 pixels to infinity.

## Live cell imaging and FRAP analyzes

Live U2OS cells expressing GFP-tagged G3BP1 and mCherry-tagged IGF2BP1 were imaged using a Nikon Ti2-E, inverse microscope with a Nikon Perfect Focus System, and a Yokogawa CSU-X1-A1 Nipkow spinning disk (50μm pinholes, spacing 253 μm, 5000 rpm). CFI Plan Apo 60x/1.42 Oil, WD 0.15 mm objective was used. Cells were imaged in Ibidi 8-well μSlide (80826). 20000 cells were seeded a day before the experiment in fully supplemented DMEM, and were treated with 500 uM Sodium arsenite in Gibco Imaging DMEM (11880-028) for 60 min before imaging. Cells were incubated at 37 °C, 5% $CO_2$ during

imaging. The fluorescence recovery after photobleaching (FRAP) experiment was conducted with 100% 561 nm laser power and with 10 ms bleach time per pixel. Images were acquired with 25% 561 nm laser power 500 ms exposure time and a gain of 150. 90 timepoints were recorded during the recovery with a 10 s interval and 3 frames pre-bleach. To be able to follow the recovery, a z-stack containing 6 slices with a spacing of 0.5 μm around bleached layer was acquired at each timepoint. Image processing was conducted in Fiji/ImageJ. Respective z-stacks were combined to a single image via Z projection of the Max intensity. Then the bleached ROI and two ROIs of the similar size with stress granules that were not bleached for bleaching correction. Two ROIs within a cell without stress granules were selected to be used as background- The bleaching factor for each timepoint was calculated by dividing the mean intensity at a given timepoint by the mean intensity of timepoint zero. The background intensity was calculated by mean of both background ROIs corrected for bleaching. The fluorescence of the bleached ROIs was normalized to the background and the mean intensity of the three pre-bleach images for all timepoints after bleaching. The resulting curve was fitted using the one phase association equation (2) $Y = Y0 + (Plateau-Y0)*(1-exp(-K*x))$ in GraphPad PRISM.

### Immunoprecipitation for LC MS/MS analysis

HEK293 cells expressing spit-GFP tagged IGF2BP1 were seeded in 15 cm dishes and grown until they reached 80% confluency ($10^6$ cells). Cells were washed with ice-cold PBS, scraped, centrifuged (1123 g, 4 min) and the pellet flash frozen. On the day of the experiment, the pellets were thawed and 750ul of lysis buffer (consisting of 20 mM HEPES pH 7.4, 150 mM NaCl, 0.5 mM EDTA, 0.1% SDS, 0.5% Triton X-100, 0.2% Deoxycholate, 1x protease inhibitor cocktail, 1x phosphatase inhibitor cocktail, 10 ug/mL RNase A, 10 U/μL RNase T1). Cells were lysed by vortexing for 15 min (3 s in 2 min intervals). The lysate was centrifuged at 20.000 g for 15 min. 50 ul Chromotek GFP-Trap magnetic beads (gtma-20) were used per 15 cm dish. Lysates were incubated at 4 °C for 2 h rotating. The supernatant was removed by pipetting, while the beads were magnetized using a magnetic rack. The beads were washed five times with wash buffer 1 (20 mM HEPES pH 7.4, 500 mM NaCl, 0.5 mM EDTA, 0.1% SDS, 0.5% Triton X-100, 0.2% Deoxycholate) and five times with wash buffer 2 (20 mM HEPES pH 7.4, 500 mM NaCl, 0.5 mM EDTA. The samples were submitted on-beads for mass-spectrometry analysis at the Max Perutz Labs Mass Spectrometry Facility.

### RNA immunoprecipitation (RIP-Seq) and total transcriptome (RNA-Seq) sequencing and analysis

HCT116 cells stably expressing mCherry-tagged wild-type IGF2BP1 or S181E, Y396E, and RQ mutants (in triplicates, dishes plated and grown independently) and parental HCT116 cells were seeded in 10 cm dishes and grown until they reached 80% confluency. To induce stress cells were treated with 500 μM sodium arsenite (Sigma) for two hours before collection. PBS was added to the control cells. Cells were washed with ice-cold PBS, scraped, centrifuged (1123 g, 4 min) and the pellet flash frozen and stored at −80 °C. On the day of the experiment, the pellets were thawed and 150 μL of lysis buffer (25 mM HEPES pH 7.3, 150 mM NaCl, 5 mM MgCl₂, 0.5% NP-40, 1 mM DTT). 1x protease inhibitor cocktail (Roche), 1x phosphatase inhibitor PhosSTOP (Roche) and 1 U / μL of Murine RNase Inhibitor (New England Biolabs) were added for lysis. Cells were lysed by vortexing for 20 min (3 s with 2-minute intervals). The lysate was centrifuged at 20.000 g for 15 min. 20 μL of RPF-Trap magnetic agarose (rtma) were used per 10 cm dish. Lysates were incubated at 4 °C for 1 h rotating, and then washed three times with lysis buffer and twice with wash buffer (25 mM HEPES pH 7.3, 150 mM NaCl, 1 mM MgCl2, 0.5% NP40). Then the beads were resuspended in 150 μL of wash buffer containing 0.1% SDS and 2 mg/mL proteinase K (Ambion), and incubated at 55 °C for 30 min. The

eluate was collected and purified with an RNA cleanup kit (Zymo Research). The purified RNA samples were rRNA depleted using human riboPOOL probes (siTOOLs) according to the manufacturer's instructions. The RNA was then purified with an RNA cleanup kit (Zymo Research) and treated with 0.2 U of RNase-free DNase I (NEB) at 37 °C for 15 min, re-purified with an RNA cleanup kit (Zymo Research), and used for library preparation with NEBNext Ultra Directional RNA Library Prep Kit for Illumina (NEB).

For total transcriptome sequencing total RNA was extracted from 5 μL aliquot of clarified cell lysate (IP input) using the KingFisher Flex Purification System (Thermo) with the High-Performance RNA Isolation kit (Molecular Tools Shop, Vienna BioCenter). During the isolation RNA was treated with RNase-free DNase I (NEB). PolyA mRNA was isolated using NEBNext Poly(A) mRNA Magnetic Isolation Module (NEB) and used for library preparation with NEBNext Ultra Directional RNA Library Prep Kit for Illumina (NEB).

Both RIP- and RNA-Seq libraries were sequenced on NovaSeqX 1.5B at SingleRead 100 mode at the Vienna BioCenter NGS facility producing ~35 million reads per sample. BCL files were converted to demultiplexed fastq files with bcl2fastq v2.20.0.422. The quality of fastq files was checked with fastqc 0.11.9. The fastq files were trimmed and aligned to human genome hg38 using GENCODE annotation (release 46) with STAR v2.7.11b[99] allowing for the two mismatches. Gene counts were obtained using FeatureCounts function of the Subread package v2.0.6[100]. Total gene counts for protein coding genes were TMM normalized to calculate the counts per million values (CPM), filtered to only include genes with CPM higher than 10 in at least half of the libraries, and the differential expression analysis was performed with edgeR glmQLFTest (Generalized Linear Model Quasi-Likelihood F-test)[101]. IGF2BP1 targets were defined as genes with RIP-Seq CPMs at least 2-fold higher than RNA-Seq CPMs and 4-fold higher than the background RIP-Seq CPMs from parental cells. For the GO term analysis GOrilla service was used[102]. Sequencing data processing was done using the Life Science Compute Cluster (LiSC) of the University of Vienna.

### Reporting summary

Further i nformation on research design is available in the Nature Portfolio Reporting Summary linked to this article.

## Data availability

The source data are provided with this paper. All raw and processed sequencing data generated in this study have been deposited in the Gene Expression Omnibus database with the accession number GSE272875 (IGF2BP1 RNA-Seq and RIP-Seq data). The mass spectrometry proteomics data have been deposited to the ProteomeXchange Consortium (http://proteomecentral.proteomexchange.org) via the PRIDE partner repository[103] with the dataset identifier PXD045761 (mass spectrometry data of IGF2BP1 from HEK293T cells) and PXD056497 (mass spectrometry data of ProAlanase digested IGF2BP1 from HEK293T cells). The NMR signal assignments are deposited with BMRB ID 52567 (wild-type IGF2BP1 linker1 157-194) and 52568 (wild-type IGF2BP1 linker2 344-404) Source data are provided with this paper.

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

## Acknowledgements

We are grateful to the late Thomas Peterbauer at the Max Perutz Labs Biooptics Light Microscopy Facility for his help and support. We thank Mila Asparuhova, Gizem Celebi and Isabell Niedermoser for their technical support and help. We appreciate the support from Julia Scholz in image analysis. We thank Gijs Versteeg and his lab for their help with lentiviral transduction and providing us with the expression plasmids. We thank Kitti Csalyi and Thomas Sauer at Max Perutz Labs Biooptics FACS facility for their help. We thank Georg Kontaxis for his continuing support with NMR measurements. Proteomics analyzes were performed by the Mass Spectrometry Facility at Max Perutz Labs using the VBCF instrument pool. We are grateful to Max Perutz Labs Mass spectrometry facility, Markus Hartl, Dorothea Anrather, Wei-qiang Chen and David Hollenstein for their help and support with measurements, data analyzes and experimental design. We are grateful to Thomas Leonard for his help with SEC-SAXS measurements and his invaluable input in data analyzes. We thank Jeffrey A. Chao for providing us the MBP-tagged IGF2BP1 full-length. We thank Arthur Sedivy from the Vienna BioCenter ProTech Facility for his value support with DLS measurement and analysis. This research was funded in whole or in part by the Austrian Science Fund (FWF) [FWF-SFB F79 and FWF-W 1261] to GEK. For open access purposes, the author has applied a CC BY public copyright license to any author accepted manuscript version arising from this submission.E.S. and R.C. acknowledge support and funding by the Frankfurt Institute of Advanced Studies, the LOEWE Center for Multiscale Modeling in Life Sciences of the state of Hesse, the Collaborative Research Center 1507 "Membrane-associated Protein Assemblies, Machineries, and Super-complexes" (Project-ID Project ID 450648163 – P09), and the International Max Planck Research School on Cellular Biophysics (to R.C.), the Centre for Scientific Computing of the Goethe University and the Jülich Supercomputing Centre for computational resources and support. The research of T.M. was supported by Austrian Science Fund (FWF) grants P28854, I3792, DOC-130, and DK-MCD W1226; Austrian Research Promotion Agency (FFG) grants 864690 and 870454; the Integrative Metabolism Research Center Graz; the Austrian Infrastructure Program 2016/2017; the Styrian Government (Zukunftsfonds, doc.fund program); the City of Graz; and BioTechMed-Graz (flagship project). The GFP-G3BP1 engineered cell lines are kind gift of Witold Szaflarski, Pavel Ivanov and Paul Anderson.

## Author contributions

HH: Conceptualization. HH, ASA, AM, BB, ES and IN: Investigation, Methodology, Formal Analyzes and Visualization. TM, RC and GEK: Conceptualization, Funding acquisition and Supervision. HH and GEK wrote the original draft. All authors contributed to review and editing.

## Competing interests

The authors declare no competing interests.
