## [Transparent Peer Review file · Nature Communications]

IGF2BP1 phosphorylation in the disordered linkers regulates ribonucleoprotein condensate formation and RNA metabolism

Corresponding Author: Dr Gülsün Karagöz

Version 0:

Reviewer comments:

Reviewer #1

(Remarks to the Author)

This manuscript aims at studying the impact of phosphorylation of the RNA binding protein IGF2BP1 on condensate formation and RNA binding. The authors first are studying using mass spectrometry the phosphorylation of the protein and could find upon stress one major phosphorylation site at position S181. Although they did not detect modification of Y396, they did study this modification as it was reported earlier. Note that S181 was also reported earlier. The author then went one to study the impact of a S181E and a Y396E mutant using various assays to study their impact primarily in the context of condensate formation in vitro and in cells. They also try to study structurally using low resolution methods (SAXS, NMR perturbation) and simulation what could be the impact of the modifications.

Overall, I found the results very descriptive and speculative. No very clear conclusion can be made except that the two phosphomimics show some differences from the WT in some of the assays. Now this is not the first time that phosphomimics of RNA binding proteins have shown an effect on phase separation. I am therefore not convinced how novel are the findings here with IGF2BP1 and what generality can be gained here. No clear molecular mechanism are proposed and no assay to validate the speculations are proposed. Here many different assays are proposed but in the end, it is not clear which feature is really functionally relevant if any. Also, no control are proposed like a S181A or a Y396F which would be unphosphorylated proteins.

In my view the manuscript in this form is too descriptive and not very conclusive with very little mechanistic insight.

Reviewer #2

(Remarks to the Author)

In their study Hornegger et al. propose that phosphorylation of the mRNA binding protein IGF2BP1 modulate low affinity associations of the protein with RNA targets and thereby modulate RNP homeostasis. The studies settle on previous findings reporting phosphorylation at S181 to enhance binding to a specific IGF2 5'UTR and phosphorylation at Y396 promoting dissociation from mRNAs to allow the spatial control of localized ACTB mRNA translation. Although the data are largely solid they only provide an incremental gain of novel insights. These are largely derived by the NMR-based studies which suggest a putatively interesting concept of low complexity regions in IGF2BP1 contributing to low affinity RNA association and intramolecular associations. These findings, however, are not evaluated by complementary approaches and most importantly are not tested in cells. Thus, a major shortcoming of the presented study is that the authors largely settle on in vitro findings. They do not provide compelling evidence that the in vitro derived findings are physiologically relevant and essentially contribute to the homeostasis of IGF2BP1 in true mRNPs and thus the post-transcriptional control of mRNA fate in cells.

Major concerns:

1. The mapping of phosphorylation sites within IGF2BP1 remains incomplete, since the authors cannot confirm nor falsify phosphorylation at Y396 since this site cannot be detected by tryptic digest for MS analyses. However, the authors could attempt distinct proteolysis protocols to derive peptides allowing analyses of this phosphorylation site.
2. Another major problem of the study is the choice of RNA targets. Although this is based on CLIP data, the authors chose

candidate target mRNAs for which regulation by IGF2BP1 has not been reported. Thus, what if the interaction is unspecific. This needs to be addressed by investigating target RNAs derived from mRNAs with validated IGF2BP1-dependent regulation, e.g. MYC.

3. In line with the previous concern, the derived affinities for RNA association are about two in some cases (e.g. KH1,2) even >3 orders of magnitude lower (for short RNAs) than typically observed for specific (regulatory relevant) association, which is typically in the low (<10nM range). Did the authors exclude contamination by unspecific nucleic acids in their recombinant protein preparation? Also, how did the authors test their claim of promiscuous RNA association of RRM1,2?

4. Studies of the KH3,4 GxxG mutant are puzzling in view of previous studies where mutation or deletion of this region was proposed to have profound effects in vivo. Thus, the authors need to explain and test in further detail if this mutant is indeed binding RNA like the wildtype protein. If so, the authors need to show that an IGF2BP1-K can indeed be recovered by IGF2BP1 wild type as well as the KH3,4 mutant protein.

5. Claiming phosphorylation dependent regulations requires testing of truly phosphorylated protein. The authors do not provide any of such studies. This is of particular importance since phosphomimic mutants do necessarily reflect true phosphorylation, especially in the case of tyrosine (Y396) modification.

6. The authors claim to study RNP formation in vitro and in cellulo. This is not correct, since an IGF2BP1-RNP is likely to contain additional proteins, in particular YBX1, and is very likely totally distinct from stress granules which are formed upon prohibiting bulk mRNA translation. Along these lines, the authors need to test dynamics of the distinct mutants in situ, not just in vitro even if using HEK293 cell lysates with the addition of G3BP1. Along these lines, the authors need to address prior studies showing the dynamic (not just static effects of SG number and size, Fig3) behavior of IGF2BPs in SGs indicating that the protein has a high immobile fraction and low exchange rates. Currently, the presented findings (Fig.3) only reveal very mild effects under non-physiological conditions. Do these hold up or are these even enhanced in cellulo?

7. The most interesting part of the manuscript suggests a low-affinity RNA-binding RGG/RG motif in linker 1 and intramolecular association of linker 2 with the RRM1-2 and KH1-2 domain. Although intriguing, these findings remain premature and further testing is required to reveal if these associations truly occur in cellulo and how these may affect IGF2BP1 function. For instance, what happens in respect to target mRNA regulation upon inactivation of the RGG/RG motif or perturbing intramolecular association?

Reviewer #3

(Remarks to the Author)

In this work, the authors investigate the phase behaviors of the protein IGF2BP1, as influenced by mutations to various substrate binding motifs / modules within the protein, and the effects of phosphorylation. The title overemphasizes one set of findings, when in fact the MS has many messages. This could be remedied by a change to the title. Overall, this is a very impressive MS that was a thoroughly riveting read. Of course, no MS is perfect, and this is true here as well. A series of issues came up during the reading of the MS. For the most part, these comments can be addressed with responsive and clarifying revisions, although some measurements may be deemed essential by the authors. The comments that came up as requiring attention are inventoried below. The list is long for sure, but this, in no way, detracts from the enthusiasm for the very detailed and thorough work reported in this MS that definitely merits publication following revision.

Specific Comments

1) Are there known phosphosites in linker 2?

2) The use of phosphomimic substitutions will be seen as suggestive, but potentially problematic. Neither Glu nor Asp are chemical equivalents of pSer. This point has been made in work over the years, including that of Hilal Lashuel and colleagues. It would be useful to discuss this issue and perhaps discuss strategies to go beyond the use of Asp / Glu substitutions.

3) The narrative emphasizes "combinatorial recognition" of RNA motifs by the individual RNA binding domains. This is a term that is open to different interpretations. One possibility is as follows: There are k RNA motifs and m RNA binding domains. Combinatorial recognition could refer to there being $m! / (k! (m-k)!)$ possibilities for 1:1 complexes. There are other interpretations as well. This will be helpful to clarify semantically, conceptually, mathematically, and schematically. It is important because it impacts the interpretation of the binding isotherms, and the way phase separation data are analyzed.

4) Dissociation constants are extracted from EMSA data as well as fluorescence anisotropy data. There are, as the authors note, multiple binding motifs on the RNA molecules, and multiple RNA recognition motifs / binding domains on the protein. However, based on the methods section, which does not go into details of how the binding isotherms were analyzed, one presumes that the binding model is based on assuming a simple 1:1 model. This will be viewed as being problematic for a few reasons: Given the multiplicity of 1:1 complexes, and the fact that there are multiple binding sites, the dissociation constant is an apparent K_d given the assumption of 1:1 binding. Second, there is no information regarding the distribution of complexes that likely form in the system. Given the multivalence in the system, and what one presumes are non-equivalent microscopic affinities between different RNA motifs and specific sites on the protein, the formal expectation will be of a higher-order binding polynomial that describes multiple binding modes in a 1:1 complex, and the very real possibility of an m:n type of complex. This makes it very difficult to leap from binding to describing phase behavior, because the multivalence will become a generator of non-saturable binding, that gives rise to a network that grows with concentration, a phenomenon known as bond percolation, which likely underlies the observed phase behavior. Please see:

<https://doi.org/10.1021/acs.chemrev.2c00814> for a useful and detailed account of these phenomena. Given all this, it would help immensely, if, at a minimum, the dissociation constants are prefaced as being apparent K_d s, all extracted using a specific binding model, and the binding model for analyzing EMSA data and anisotropy data are spelled out in detail. These

details are important, because a two-fold increase in apparent affinity could have, due to combinatorial considerations, an influence on the growths of networks as espoused in the published literature: <https://www.nature.com/articles/s41467-020-15395-6>. Furthermore, the binding isotherms might need reanalyzing using different models that help extract the apparent site size and how this is affected by phosphorylation. The treatise by Wyman and Gill and many others, including the book of Doug Barrick provides several useful ways of thinking about the problem.

5) On line 189, the authors state that many RNPs form through LLPS. No references are provided in support of this assertion, and the specific concerns apply to the choice of LLPS as the phenomenon when quite a bit of work over the past few years has clearly shown that the prefix LL is likely misleading. The phenomena combine continuous associative phase transitions and first order segregative phase transitions, and does not involve the type of separation evident in aqueous two-phase separation or the separation of oil from water. Semantics are important because they impart rigor to the process being described. Phase separation is driven by solubility considerations, percolation by multivalence. The two are coupled in macromolecules.

6) On line 197, the authors describe data pertaining to the abrogation of phase separation in the presence of excess RNA. These data are not to be found in the MS. Does the abrogation pertain to reentrant behavior tied to an imbalance of charge, as proposed by Banerjee and colleagues (<https://onlinelibrary.wiley.com/doi/abs/10.1002/anie.201703191> and <https://pubs.acs.org/doi/abs/10.1021/jacs.9b03689>) or alternatively, is this a manifestation of a pure stoichiometry imbalance seen by Rosen and coworkers because the phase behavior is driven purely by heterotypic interactions? To answer the latter question, it will be important to reanalyze the data using a formalism where the axes are not molecular concentrations, but instead are concentrations of complementary motifs. This, as shown recently, will help with identifying whether the phase behavior is driven by purely heterotypic interactions and also help uncover the balance of heterotypic and homotypic interactions (<https://www.nature.com/articles/s41467-023-43489-4>). In its current form, the analysis in favor of purely heterotypic interactions, especially given what follows, will have to be seen as being incomplete.

7) On line 202, the authors pivot to describing data for binding to model RNAs. It would help to motivate this pivot and explain the specific choices made.

8) A puzzler for many is the "avidity effect". This is invoked by the authors, but a quantitative analysis of this effect is missing. The challenge lies in defining and analyzing avidity. Does this refer to incomplete dissociation because of the spatial proximity of equivalent or competing binding sites? Does it refer to a mean-field contribution from the high local / effective concentration of binding sites? Does it refer to positive cooperativity? Or does it refer to the convoluted descriptions one sees in the literature where avidity is described via a combination of kinetics and thermodynamics? This is important in the current context because the authors lean fairly heavily on this concept for interpreting some of their results.

9) The data in Figures 2E and 2F are compared in terms of molar equivalents of concentration. However, the molar concentrations of RNA motifs are different. Additionally, there are clear differences in the concentration / stoichiometry dependencies between the two scenarios. These differences, if analyzed quantitatively, can be used to paint a clearer, thermodynamic interpretation of the contribution of different domains and motifs to the overall phase behavior. This would be very helpful. The comment is germane given the conclusion regarding necessity and sufficiency. This conclusion leaves one wondering about the contributions from regions outside KH3-4. It is also relevant, given the somewhat puzzling inference drawn regarding the GEEG mutant in the full-length protein. The authors argue that 185.8 nM is of high affinity. However, this is not true when compared to nearly 5-fold or higher difference when compared to the WT protein. These small differences can have multiplicative effects when thinking about the impact on percolation thresholds, the relevant parameter when thinking about the contributions of multivalent interactions. So, the conclusions drawn on lines 239-242 are puzzling. Please also note that it is very difficult to draw the types of definitive inferences made in the MS without seeing a full phase boundary.

10) There is a remarkable result, which is only emphasized in passing in the MS. This pertains to the data in Fig. Supp 4E vs. Fig. Supp 4F. The authors finding it remarkable, which it indeed is, because this shows very clearly the separation between percolation sans phase separation and the contribution of regions outside KH1-4 to the coupling of percolation and phase separation that gives rise to condensate formation. How this happens will be very interesting to investigate, although one would stipulate, rightly so, that it is beyond the scope of the current MS. However, this is a really important finding. The truncated version drives the formation of a system-spanning network, referred to as a meshwork. In contrast, the full length protein undergoes percolation and phase separation, giving rise to, presumably, a condensate-spanning network. These results are hard to come by in the literature and given their utmost importance in clarifying the nature of the phenomenology, it would be useful to move these data to the main text, and to emphasize the nature of what is being observed using rigorous and conceptually accurate terminology.

11) The driving forces for condensate formation are analyzed mainly in terms of the sizes of condensates. It would be more comforting to have the threshold solubility product (<https://elifesciences.org/articles/67176>) i.e., threshold joint concentration of protein AND RNA required for phase separation for the unphosphorylated protein and the phosphomimic. Absent this, one does not know what to make of the quantified differences in sizes of condensates, because one does not know the magnitude of the quench depth into the two-phase regime. Knowledge of the threshold product, helps with the analysis whereby one can then, if one wishes, quantify condensate sizes at equivalent concentrations, which may or may not pertain to different degrees of supersaturation, or quantify condensate sizes at equivalent degrees of supersaturation if the thresholds are different.

12) An important aspect that requires quantification is the presence of clusters in sub-saturated solutions (defined as

concentrations below the threshold concentration - joint of course - for phase separation). Recent studies have shown that such clusters exist for proteins alone (<https://www.pnas.org/doi/full/10.1073/pnas.220222119> and <https://www.nature.com/articles/s41467-023-40540-2>) and for protein-RNA mixtures (<https://www.pnas.org/doi/10.1073/pnas.2120799119>). One would expect that these clusters, referred to by some as pre-percolation clusters, will likely influence the driving forces for condensation, and explain the abrogation of phase separation studied under identical conditions / concentrations. At a minimum, it would help to acknowledge this possibility in the discussion as a direction to pursue in future work.

13) Kinetics of phase separation are difficult to compare across constructs if the concentrations are kept the same, and the driving forces, i.e., threshold concentrations are different. In such cases, it becomes important to measure kinetics at equivalent degrees of supersaturation, because this is the only way a molecular perspective emerges regarding the differences in mechanisms. Otherwise, the kinetics data become qualitative because they are a convolution of somewhat trivial concentration-dependent effects, that are clearly unmasked using equivalent degrees of supersaturation, and the heights of barriers that are system-specific, which one wishes to unmask. The request is for these data to be de-emphasized and moved to the SI because their interpretation requires very different approaches and measurements.

14) Does FRAP measure fluidity of condensates? This is a complex issue because what it measures is a convolution of the contributions from physical crosslinking and hence the lifetimes of "bonds" formed between molecules, and the contributions of molecular transport (typically viewed as diffusion). Condensates are viscoelastic materials, and therefore FRAP measures convolutions of the contributions from viscous and elastic moduli. Despite its widespread usage, FRAP is not a readout of fluidity. The critique here is a semantic one, and the request is to delete or deemphasize the assertion that FRAP measures fluidity. The data showing how G3BP1 influences FRAP dynamics and the fractions of mobile vs. immobile species are very elegant and compelling.

15) The back end of the MS makes a clear case for the presence of homotypic interactions among the IGF2BP1 molecules. There is precedent for substantial changes in charge, via multi-site phosphorylation not impacting measured SAXS profiles even though there are local and non-local changes to interaction profiles that are mutually compensatory. Please see: <http://pubs.acs.org/doi/abs/10.1021/jacs.6b10272>. Might there be similar effects in play here? The analysis of the simulations would benefit from the inclusion of pair distribution functions that quantify the effective associations among the molecules represented in the simulations. Please note that linkers can contribute in two ways. Through the effective solvation volumes for sure, but also through interactions in cis and hence in trans as well with the substrate binding domains. This point, which the authors make very clearly and elegantly, was first reported in 2015 using very similar methodologies. Please see: <http://www.pnas.org/content/112/47/E6426.long>. Such effects have been reported for linkers in multidomain proteins, whereby even well-solvated linkers, when modeled as autonomous units, can have complementary interactions with folded domains. Please see: <https://www.sciencedirect.com/science/article/pii/S0022283618304078> and <https://doi.org/10.1016/j.crstbi.2021.08.002>. Such effects will influence the interplay of homotypic and heterotypic interactions as drivers of phase separation. Given the systematic probing of the effects of linker-domain interactions, there ends up being a bit of cognitive dissonance between the inferences made in the front end vs. the back end of the MS, which are left unreconciled in the discussion. It would help to remedy this, and also highlight synergies with other systems in the literature, including observations made by three different groups for the phase behaviors of G3BP1/2. In this context, one would recommend that the speculations starting on line 538 be deleted.

16) Finally, the impressive corpus of data could be bolstered by a more expansive and inclusive discussion of many foundational papers, outside the IGF2BP1 literature, on which this work is built. The field of condensates could benefit from knowing about the current work, and the converse is true for the IGF2BP1 field as well. It is a testament to the maturing nature of the condensate field that contributions in this field are now cited as axioms sans citation. However, from a scholarship perspective, this is less than ideal or appropriate. This is easily remedied by including citations, and clarifying some of the verbiage to be in line with the emerging and established concepts in the condensate field.

Reviewer #4

(Remarks to the Author)

The manuscript by Hornegger et al. describes the implementation of a huge range of approaches to investigate the phosphorylation status of the protein IGF2BP1 under stress conditions, and determine how two of these phosphorylation modifications affects its ability to bind RNA and form condensates. The authors used mass spectrometry to identify the phosphorylation sites, and use phosphomimetic mutants S181E and Y396E to identify whether these mutations affect the RNA binding capacity of the full length protein, and identify no effects. It is shown that IGF2BP1 + RNA form RNP granules in vitro, and that this is decreased in the S181E mutant and increased in the Y396E mutant. The authors then used a very neat approach to study mcherry-tagged IGF2BP1 condensates in a cellular environment by adding recombinant G3BP1 into a whole cell lysate. Here they found that the S181E mutant resulted in more stable condensates than the WT, and that the condensates formed by Y396E were more dynamic. This also held true in cells.

The authors then went on to carry out some structural studies on IGF2BP1. SAXS revealed that the WT protein and the phosphomimetic mutants have an overall similar conformational ensemble. MD simulations suggested that the isolated linkers do not interact, and this was experimentally confirmed by NMR. Additional specific interactions were obtained by NMR, such as linker 1 binding to RNA and linker 2 binding to folded domains of IGF2BP1.

The manuscript contains a huge amount of information that is extremely difficult to digest, even after reading it many times. However, I believe that the authors have successfully identified phosphorylation states that occur under stress, identified

how these phosphorylation states affect the macroscopic properties of RNPs and delineated the structural basis that underlie these RNP properties. The work is original, and the conclusion are well supported by the results. The methodology is sound and contains enough detail for reproduction of the experiments.

Overall, I believe that the work is of high enough quality and originality to warrant publication in Nature Communications, but the narrative could be greatly improved to increase readability and hence interest from readers. The text could be more succinct. As a suggestion, for each section it would be easier to read if the main results were described first, for quick understanding, and the detailed SI data were described after for those that are interested. The use of many methods to support the same findings is of course scientifically strong, but challenging to understand for the reader. I appreciate it's a difficult thing to balance.

Minor points-

Paragraph beginning at page 3 line 122. Please understand that you're comparing MS intensity of the phosphorylated vs. non-phosphorylated peptides, which may not directly correlate to protein amounts. Therefore please change all mentions of changes in amount of phosphorylation to changes in signal corresponding to phosphorylated peptides. 64% of signal intensity is unlikely to equate to 64% of protein molecules due to changes in ionisation efficiency when adding a negative charge to the peptide.

Please provide the sequences of all recombinant proteins used in the study, including those of the isolated domains.

Version 1:

Reviewer comments:

Reviewer #1

(Remarks to the Author)

The authors added new data to make the study more functionally relevant. I am happy with the manuscript in this form.

Reviewer #2

(Remarks to the Author)

The revised study by Hornegger et al. addressed most of the previously raised concerns raised by referees. In sum, this supports the previous notion that the IDRs of IGF2BP1 and their post-translational modification has a severe impact on modulating the properties of IGF2BP1-RNA association and mRNP formation under normal as well as stress conditions. Most notably, the authors for the first time – to the best of this referee's knowledge – provide evidence by MS that Y396 is phosphorylated by SRC, at least in vitro. However, the authors need to indicate this in the main text more precisely. Currently, the reader is left guessing why the authors continue with this modification although it is not affected by cellular stress nor is strikingly abundant at steady state in non-stressed cells.

The bulk of mostly concise data provided by the authors is beneficial and this referee appreciates the author's efforts in addressing the various concerns by various new or substantially revised investigations. However, the presentation of findings is quite extensive and hard to digest. Thus, extensive editing of the manuscript (not more data) are required to support readers in following conclusions and key findings. One remaining conceptual weakness of the study, however, is that despite the biophysical insights provided, it remains vague if and how PTMs in IDRs ultimately affect physiological regulation – not the binding (!) - of target mRNAs by IGF2BP1 and consequently gene expression. This must be addressed in view of the title and abstract claiming that PTMS in IDRs RNA metabolism, which at present is not addressed beyond binding. In sum, the revised manuscript provides an extensive work on biophysical properties of PTMs, IDRs and folded RNA-binding domains but partially lacks the extraction of key insights on how these ultimately affect target mRNA fate under normal and or stressed conditions. These aspects need to be addressed by revision.

Reviewer #3

(Remarks to the Author)

The authors have, in my view, set a new standard for being responsive in terms of the revisions they have made, their overall scholarship, and the new data / analyses they have included in the revised MS. These types of responses make a detailed review and the investment of time well worth the effort. I have no further revisions to suggest or requests to make. This MS deserves to be published in its current form, and it will be viewed as a very important contribution to the field of phase separation and beyond.

Reviewer #4

(Remarks to the Author)

The authors have done a great job at restructuring the manuscript and making it easier to follow.

REVIEWER COMMENTS

Reviewer #1 (Remarks to the Author):

This manuscript aims at studying the impact of phosphorylation of the RNA binding protein IGF2BP1 on condensate formation and RNA binding. The authors first are studying using mass spectrometry the phosphorylation of the protein and could find upon stress one major phosphorylation site at position S181. Although they did not detect modification of Y396, they did study this modification as it was reported earlier. Note that S181 was also reported earlier. The author then went one to study the impact of a S181E and a Y396E mutant using various assays to study their impact primarily in the context of condensate formation in vitro and in cells. They also try to study structurally using low resolution methods (SAXS, NMR perturbation) and simulation what could be the impact of the modifications.

Overall, I found the results very descriptive and speculative. No very clear conclusion can be made except that the two phosphomimics show some differences from the WT in some of the assays. Now this is not the first time that phosphomimics of RNA binding proteins have shown an effect on phase separation. I am therefore not convinced how novel are the findings here with IGF2BP1 and what generality can be gained here. No clear molecular mechanism are proposed and no assay to validate the speculations are proposed. Here many different assays are proposed but in the end, it is not clear which feature is really functionally relevant if any. Also, no control are proposed like a S181A or a Y396F which would be unphosphorylated proteins.

In my view the manuscript in this form is too descriptive and not very conclusive with very little mechanistic insight.

Answer:

We thank the reviewer for disclosing their point of view. IGF2BP1 regulates several RNAs essential for early development, moreover its deregulation contributes to tumorigenesis in several cancers. While phosphorylation was proposed to regulate IGF2BP1's interaction with select RNAs, we lacked the underlying mechanism. Our data shed light on the mechanistic basis of how the two best-described phosphorylation events regulate IGF2BP1 function. We show that phosphomimetic mutants impact RNP formation both in vitro and in cells. Moreover, during the revision process, we also obtained data on the physiological significance of these phosphorylation events using transcriptomics and RNA-immunoprecipitation (RIP) coupled to deep sequencing in cells expressing wild-type IGF2BP1 and the phosphomimetic mutants. Using RIP-Seq, we found that phosphomimetic mutants show differences in their binding affinity to several RNAs. Moreover, the expression of IGF2BP1 phosphomimetic mutants in cells largely remodels transcriptome compared to the cells expressing the wild-type IGF2BP1, underlining the physiological significance of our findings and the regulatory potential of the linkers by phosphorylation in cells. These findings are a valuable contribution to the fields of posttranscriptional regulation by RNA-binding proteins and IGF2BPs thus, we are confident that the community will benefit from its publication.

Reviewer #2 (Remarks to the Author):

In their study Hornegger et al. propose that phosphorylation of the mRNA binding protein IGF2BP1 modulate low affinity associations of the protein with RNA targets and thereby modulate RNP homeostasis. The studies settle on previous findings reporting phosphorylation at S181 to enhance binding to a specific IGF2 5'UTR and phosphorylation at Y396, promoting dissociation from mRNAs to allow the spatial control of localized ACTB mRNA translation. Although the data are largely solid they only provide an incremental gain of novel insights. These are largely derived by the NMR-based studies which suggest a putatively interesting concept of low complexity regions in IGF2BP1 contributing to low affinity RNA association and intramolecular associations. These findings, however, are not evaluated by complementary approaches and most importantly are not tested in cells. Thus, a major shortcoming of the presented study is that the authors largely settle on in vitro findings. They do not provide compelling evidence that the in vitro derived findings are physiologically relevant and essentially contribute to the homeostasis of IGF2BP1 in true mRNPs and thus the post-transcriptional control of mRNA fate in cells.

Major concerns:

1. The mapping of phosphorylation sites within IGF2BP1 remains incomplete, since the authors cannot confirm nor falsify phosphorylation at Y396 since this site cannot be detected by tryptic digest for MS analyses. However, the authors could attempt distinct proteolysis protocols to derive peptides allowing analyses of this phosphorylation site.

Answer:

We thank the reviewer for the input. We agree that getting information on the phosphorylation status of the Y396 site in cells is invaluable. To address this point, based on the recommendation from the experts in our Mass Spectrometry facility, we tried another peptidase Pro-Alanase to increase coverage in linker 2. Our data showed that we could successfully generate peptides from this region that are compatible with the MS analyses and have a high coverage of this region (**Supp. File1**). To assess whether we could detect phosphorylated IGF2BP1, we pretreated purified IGF2BP1 with the Src kinase in vitro and performed analyses. Those data revealed that we can readily detect the in vitro phosphorylated peptides mapping to the Y396 site. In cells, 95% of signal from the peptides covering the Y396 site were in the unphosphorylated state under control conditions and during arsenite and ER stress. While we cannot exclude the possibility that this site is phosphorylated in a sub-stoichiometric manner and it is under the detection limit of our approaches, it is clear that this would be a minor phosphorylation event that shall not have a significant impact on the overall behavior of the protein under those conditions. Please see **Fig. Supp. 1B,C and materials and methods**.

2. Another major problem of the study is the choice of RNA targets. Although this is based on CLIP data, the authors chose candidate target mRNAs for which regulation by IGF2BP1 has not been reported. Thus, what if the interaction is unspecific. This needs to be addressed by investigating target RNAs derived from mRNAs with validated IGF2BP1-dependent regulation, e.g. MYC.

Answer:

We are grateful for your thoughtful input on this topic. To address this concern, in addition to the XBP1 and EIF2A RNA, we now used two more RNAs derived from the well-characterized IGF2BP1 targets ACTB and MYC (Wächter 2013, Bernstein 1992) (please see **Fig. Supp. 2 G-J, Fig. Supp. 4F-N**). Our new data supports our earlier findings and demonstrates that the effects of the phosphomimetic mutants (S181E and Y396E) on IGF2BP1 phase separation are independent of the type of RNA used in the assays.

3. In line with the previous concern, the derived affinities for RNA association are about two in some cases (e.g. KH1,2) even >3 orders of magnitude lower (for short RNAs) than typically observed for specific (regulatory relevant) association, which is typically in the low (<10nM range). Did the authors exclude contamination by unspecific nucleic acids in their recombinant protein preparation? Also, how did the authors test their claim of promiscuous RNA association of RRM1,2?

Answer:

The RRM12 domains of the IGF2BP3 paralogue have been shown to have low specificity and low affinity for RNAs (Jia, Gut, Chao, 2018). Human IGF2BP1 and IGF2BP3 are highly conserved in their RNA-binding domains (73.08 % sequence identity). Therefore, as expected, the affinities of the IGF2BP1 RRM12 domains for the select model target RNAs we measured here are in a similar range (>10 μ M) as the published data for IGF2BP3 (please see Fig **Supp. 3D, E, Table 2**). Avoiding RNA contaminations during the protein purification was also a serious concern for us. Thank you for highlighting this critical possible issue. We use a Heparin-affinity step during our purification, which enables us to remove bound RNAs after the first HisTRAP affinity purification step. Using Heparin-affinity columns is a common strategy used to eliminate RNA or nucleotide contamination (Gao 2021). We also make sure that the purified protein has a 260/280 nm ratio of < 0.6, demonstrating the absence of RNA contamination in the purified protein; please see **Figure 1** below.

Figure 1 The absorption spectrum of the purified IGF2BP1 (left). The SDS-PAGE of purified IGF2BP1 stained with Coomassie blue (left middle). The absorption spectrum of the purified IGF2BP1 KH 1-2 (right middle). The SDS-PAGE of purified IGF2BP1 KH1-2 stained with Coomassie blue (right).

4. Studies of the KH3,4 GxxG mutant are puzzling in view of previous studies where mutation or deletion of this region was proposed to have profound effects in vivo. Thus, the authors need to explain and test in further detail if this mutant is indeed binding RNA like the wildtype protein. If so, the authors need to show that an IGF2BP1-K can indeed be recovered by IGF2BP1 wild type as well as the KH3,4 mutant protein.

Answer:

We greatly appreciate your valuable feedback on this matter. Our data is in line with the published results, which showed that the introduction of a KH34 GxxG double mutant can have relatively mild effects on the RNA affinity of the full-length protein depending on the type of RNA (Wächter et al. 2013). Using EMSA assays, Wächter et al. (2013) showed that while mutating the GxxG motif of both KH3 and KH4 reduces the affinity of IGF2BP1 to the ACTB-zipcode by one order of magnitude, this mutant reduced the affinity of IGF2BP1 for the MYC-CRD only 2.5 fold. This indicates that the effect of the mutations is sequence-specific. Supporting this notion, Schneider et al. (2019) demonstrated a similar effect for the paralog IGF2BP3. In our hands, The KH3-4 GxxG double mutant bound strongly to a model RNA with a Kd of 33.4 nM. We hypothesize that the high apparent Kd of this mutant is due to multidomain engagement of the RRM12 and KH12 with RNAs containing multiple IGF2BP1 binding sites, which can compensate for the loss of protein-RNA interactions mediated by the KH34 domains. Wächter et al. 2013 also showed that KH34 mutants are impaired in the formation of RNP granules in cells, indicating that while KH34 GxxG double mutant can bind RNA, it is impaired in condensate formation in line with our findings here.

5. Claiming phosphorylation dependent regulations requires testing of truly phosphorylated protein. The authors do not provide any of such studies. This is of particular importance since phosphomimetic mutants do necessarily reflect true phosphorylation, especially in the case of tyrosine (Y396) modification.

Answer:

Thank you for emphasizing the importance of this issue. We agree that using phosphomimetic mutants do not fully recapitulate the physiochemical properties of phosphorylation of the residues. Generating proteins with specific phosphorylation patterns are highly challenging, especially in the amounts required for the thorough characterization and the assays used here. Notably, while the kinase phosphorylating the Y396 is well-established, currently, we lack such information for the S181 site. Thus, we applied the glutamate mutations to mimic the phosphorylation of the linkers, which is a commonly used strategy. We are very much aware of its shortcomings, and to make it clear to the readers, we have now added a clarification of this in the discussion and in the results; please see lines 894-899. Notably, the mutants clearly impact IGF2BP1 RNP granule formation in vitro and in cells, underlining the significance of these regions for IGF2BP1 regulation.

6. The authors claim to study RNP formation in vitro and cellulo. This is not correct, since an IGF2BP1-RNP is likely to contain additional proteins, in particular YBX1, and is very likely totally distinct from stress granules which are formed upon prohibiting bulk mRNA translation. Along these lines, the authors need to test dynamics of the distinct mutants in situ, not just in vitro even if using HEK293 cell lysates with the addition of G3BP1. Along these lines, the authors need to address prior studies showing the dynamic (not just static effects of SG number and size, Fig3) behavior of IGF2BPs in SGs indicating that the protein has a high immobile fraction and low exchange rates. Currently, the presented findings (Fig.3) only reveal very mild effects under non-physiological conditions. Do these hold up or are these even enhanced in cellulo?

Answer:

We greatly appreciate your valuable feedback on this matter. As the referee brought up, the literature suggests that IGF2BP1 is part of various RNP granules in cells. While some of these granules are microns size, such as stress granules, others are too small to be observed by confocal microscopy approaches. Due to this challenge, we mainly focused on

studying IGF2BP1 and its mutants in stress granules to enable us to quantify their size and biochemical properties by FRAP methods. Consistent with the published results, upon arsenite-induced stress, mCherry-tagged IGF2BP1 localized to stress granules marked by G3BP1 in U2OS and HCT116 cells. We agree with the referee that the composition of condensates might impact the outcome. Therefore, we next wanted to test IGF2BP1 properties in P-bodies. While IGF2BP1 was suggested to be part of P-bodies under steady-state conditions, IGF2BP1 showed an almost completely diffuse signal, and we did not observe colocalization with the P-body marker EDC4 under control conditions and under stress (please see **Figure 2** below). Due to those technical reasons, we focused on characterizing IGF2BP1's properties in stress granules. Based on the referee's suggestion, we now performed fluorescence recovery after photobleaching (FRAP) experiments on IGF2BP1 in cells. In agreement with the published results, IGF2BP1 was less dynamic with longer recovery half times (IGF2BP1 wild-type FRAP recovery half time= 255 sec) in stress granules compared to G3BP1 (FRAP recovery half time= ~3-20 sec, Bley et al. 2015, Niewidok et al. 2018, Yang et al. 2020).

Interestingly, the FRAP curves of the mCherry-IGF2BP1 and its mutants showed that the phosphomimetic S181E mutation impacts IGF2BP1 dynamics similarly in cells (dynamic populations wild type: 77.9%, S181E: 61.3%, Y396E: 69.4 %) as in the reconstituted stress granules in **Fig.3C** (dynamic population: WT: 68.9%, S181E: 62.2%, Y396E 74.1%). In contrast, we see an opposite trend for the Y396E mutant in cells. While reconstituted SGs in lysates were shown to have a similar composition as in cells, the stoichiometry of the components is different due to the addition of recombinant G3BP1. As brought up by the referee, these data underline that stoichiometric and compositional differences in the RNP condensates influence the effect of the phosphomimetic mutants; we now discussed this in the text; please see the lines 473-487.

Figure 2 – Z-projection of immunofluorescence images of fixed HCT116 cells expressing mCherry-IGF2BP1 untreated (top row) and after 1h with 500 µM arsenite treatment (bottom row).

7. The most interesting part of the manuscript suggests a low-affinity RNA-binding RGG/RG motif in linker 1 and intramolecular association of linker 2 with the RRM1-2 and KH1-2 domain. Although intriguing, these findings remain premature and further testing is required to reveal if these associations truly occur in cellulo and how these may affect IGF2BP1 function. For instance, what happens in respect to target mRNA regulation upon inactivation of the RGG/RG motif or perturbing intramolecular association?

Answer:

Thank you for bringing up this crucial aspect. To investigate the functional role of the low-affinity interactions of the linker 1 with RNA through the RGG/RG motif, we now generated an IGF2BP1 mutant, in which the crucial arginine residues in linker 1 are mutated to glutamine, which we called RQ mutant. We confirmed that the linker 1 RQ mutant completely abolishes the interaction of linker 1 alone with model RNAs by NMR spectroscopy approaches (**Fig. 5 I, J**). We also found that the full-length IGF2BP1 RQ mutant bound to RNAs similarly to wild-type IGF2BP1, consistent with our findings that folded RNA-binding domains mediate RNA interactions in IGF2BP1 (**Fig. 5H**). To decipher the possible physiological function of linker1-RNA interactions, we next tested whether IGF2BP1 RQ mutant impacts IGF2BP1's interaction with RNAs. To this end, we generated HCT116 cells expressing the mCherry-IGF2BP1 RQ mutant using lentiviral transduction (**Fig. Supp. 10A,B**). The RNA-immunoprecipitation (RIP-Seq) analyses of mCherry-IGF2BP1 wild-type and the RQ mutant revealed that there are distinct differences in the RNA-binding strength of the RQ mutant in comparison to the wild-type IGF2BP1. To further study the impact of this mutation on RNA metabolism and, thereby, its role in RNA regulation, we performed global transcriptomics analyses in cells expressing IGF2BP1 wild-type or the RQ mutant. Our results revealed distinct differences in the RNA levels of several IGF2BP1 targets, validating the functional importance of this region for controlling IGF2BP1-driven RNA regulation (**Fig. 6A, B**).

Reviewer #3 (Remarks to the Author):

In this work, the authors investigate the phase behaviors of the protein IGF2BP1, as influenced by mutations to various substrate binding motifs / modules within the protein, and the effects of phosphorylation. The title overemphasizes one set of findings, when in fact the MS has many messages. This could be remedied by a change to the title. Overall, this is a very impressive MS that was a thoroughly riveting read. Of course, no MS is perfect, and this is true here as well. A series of issues came up during the reading of the MS. For the most part, these comments can be addressed with responsive and clarifying revisions, although some measurements may be deemed essential by the authors. The comments that came up as requiring attention are inventoried below. The list is long for sure, but this, in no way, detracts from the enthusiasm for the very detailed and thorough work reported in this MS that definitely merits publication following revision.

Specific Comments

1) Are there known phosphosites in linker 2?

Answer:

There is no data on human IGF2BP1 that shows phosphorylation in the linker 2 region other than Y396 (phosphosite.org). This might be due to a lack of coverage of this area using common tryptic digestion methods that are used in high-throughput proteomics studies. To

overcome this shortcoming, we now used a different peptidase (ProAlanase) to generate MS-compatible peptides from linker 2. We were successful in generating MS-compatible peptides from linker 1 with around 90% coverage of this area using in vitro purified IGF2BP1 and IGF2BP1 purified from mammalian cells (**Supp. File 1**). Our MS analyses in cells showed that the peptides mapping to the linker 2 are mainly (> 95% total protein) non-phosphorylated in HEK293 cells under control and proteotoxic stress (**please see Supp. Fig. 1B,C**).

Based on the suggestion of the referee, we modified the title to better emphasize the role of the linkers in regulating RNP condensate formation, please see the title below:

“IGF2BP1 phosphorylation in the disordered linkers regulates ribonucleoprotein condensate formation and RNA metabolism”

2) The use of phosphomimic substitutions will be seen as suggestive, but potentially problematic. Neither Glu nor Asp are chemical equivalents of pSer. This point has been made in work over the years, including that of Hilal Lashuel and colleagues. It would be useful to discuss this issue and perhaps discuss strategies to go beyond the use of Asp / Glu substitutions.

Answer:

We value your insight on this significant matter. Our goal was to characterize the effect of phosphorylation by applying biophysical and biochemical methods. While phosphomimic substitutions do not fully recapitulate the effect of phosphorylation, they are still commonly used in a variety of different approaches. The main reason behind this is that it is challenging to generate proteins with individual, specific modifications at amounts required for characterization. Genetic code expansion has been employed to site specifically generate proteins phosphorylated at distinct serine residues in bacteria and in mammalian cells. However, it is highly challenging to implement these methods to yield proteins phosphorylated in stoichiometric high levels for in vitro experiments. Moreover, it is also challenging to ensure comparable expression levels of the constructs carrying amber stop codons as the wild-type counterparts (Beranek et al., 2018, [10.1016/j.chembiol.2018.05.013](https://doi.org/10.1016/j.chembiol.2018.05.013)). As an alternative in a trial experiment, we used recombinant Src kinase to phosphorylate IGF2BP1. Our data showed that the Src kinase phosphorylated Y396 in IGF2BP1 in vitro. However, in addition to this site, we observed phosphorylation of Y206, impeding any potential experiments that could overcome the limitations of phosphomimetic mutations. Thus, we decided to apply the glutamate mutations to mimic phosphorylation, knowing its shortcomings. We added clarification of this to the discussion.

3) The narrative emphasizes "combinatorial recognition" of RNA motifs by the individual RNA binding domains. This is a term that is open to different interpretations. One possibility is as follows: There are k RNA motifs and m RNA binding domains. Combinatorial recognition could refer to there being $m! / (k! (m-k)!)$ possibilities for 1:1 complexes. There are other interpretations as well. This will be helpful to clarify semantically, conceptually, mathematically, and schematically. It is important because it impacts the interpretation of the binding isotherms, and the way phase separation data are analyzed.

Answer:

Thank you for bringing this critical issue to our attention. Due to technical and biological limitations, we cannot draw conclusions that would allow us to deduce the number of possible protein-RNA interactions in our assays. The “combinatorial recognition” has been used in a recent paper (Schneider 2019) in which the authors show that all RNA-binding

domains contribute to the interaction of IGF2BP1 with specific RNA motifs. The avidity effect we observe in the assays using full-length protein compared to the individual domains clearly shows that there is combinatorial binding occurring in our system. In our assays, we used RNAs that contain two predicted binding sites by IGF2BP1s. However, due to the highly promiscuous recognition of RNAs by both KH and RRM domains, we can not exclude the possibility of other interactions in the multidomain protein. We have now tried to clarify this in the text; please see lines 261-277 and **Supp. Fig. 3O**.

4) Dissociation constants are extracted from EMSA data as well as fluorescence anisotropy data. There are, as the authors note, multiple binding motifs on the RNA molecules, and multiple RNA recognition motifs / binding domains on the protein. However, based on the methods section, which does not go into details of how the binding isotherms were analyzed, one presumes that the binding model is based on assuming a simple 1:1 model. This will be viewed as being problematic for a few reasons: Given the multiplicity of 1:1 complexes, and the fact that there are multiple binding sites, the dissociation constant is an apparent K_d given the assumption of 1:1 binding. Second, there is no information regarding the distribution of complexes that likely form in the system. Given the multivalence in the system, and what one presumes are non-equivalent microscopic affinities between different RNA motifs and specific sites on the protein, the formal expectation will be of a higher-order binding polynomial that describes multiple binding modes in a 1:1 complex, and the very real possibility of an m:n type of complex. This makes it very difficult to leap from binding to describing phase behavior, because the multivalence will become a generator of non-saturable binding, that gives rise to a network that grows with concentration, a phenomenon known as bond percolation, which likely underlies the observed phase behavior. Please see: <https://doi.org/10.1021/acs.chemrev.2c00814> for a useful and detailed account of these phenomena. Given all this, it would help immensely, if, at a minimum, the dissociation constants are prefaced as being apparent K_d s, all extracted using a specific binding model, and the binding model for analyzing EMSA data and anisotropy data are spelled out in detail. These details are important, because a two-fold increase in apparent affinity could have, due to combinatorial considerations, an influence on the growths of networks as espoused in the published literature: <https://www.nature.com/articles/s41467-020-15395-6>. Furthermore, the binding isotherms might need reanalyzing using different models that help extract the apparent site size and how this is affected by phosphorylation. The treatise by Wyman and Gill and many others, including the book of Doug Barrick provides several useful ways of thinking about the problem.

Answer:

We are grateful for your thoughtful input on this topic; it is indeed very important. We have now clarified the binding model in the results section and the materials and methods. To calculate the affinities, we used the Hill equation. We agree that our binding assays EMSA and fluorescence anisotropy approach only yield apparent K_D s using the Hill equation; we now clarified this in the text and referred to this as " $K_{1/2}$." In the concentration regimes we worked on the fluorescence anisotropy experiments, we only see a single transition therefore it is not plausible to extract different binding models from this data. While EMSA assays would allow us to study different binding events, drawing conclusions about possible small differences is challenging due to the technical difficulties in quantifying the bands in EMSA assays. However, our data clearly show that wild-type IGF2BP1 and the mutants bind to model RNAs similarly (**Fig2 A-D, Fig. Supp. 2C-J**). We considered using ITC measurements as an alternative method, but due to phase separation, we could not find a concentration regime that is technically feasible.

5) On line 189, the authors state that many RNPs form through LLPS. No references are provided in support of this assertion, and the specific concerns apply to the choice of LLPS as the phenomenon when quite a bit of work over the past few years has clearly shown that the prefix LL is likely misleading. The phenomena combine continuous associative phase transitions and first order segregative phase transitions, and does not involve the type of separation evident in aqueous two-phase separation or the separation of oil from water. Semantics are important because they impart rigor to the process being described. Phase separation is driven by solubility considerations, percolation by multivalence. The two are coupled in macromolecules.

Approach:

Answer:

Thank you for bringing up this important issue. We modified the text to clarify this point in **Line 305-306** and also enclosed the corresponding references. Thereby we attribute to the complexity of the topic.

6) On line 197, the authors describe data pertaining to the abrogation of phase separation in the presence of excess RNA. These data are not to be found in the MS. Does the abrogation pertain to reentrant behavior tied to an imbalance of charge, as proposed by Banerjee and colleagues (<https://onlinelibrary.wiley.com/doi/abs/10.1002/anie.201703191> and <https://pubs.acs.org/doi/abs/10.1021/jacs.9b03689>) or alternatively, is this a manifestation of a pure stoichiometry imbalance seen by Rosen and coworkers because the phase behavior is driven purely by heterotypic interactions? To answer the latter question, it will be important to reanalyze the data using a formalism where the axes are not molecular concentrations, but instead are concentrations of complementary motifs. This, as shown recently, will help with identifying whether the phase behavior is driven by purely heterotypic interactions and also help uncover the balance of heterotypic and homotypic interactions (<https://www.nature.com/articles/s41467-023-43489-4>). In its current form, the analysis in favor of purely heterotypic interactions, especially given what follows, will have to be seen as being incomplete.

Answer:

Thank you for emphasizing the importance of this issue. We now generated a more detailed phase diagram for protein and RNA interactions. In the phase diagram we added the information of the valence indicating binding sites in both RNA and protein (**Fig. 2K-M**). In the phase separation assays, we used 4 model RNAs, all containing two predicted IGF2BP1 binding sites. We can not exclude the fact that charge imbalance contributes to the dissolution of RNPs at higher RNA concentrations. The phase behavior in our system is driven purely by heterotypic interactions. Therefore, we anticipate that it is highly likely that stoichiometric imbalance yields this behavior. We have now clarified this in the text; please see lines 232-240.

7) On line 202, the authors pivot to describing data for binding to model RNAs. It would help to motivate this pivot and explain the specific choices made.

Answer:

We are grateful for your attention to this vital issue. We added a reference and elucidated our intentions. As shown by Wächter in 2013, the KH domains contribute differently to

IGF2BP1 binding to different RNAs in in vitro binding assays. Thus, we first investigated how IGF2BP1 interacts with our model RNA to be able to decipher the effects of phosphorylation on protein-RNA interaction and phase separation.

8) A puzzler for many is the "avidity effect". This is invoked by the authors, but a quantitative analysis of this effect is missing. The challenge lies in defining and analyzing avidity. Does this refer to incomplete dissociation because of the spatial proximity of equivalent or competing binding sites? Does it refer to a mean-field contribution from the high local / effective concentration of binding sites? Does it refer to positive cooperativity? Or does it refer to the convoluted descriptions one sees in the literature where avidity is described via a combination of kinetics and thermodynamics? This is important in the current context because the authors lean fairly heavily on this concept for interpreting some of their results.

Answer:

Thank you for highlighting this significant issue. In the manuscript, we used the term "avidity" to describe the stronger binding of an RNA-binding protein (RBP) with multiple RNA-binding domains to a model RNA with multiple binding sites compared to the binding of its individual domains to the same RNA. In our case, the most likely explanation for the avidity is two-fold: i. due incomplete dissociation because of proximal binding sites ii. Secondly, as the referee mentioned, the binding of the protein to one site in RNA increases the effective concentration for the subsequent interactions (Erlendsson and Teilmann 2021). We clarified this point in the results and discussion sections (**Line 264- 266, Fig. Supp 3O**).

9) The data in Figures 2E and 2F are compared in terms of molar equivalents of concentration. However, the molar concentrations of RNA motifs are different. Additionally, there are clear differences in the concentration / stoichiometry dependencies between the two scenarios. These differences, if analyzed quantitatively, can be used to paint a clearer, thermodynamic interpretation of the contribution of different domains and motifs to the overall phase behavior. This would be very helpful. The comment is germane given the conclusion regarding necessity and sufficiency. This conclusion leaves one wondering about the contributions from regions outside KH3-4. It is also relevant, given the somewhat puzzling inference drawn regarding the GEEG mutant in the full-length protein. The authors argue that 185.8 nM is of high affinity. However, this is not true when compared to nearly 5-fold or higher difference when compared to the WT protein. These small differences can have multiplicative effects when thinking about the impact on percolation thresholds, the relevant parameter when thinking about the contributions of multivalent interactions. So, the conclusions drawn on lines 239-242 are puzzling. Please also note that it is very difficult to draw the types of definitive inferences made in the MS without seeing a full phase boundary.

Answer:

Thank you for bringing this up. All RNP granule formation assays, except for Fig. Supp. 3L is performed with RNAs that contain two binding sites (see also the reply to point 6). These experiments differ in the amount of RNA-binding proteins, binding mutants (GEEG), and phosphomimetic mutations. The affinity of the GEEG mutant for XBP1 36 nt is 185.8 nM and is comparable to wild-type IGF2BP1 ($K_{1/2} = 311.7$ nM). Above that, the binding of the GEEG mutant to the XBP1 201 nt is also similar ($K_{1/2} = 64.74$ nM) to the wild-type ($K_{1/2} = 41.01$ nM). While the KH34 RNA binding mutation does not affect high-affinity interactions with the RNA, the mutant impairs the formation of IGF2BP1 – RNA granules, indicating that this mutant is abolished in forming the protein-RNA networks under the experimental conditions we tested.

Supporting this notion, our EMSA assays show that higher-order interactions that appear at 500 nM in the wild-type IGF2BP1 are absent in the GEEG double mutant (**Fig. Supp. 3H**). We agree with the referee that we can not exclude the possibility that this mutant can still form condensates at higher protein and RNA concentrations that were not tested here. Yet, it clearly shows that the phase boundary is shifted. Importantly, in line with our in vitro results, this mutant does not localize to stress granules in cells (Wächter et al., 2013).

10) There is a remarkable result, which is only emphasized in passing in the MS. This pertains to the data in Fig. Supp 4E vs. Fig. Supp 4F. The authors finding it remarkable, which it indeed is, because this shows very clearly the separation between percolation sans phase separation and the contribution of regions outside KH1-4 to the coupling of percolation and phase separation that gives rise to condensate formation. How this happens will be very interesting to investigate, although one would stipulate, rightly so, that it is beyond the scope of the current MS. However, this is a really important finding. The truncated version drives the formation of a system-spanning network, referred to as a meshwork. In contrast, the full length protein undergoes percolation and phase separation, giving rise to, presumably, a condensate-spanning network. These results are hard to come by in the literature and given their utmost importance in clarifying the nature of the phenomenology, it would be useful to move these data to the main text, and to emphasize the nature of what is being observed using rigorous and conceptually accurate terminology.

Answer:

Thank you for pointing out this crucial aspect. Our data (e.g., DLS shown in Fig. 2K) suggests that the formation of IGF2BP1-containing granules follows the theory of phase separation coupled to percolation (Mittag and Pappu 2022, Kar et al. 2022). To emphasize this, we moved this figure to the main figures (Fig. 2G) and addressed the phenomenon more thoroughly throughout the manuscripts (Lines**296-299** and **804-823**).

11) The driving forces for condensate formation are analyzed mainly in terms of the sizes of condensates. It would be more comforting to have the threshold solubility product (<https://elifesciences.org/articles/67176>) i.e., threshold joint concentration of protein AND RNA required for phase separation for the unphosphorylated protein and the phosphomimic. Absent this, one does not know what to make of the quantified differences in sizes of condensates, because one does not know the magnitude of the quench depth into the two-phase regime. Knowledge of the threshold product, helps with the analysis whereby one can then, if one wishes, quantify condensate sizes at equivalent concentrations, which may or may not pertain to different degrees of supersaturation, or quantify condensate sizes at equivalent degrees of supersaturation if the thresholds are different.

Answer:

Thank you for pointing out this significant concern. To determine the effect of the phosphomimetic mutations in terms the saturation concentration, we obtained data at different concentrations of protein and RNA generate a more detailed phase diagram, please see **Fig. 2K-M**. For the phase diagram, we defined “condensates” as particles visible in bright field microscopy. We excluded particles that were not evenly distributed over the wells, these are common artefacts from either the manufacturing or also the coating process, and aggregates. We could not determine any difference between the wild-type and the S181E mutant at the phase boundary, however close to the phase boundary at 1.5 μ M protein and 1.0 μ M RNA as well as 2.0 μ M protein and 2.0 RNA concentration the wild-type protein formed higher number and larger condensates compared to S181E. Instead, the Y396E

mutant showed a slightly shifted phase boundary where Y396E formed condensates at 1.5 μM protein and 1.5 μM RNA concentration with the XBP1 36 nt RNA, while the wild-type did not. Based on the referee's suggestion, we investigated cluster formation in sub-saturation conditions and found no differences between wild-type IGF2BP1 and the phosphomimetic mutants (**Supp. Fig.4T**). We hypothesize that sub-saturation cluster formation and, thus, the threshold for phase separation coupled to percolation is facilitated mainly by the folded domains of the protein. The effect of the linker seems to regulate properties of the condensates, their formation kinetics and dynamics, which we demonstrate in the turbidity assay, and the properties of the granules as shown by FRAP.

12) An important aspect that requires quantification is the presence of clusters in sub-saturated solutions (defined as concentrations below the threshold concentration - joint of course - for phase separation). Recent studies have shown that such clusters exist for proteins alone (<https://www.pnas.org/doi/full/10.1073/pnas.2202222119> and <https://www.nature.com/articles/s41467-023-40540-2>) and for protein-RNA mixtures (<https://www.pnas.org/doi/10.1073/pnas.2120799119>). One would expect that these clusters, referred to by some as pre-percolation clusters, will likely influence the driving forces for condensation, and explain the abrogation of phase separation studied under identical conditions / concentrations. At a minimum, it would help to acknowledge this possibility in the discussion as a direction to pursue in future work.

Answer:

We greatly appreciate your valuable feedback on this matter. By applying DLS measurements, we now demonstrate in **Fig. Supp. 4T** that all IGF2BP1 full-length constructs form small clusters in the presence of RNA in sub-saturation concentrations. Interestingly, under those conditions, the saturation concentration for wild-type IGF2BP1 and the phosphomimetic mutants are almost identical conditions as depicted in the phase diagram; please see Fig **2K-M**. While the phase diagram shows a similar saturation threshold, we see distinct differences in the size of the condensates and biophysical properties of IGF2BP1 in the condensates. Based on our results, we propose that lower dynamics in the S181E phosphomimetic mutant measured in FRAP assays impact their growth and thereby result in the formation of smaller granules.

13) Kinetics of phase separation are difficult to compare across constructs if the concentrations are kept the same, and the driving forces, i.e., threshold concentrations are different. In such cases, it becomes important to measure kinetics at equivalent degrees of supersaturation, because this is the only way a molecular perspective emerges regarding the differences in mechanisms. Otherwise, the kinetics data become qualitative because they are a convolution of somewhat trivial concentration-dependent effects, that are cleanly unmasked using equivalent degrees of supersaturation, and the heights of barriers that are system-specific, which one wishes to unmask. The request is for these data to be de-emphasized and moved to the SI because their interpretation requires very different approaches and measurements.

Answer:

Your feedback on this observation is very important and we appreciate it greatly. As demonstrated in **Fig. 2K-M**, the saturation level is highly comparable for wild-type IGF2BP1 and the phosphomimetic mutants S181E and Y396E. While additional concentration-dependent effects cannot be excluded, this data suggests that the conditions used in these experiments allow us to compare phase separation kinetics. We put the in the supplementary and carefully rephrased our interpretation of the data; please see **Fig Supp. 4S** and the lines 374-375.

14) Does FRAP measure fluidity of condensates? This is a complex issue because what it measures is a convolution of the contributions from physical crosslinking and hence the lifetimes of “bonds” formed between molecules, and the contributions of molecular transport (typically viewed as diffusion). Condensates are viscoelastic materials, and therefore FRAP measures convolutions of the contributions from viscous and elastic moduli. Despite its widespread usage, FRAP is not a readout of fluidity. The critique here is a semantic one, and the request is to delete or deemphasize the assertion that FRAP measures fluidity. The data showing how G3BP1 influences FRAP dynamics and the fractions of mobile vs. immobile species are very elegant and compelling.

Answer:

Thank you for pointing out this significant concern. We agree with the referee that fluidity is not a parameter we can measure with FRAP. We have now rephrased the paragraph to be more clear and more precise in terms of the interpretation of the FRAP data; please see lines 379-385.

15) The back end of the MS makes a clear case for the presence of homotypic interactions among the IGF2BP1 molecules. There is precedent for substantial changes in charge there are local and non-local changes to interaction profiles that are mutually compensatory. Please see: <http://pubs.acs.org/doi/abs/10.1021/jacs.6b10272>. Might there be similar effects in play here? The analysis of the simulations would benefit from the inclusion of pair distribution functions that quantify the effective associations among the molecules represented in the simulations. Please note that linkers can contribute in two ways. Through the effective solvation volumes for sure, but also through interactions in cis and hence in trans as well with the substrate binding domains. This point, which the authors make very clearly and elegantly, was first reported in 2015 using very similar methodologies. Please see: <http://www.pnas.org/content/112/47/E6426.long>. Such effects have been reported for linkers in multidomain proteins, whereby even well-solvated linkers, when modelled as autonomous units, can have complementary interactions with folded domains. Please see: <https://www.sciencedirect.com/science/article/pii/S0022283618304078> and <https://doi.org/10.1016/j.crstbi.2021.08.002>. Such effects will influence the interplay of homotypic and heterotypic interactions as drivers of phase separation. Given the systematic probing of the effects of linker-domain interactions, there ends up being a bit of cognitive dissonance between the inferences made in the front end vs. the back end of the MS, which are left unreconciled in the discussion. It would help to remedy this, and also highlight synergies with other systems in the literature, including observations made by three different groups for the phase behaviors of G3BP1/2. In this context, one would recommend that the speculations starting on line 538 be deleted.

Answer:

We thank the reviewer for their careful consideration and valuable feedback. We agree that the SAXS experiments do not provide sufficient resolution to monitor all the possible conformational changes and their frequencies. Based on the suggestion of the Referee, we now included the analyses of pair distribution functions to quantify associations among linkers in the MD simulations (**Fig. Supp. 6E,F**). Our studies showed an exciting phenomenon, where the linker 2 Y396E mutant showed an enhanced propensity to self-associate compared to the wild-type linker 2; in contrast, the linker 1 did not establish low-affinity contacts for both wild-type and the S181E mutant. We have now added this data and

the text to the manuscript, citing the related literature. Thanks a lot for the input; please see the lines 536-544.

16) Finally, the impressive corpus of data could be bolstered by a more expansive and inclusive discussion of many foundational papers, outside the IGF2BP1 literature, on which this work is built. The field of condensates could benefit from knowing about the current work, and the converse is true for the IGF2BP1 field as well. It is a testament to the maturing nature of the condensate field that contributions in this field are now cited as axioms sans citation. However, from a scholarship perspective, this is less than ideal or appropriate. This is easily remedied by including citations, and clarifying some of the verbiage to be in line with the emerging and established concepts in the condensate field.

Answer:

Your feedback is very important to us, and we appreciate it greatly. We tried to improve the manuscript in terms of semantics and physical concepts, adding seminal and recent papers. Your input was very valuable in this effort, and we hope that the manuscript is now on par with the state-of-the-art in the field of phase separation.

Reviewer #4 (Remarks to the Author):

The manuscript by Hornegger et al. describes the implementation of a huge range of approaches to investigate the phosphorylation status of the protein IGF2BP1 under stress conditions, and determine how two of these phosphorylation modifications affects its ability to bind RNA and form condensates. The authors used mass spectrometry to identify the phosphorylation sites, and use phosphomimetic mutants S181E and Y396E to identify whether these mutations affect the RNA binding capacity of the full length protein, and identify no effects. It is shown that IGF2BP1 + RNA form RNP granules in vitro, and that this is decreased in the S181E mutant and increased in the Y396E mutant. The authors then used a very neat approach to study mcherry-tagged IGF2BP1 condensates in a cellular environment by adding recombinant G3BP1 into a whole cell lysate. Here they found that the S181E mutant resulted in more stable condensates than the WT, and that the condensates formed by Y396E were more dynamic. This also held true in cells.

The authors then went on to carry out some structural studies on IGF2BP1. SAXS revealed that the WT protein and the phosphomimetic mutants have an overall similar conformational ensemble. MD simulations suggested that the isolated linkers do not interact, and this was experimentally confirmed by NMR. Additional specific interactions were obtained by NMR, such as linker 1 binding to RNA and linker 2 binding to folded domains of IGF2BP1.

The manuscript contains a huge amount of information that is extremely difficult to digest, even after reading it many times. However, I believe that the authors have successfully identified phosphorylation states that occur under stress, identified how these phosphorylation states affect the macroscopic properties of RNPs and delineated the structural basis that underlie these RNP properties. The work is original, and the conclusion are well supported by the results. The methodology is sound and contains enough detail for reproduction of the experiments.

Overall, I believe that the work is of high enough quality and originality to warrant publication in Nature Communications, but the narrative could be greatly improved to increase

readability and hence interest from readers. The text could be more succinct. As a suggestion, for each section it would be easier to read if the main results were described first, for quick understanding, and the detailed SI data were described after for those that are interested. The use of many methods to support the same findings is of course scientifically strong, but challenging to understand for the reader. I appreciate it's a difficult thing to balance.

We greatly value your positive feedback and critical input on our manuscript. With this revision, we aimed to improve and clarify our phrasing to enhance the readability and precision of our statements.

Minor points-

Paragraph beginning at page 3 line 122. Please understand that you're comparing MS intensity of the phosphorylated vs. non-phosphorylated peptides, which may not directly correlate to protein amounts. Therefore please change all mentions of changes in amount of phosphorylation to changes in signal corresponding to phosphorylated peptides. 64% of signal intensity is unlikely to equate to 64% of protein molecules due to changes in ionisation efficiency when adding a negative charge to the peptide.

Thank you for pointing out this crucial aspect! We changed the phrasing in that paragraph to be more precise.

Please provide the sequences of all recombinant proteins used in the study, including those of the isolated domains.

Answer:

Thank you for bringing up this important point. We added a table with the amino acid sequences of all proteins appearing in this manuscript; please see Table 9 for further information.

REVIEWERS' COMMENTS

Reviewer #1 (Remarks to the Author):

The authors added new data to make the study more functionally relevant. i am happy with the manuscript in this form.

Response: We are very happy that we could address the concerns of the Reviewer #1.

Reviewer #2 (Remarks to the Author):

The revised study by Hornegger et al. addressed most of the previously raised concerns raised by referees. In sum, this supports the previous notion that the IDRs of IGF2BP1 and their post-translational modification has a severe impact on modulating the properties of IGF2BP1-RNA association and mRNP formation under normal as well as stress conditions.

Most notably, the authors for the first time – to the best of this referee's knowledge – provide evidence by MS that Y396 is phosphorylated by SRC, at least in vitro. However, the authors need to indicate this in the main text more precisely. Currently, the reader is left guessing why the authors continue with this modification although it is not affected by cellular stress nor is strikingly abundant at steady state in non-stressed cells.

Response: We thank the Reviewer #2 for their assessment. We have now edited the text to emphasize the results of the in vitro phosphorylation assays clearly; please see lines 144-147. We have also clarified our motivation to move on with the characterization of the Y396E phosphomimetic mutant; please see lines 161-162.

The bulk of mostly concise data provided by the authors is beneficial and this referee appreciates the author's efforts in addressing the various concerns by various new or substantially revised investigations. However, the presentation of findings is quite extensive and hard to digest. Thus, extensive editing of the manuscript (not more data) are required to support readers in following conclusions and key findings.

Response: We have now extensively edited the manuscript to make the key findings and conclusions more accessible to the readers.

One remaining conceptual weakness of the study, however, is that despite the biophysical insights provided, it remains vague if and how PTMs in IDRs ultimately affect physiological regulation – not the binding (!) - of target mRNAs by IGF2BP1 and consequently gene expression. This must be addressed in view of the title and abstract claiming that PTMS in IDRs RNA metabolism, which at present is not addressed beyond binding. In sum, the revised manuscript provides an extensive work on biophysical properties of PTMs, IDRs and folded RNA-binding domains but partially lacks the extraction of key insights on how these ultimately affect target mRNA fate under normal and or stressed conditions. These aspects need to be addressed by revision.

Response: To address the regulation of RNA metabolism, we performed RNA-seq analyses of cells expressing IGF2BP1-phosphomimetic mutants. Our data show a clear impact of these mutants on the levels of IGF2BP1 target RNAs, indicating their regulatory role. We clarified these points in the revised manuscript.

Reviewer #3 (Remarks to the Author):

The authors have, in my view, set a new standard for being responsive in terms of the revisions they have made, their overall scholarship, and the new data / analyses they have included in the revised MS. These types of responses make a detailed review and the investment of time well worth the effort. I have no further revisions to suggest or requests to make. This MS deserves to be published in its current form, and it will be viewed as a very important contribution to the field of phase separation and beyond.

Response: We are glad we could address Reviewer #3's concerns during revision, and we are grateful for their careful assessment and invaluable input in this process.

Reviewer #4 (Remarks to the Author):

The authors have done a great job at restructuring the manuscript and making it easier to follow.

Response: We thank Reviewer #4 for their assessment and are pleased to be able to address their concerns during revision.